



**Deglaciation of the Sierra Nevada (USA) during Heinrich Event 1**
Richard A. Becker[1], Aaron M. Barth[2], Shaun A. Marcott[3], Basil Tikoff[3], and Marc W. Caffee[4,5]
[1]Department of Earth and Environment, Boston University, 685 Commonwealth Ave, Boston,
MA 02215, USA
[2]Department of Geology, Rowan University, 201 Mullica Hill Rd, Glassboro, NJ 08028, USA
[3]Department of Geoscience, University of Wisconsin-Madison, 1215 W Dayton St, Madison, WI
53706 USA
[4]Department of Physics and Astronomy, Purdue University, 525 Northwestern Ave, West
Lafayette, IN 47907, USA
[5]Department of Earth, Atmospheric, and Planetary Sciences, Purdue University, 550 Stadium
Mall Dr, West Lafayette, IN 47907, USA
*Correspondence to:* Richard A. Becker (rabecker@bu.edu)

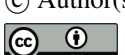



**Abstract** (≤250 words)
A polar jet stream (PJS) split by the Laurentide Ice Sheet (LIS) is a well-established
feature of Ice-Age atmospheric circulation. California's central Sierra Nevada Mountains (37–
38° N) lie near the reconstructed position of the PJS's southern branch. Previous studies
concluded that rapid deglaciation began here at ca. 16–15 ka after millennia of relatively stability
at ~60% LGM length. However, this conclusion is largely based on the behavior of glaciers in a
single valley, Bishop Creek Canyon. We report 31 new $^{10}$Be samples from two new locations –
Lyell Canyon and Mono Creek Canyon – and 26 recalculated $^{36}$Cl dates from Bishop Creek
Canyon ($n = 57$). These dates indicate rapid deglaciation began at $16.4 \pm 0.8$ ka and lasted for ca.
1.0 kyr. Placing two previously published paleoenvironmental reconstructions (Swamp Lake and
McLean's Cave) with centennial-or-better-scale resolution on new age-depth models that provide
age-uncertainty estimates, we find evidence for warming in the central Sierra Nevada at $16.4 \pm$
0.4 ka and drying at $16.20 \pm 0.13$ ka. Collectively, we interpret that rapid deglaciation began at
$16.20 \pm 0.13$ ka. This timing is indistinguishable from that of Heinrich Event 1 (HE1), which
occurred between $16.22 \pm 0.04$ ka and $16.04 \pm 0.04$ ka. We hypothesize that the Sierra Nevada's
deglaciation was driven by a northward repositioning and focusing of the winter-storm track over
western North America in response to PJS reunification, bringing warmer and drier weather to
the central Sierra Nevada, and that PJS reunification occurred in response HE1 thinning the LIS.
**Keywords**
Polar Jet Stream, Laurentide Ice Sheet, Sierra Nevada Mountains, Yosemite National Park,
Heinrich Event 1, Be-10, Cl-36, Atmospheric Reorganization





**Graphical Abstract**

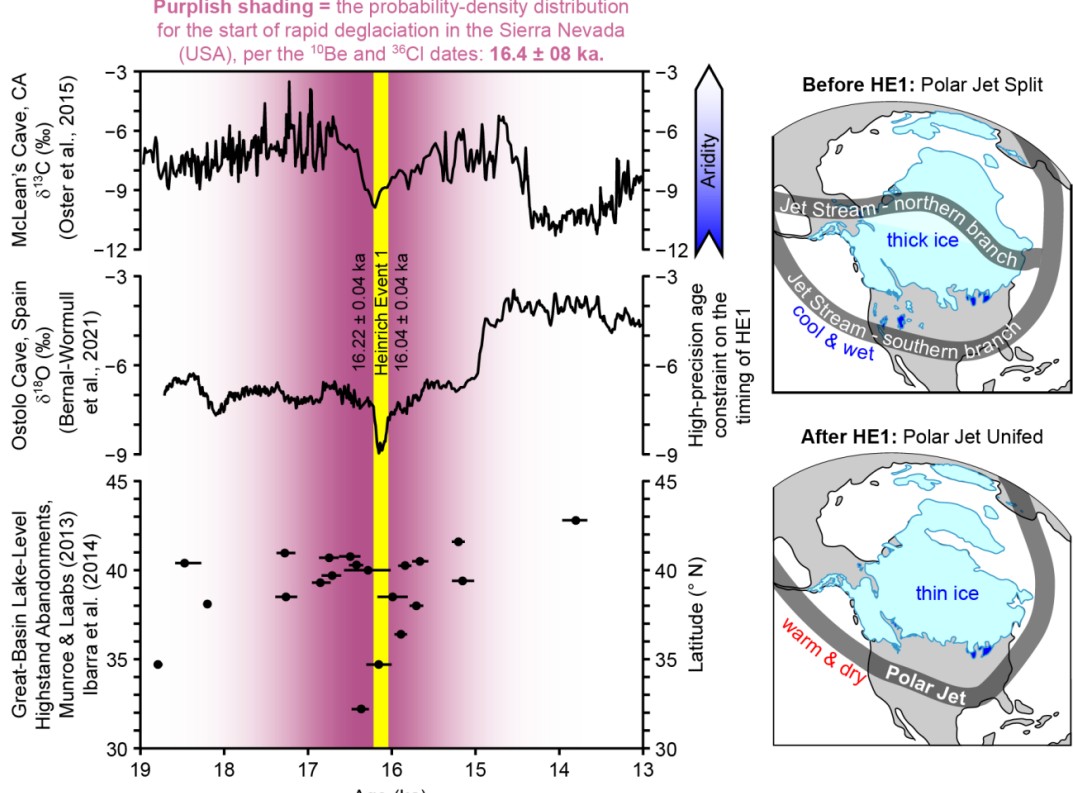





**Short Summary** (≤500 characters including spaces)
We report 31 new $^{10}$Be and 26 recalculated $^{36}$Cl dates from the Sierra Nevada Mountains (USA)
and conclude that deglaciation's final and rapid phase began at $16.4 \pm 0.8$ ka. In comparing this
timing with high-resolution regional paleoclimate proxies, we interpret that rapid deglaciation
most likely began at $16.20 \pm 0.13$ ka, which is indistinguishable in timing from Heinrich Event 1.
We interpret that the range's deglaciation was likely driven by a reunification of the polar jet
stream at this time.





## 1. Introduction

Abrupt, millennial- and centennial-scale climate changes are a prominent characteristic of the last deglaciation in contrast to the relative stability of the Holocene (e.g., Mayewski et al., 2004; Denton et al., 2010; Clark et al., 2012; Shakun et al., 2012; Marcott et al., 2013). Numerous climate reversals of varying extents and magnitudes punctuated the latest Pleistocene, including the Younger Dryas (e.g., Mangerud et al., 1974; Mangerud, 2021), the Antarctic Cold Reversal (Jouzel et al., 1995), and the Heinrich and Dansgaard-Oeschger events (Heinrich, 1988; Dansgaard et al., 1993). Understanding these events – including their rate, timing, magnitude, geographical extent, duration, and ultimately their causes and consequences – has been a principal endeavor within Quaternary science (e.g., Alley, 2000; Ganopolski and Rahmstorf, 2001; Hemming, 2004; Douglass et al., 2006; Firestone et al., 2007; Ackert et al., 2008; Putnam et al., 2010a; Laabs et al., 2011; Rasmussen et al., 2014; Bahr et al., 2018; Barth et al., 2018; Pedro et al., 2018; Li and Born, 2019).

Field-based studies in the Sierra Nevada Mountains (USA) – such as geomorphic observations (Clark, 1976; Clark and Clark, 1995) and $^{10}$Be, $^{26}$Al, and $^{36}$Cl surface-exposure dating (James et al., 2002; Phillips et al., 2009) – indicate that the range's glaciers responded strongly to a large and abrupt change in climate near the end of the last deglaciation. After Sierra Nevadan glaciers reached their greatest extents, broadly in-phase with the global Last Glacial Maximum (LGM; Phillips et al., 2009; Rood et al., 2011), they receded for several thousand years – with episodic advances superimposed on this general trend – before readvancing to ~60 % LGM-length. Glaciers in the Sierra Nevada rapidly melted following this readvance (Clark, 1976; Clark and Clark, 1995), receding from ~60 % LGM-length to <3 km length (0–23 % LGM-length) in "500 yr or less, and certainly less than 1000 yr" (Phillips et al., 2009, p. 1031).



This rapid ice recession may have been driven by a sharp reduction in precipitation either
contemporaneously with or immediately following Heinrich Event 1 in the North Atlantic
(Benson et al., 1996; Phillips et al., 1996).

This readvance and subsequent deglaciation suggest two reorganizations in atmospheric

circulation over the northeastern Pacific Ocean. An earlier reorganization that brought cooler
summers and/or wetter winters to the Sierra Nevada (driving the readvance) and a later, larger
magnitude and more permanent reorganization that brought warmer summers and/or drier
winters to the range (driving the deglaciation). This inference, derived from the Sierra Nevada's
geomorphology, is supported by a regional reconstruction of sea-surface temperatures (SSTs) in
the northeast Pacific that records cooling between ca. 18 ka and ca. 17 ka and warming after ca.
16.5 ka (Praetorius et al., 2020). Numerical modeling shows these cooler SSTs would have
generated a deeper and longer lasting snowpack in the Sierra Nevada, with warmer SSTs having
the opposite effect (Peteet et al., 1997).

The duration of the Tioga 4 deglaciation provides a maximum constraint on the duration

of the second reorganization. However, the cosmogenic surface-exposure ages constraining the
duration of this deglaciation to <500–1000 years (Phillips et al., 2009) are all associated with a
single former glacier – the Bishop Creek Glacier – thus permitting the possibility that glacial
recession was substantially slower (or more rapid) elsewhere in the range.

Here, we investigate the timing and duration of the final deglaciation in the Sierra

Nevada with thirty-one new [10]Be-concentration measurements from two other drainage basins in
the central portion of the range: (1) Tuolumne Meadows and Lyell Canyon in Yosemite National
Park and (2) Lake Thomas A. Edison (which was Vermillion Valley prior to 1954 C.E.) and
Mono Creek Canyon in the Sierra National Forest (Fig. 1). In converting these [10]Be





concentrations into surface-exposure ages, we consider multiple sources of uncertainty –
especially with regards to the snow-shielding correction.

Additionally, we further constrain the timing and nature of the climate changes that drove

the range's deglaciation – beyond the constraints imposed by the thirty-one new [10]Be dates – by
reevaluating four legacy datasets. In particular, we (1) recalculate twenty-six [36]Cl dates on the
timing and duration of the deglaciation within the vicinity of Bishop Creek Canyon (Phillips et
al., 2009) using updated [36]Cl production rates and scaling schemes (e.g., Marrero, 2012; Lifton et
al., 2014; Marrero et al., 2016), (2) recalibrate the age-depth model for the Swamp Lake
paleoenvironmental reconstruction (Street et al., 2012) and (3) the radiocarbon dates reported by
Munroe and Laabs (2013) and Ibarra et al. (2014) on the highstands of seventeen Great Basin
Pleistocene lakes with IntCal 20 (Reimer et al., 2020) and Bchron 4.7.6 (Haslett and Parnell,
2008), and (4) recalculate the age-depth model for speleothem ML1 from McLean's Cave (Oster
et al., 2015) using Bchron. We find evidence of warming and drying in the Sierra Nevada and
adjacent regions ca. 16.2 ka and interpret these changes as resulting from a northward shift and
focusing of the winter-storm track over North America in response to Heinrich Event 1, which is
indistinguishable in timing from the start of the Tioga 4 deglaciation (Bernal-Wormull et al.,
2021; Pérez-Mejías et al., 2021).





**Figure 1.** Maps showing the locations of the field areas and relevant data sets. **(a)** Location of
the central Sierra Nevada Mountain range (panel b) in the western United States. Pleistocene
lakes mentioned in the text are abbreviated as follows (north to south): LCh – Lake Chewaucan;
LS – Lake Surprise; SL – Spring Lake; OL – Owens Lake; LE – Lake Estancia; and LC – Lake
Cochise. For the names of the unlabeled lakes, please see Munroe and Laabs (2013). **(b)**
Location of the field areas within the central Sierra Nevada. Abbreviations (north to south): GC



of the T – Grand Canyon of the Tuolumne River; YV – Yosemite Valley; and SF SJR – South
Fork of the San Joaquin River. **(c)** Map of the Tuolumne Meadows and Lyell Canyon field area.
Sampling-location abbreviations (from distal to proximal): LM – lower Tuolumne Meadows; PD
– Pothole Dome; LD – Lembert Dome; TM-LC – unnamed roche moutonnée near the junction of
Tuolumne Meadows with Lyell Canyon; BLC – bottom of Lyell Canyon; MLC – a mid-
elevation within Lyell Canyon; and TLC – top of Lyell Canyon. Other abbreviations on panel c:
LVC – Lee Vining Canyon; TP – Tioga Pass; and GC of the T – Grand Canyon of the Tuolumne
River. **(d)** Map of the Lake Edison and Mono Creek Canyon field area. Sampling-location
abbreviations (distal to proximal): LEMouter – the outermost moraine beneath Lake Edison
(which was created in 1954 C.E. by the damming of Mono Creek and the flooding Vermillion
Valley); LEMinner – the innermost moraine beneath Lake Edison; lowMC – lower Mono Creek;
midMC – middle Mono Creek; and uppMC – upper Mono Creek. The other abbreviation on
panel d: PB – Pioneer Basin. **(e)** 2014 aerial photo of the Lake Edison moraine complex, which
was exposed by drought. **(f)** Map of Bishop Creek Canyon and vicinity. Sampling-location
abbreviations: T4m – the Tioga 4 moraine; HB – Humphrey's Basin; and T4-RP – scattered
sampling locations between the Tioga 4 moraine and the Recess Peak moraines. Other
abbreviations on panel f: NF BCC – the North Fork of Bishop Creek Canyon; MF BCC – the
Middle Fork of Bishop Creek Canyon; and SF BCC – the South Fork of Bishop Creek Canyon.





## 2.    Regional setting

### 2.1.    *Physical geography of the Sierra Nevada*

The Sierra Nevada is a ~600-km-long mountain range that parallels the western boundary

of North America and hosts the highest elevation within the contiguous United States: Mt.

Whitney at 4421 m (14,505 ft). The range formed during the Mesozoic as a result of subduction

and arc magmatism (Gilluly, 1969; Hamilton, 1969; McPhillips and Brandon, 2010; Gabet,

2014). Quartz-bearing lithologies are abundant in the central and southern Sierra Nevada. That

portion of the range is primarily granitic plutons with scattered metasedimentary and

metavolcanic wall rocks (e.g., Bateman, 1992).

The range's length, orientation, and elevation make it an effective barrier to maritime air

masses originating over the Pacific (e.g., Pandey et al., 1999). Furthermore, the range

experiences a Mediterranean climate with dry summers and wet winters characterized by

abundant snowfall. In Tuolumne Meadows for example (~2600 m elevation), 82% of its 1985–

2017 precipitation arrived in the six months between November 1 and April 30 and 50% of that

precipitation was snow (http://cdec.water.ca.gov). Average April snow depths at the elevation of

Tuolumne Meadows were approximately 2 m between 1946 and 2008 (Fig. S1).

### 2.2.    *Glacial history of the Sierra Nevada*

Within the Sierra Nevada, the last glaciation is termed the Tioga (Blackwelder, 1931) and

is divided into five glacial advances: from oldest to youngest these are Tioga 1–4 (Phillips et al.,

1996; Phillips et al., 2009) and then the comparatively minor Recess Peak (Birman, 1964; Clark

and Gillespie, 1997). Tioga 1 predates both the local and global LGM (Phillips et al., 1996;

Phillips et al., 2009); Tioga 2 marks the local LGM, which was ca. 24–21 ka (e.g., Clark et al.,

2003; Phillips et al., 2009; Amos et al., 2010; Rood et al., 2011); Tioga 3 represents a readvance



to ~90 % the LGM ice extent ca. 23–20 ka (e.g., Schaefer et al., 2006; Phillips et al., 2009; Stock
and Uhrhammer, 2010); and Tioga 4 marks a readvance to ~60 % LGM ice extent ca. 16 ka
(Phillips et al., 2009; Phillips, 2016).

Rapid deglaciation of the Sierra Nevada began immediately after the Tioga 4 readvance

(Clark and Gillespie, 1997). During the subsequent Recess Peak readvance, which culminated ca.
13.3 ka (Phillips, 2016; Marcott et al., 2019), northward-flowing glaciers deposited moraines
~0–3 km from their cirque headwalls (~1–23 % LGM length; Moore, 1981; Clark and Gillespie,
1997; Clark et al., 2003) while southward-facing cirques rarely contain Recess Peak moraines
(Clark and Gillespie, 1997). This paper focuses the Tioga 4 readvance and the subsequent
deglaciation.
*2.2.1.  Geomorphic constraints on the magnitude and duration of the Tioga 4 deglaciation*

Multiple geomorphic observations suggest Sierra Nevadan glaciers rapidly melted

following the Tioga 4 readvance (Birman, 1964; Clark, 1976; Clark and Clark, 1995). First, other
than the relatively minor Recess Peak deposits (e.g., Birman, 1964; Clark and Gillespie, 1997;
Phillips et al., 2009; Marcott et al., 2019), the innermost end moraines of the Sierra Nevada − the
Tioga 4 moraines − are ~60 % the distance (15 and 22 km in Bishop Creek and Mono Creek
Canyons, respectively) from the cirques to the Tioga 2 moraines (Clark, 1976; Clark and Clark,
1995; Phillips et al., 2009). Distal of ~60 %, end moraines are common and bedrock basins are
typically sediment-filled; proximal of ~60 %, bedrock basins are water-filled, bedrock-outcrop
exposure is excellent, and glacial sediments are primarily isolated boulders and till patches
(Clark, 1976; Clark and Clark, 1995).

Second, within the Lake Edison moraine complex − which is an assemblage of

approximately eleven end moraines in the Mono Creek drainage at 63–58 % LGM-glacier length





(24–22 km; Figs. 1d–e and 2) – till thickness systematically decreases from the outermost to
innermost moraine; immediately behind the innermost moraine, till is absent for 4 km (Birman,
1964). Although Birman (1964) interpreted the decreased till thickness as a temporal evolution
toward lower debris concentrations in the Late Pleistocene glaciers of the Sierra Nevada, it is
also consistent with rapid deglaciation. Phillips (2016) correlated the entire Lake Edison moraine
complex with the Tioga 4 glacial advance and used the elevation difference between the
moraines and various unspecified cirque headwalls in the Mono Creek drainage to reconstruct an
equilibrium-line altitude (ELA) of 2800 m for the Tioga 4 advance. Based on Pioneer Basin's
cirque-floor elevation (~3400 m), this reconstruction implies the Tioga 4 deglaciation was driven
by a ≥600 m ELA rise. This ELA rise is a minimum estimate because the climatological
snowline may have risen above the elevation of the cirque floors.






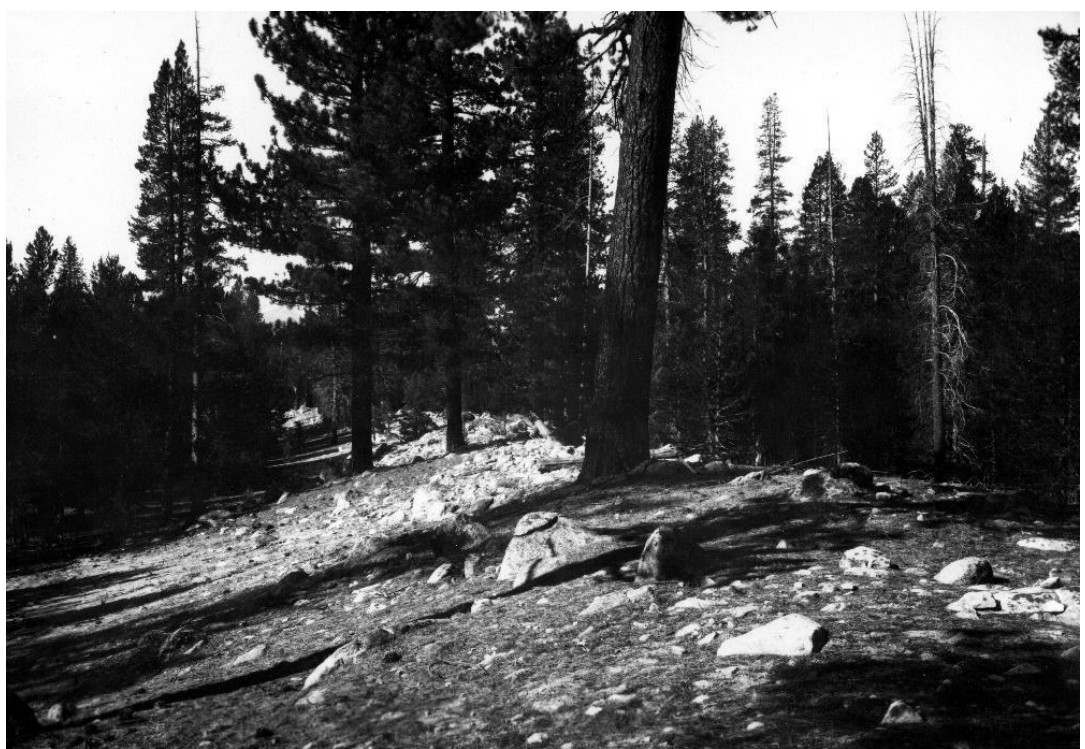

**Figure 2.** A 1908 photograph by G. K. Gilbert of one of the moraines within what is now the
Lake Edison moraine complex. Photo courtesy of the United States Geological Survey's
(USGS's) Denver Library Photographic Collection; item identifier ggk03340.



Finally, upvalley from the Tioga 4 end moraines − and presumably within the footprint of
the former Tioga 4 accumulation zone − low-relief ridges emanate from tributary-valley
junctions and parallel the inferred direction of former ice flow (e.g., Birman, 1964, Plate 1).
Clark (1976) observed numerous such ridges in the vicinity of Tuolumne Meadows (at ~30 %
LGM-glacier length) and Clark and Clark (1995) observed similar ridges in the upper reaches of
Mono Creek Canyon (at ~10 % LGM-glacier length). These low-relief, ice-flow parallel ridges
are inferred to be medial moraines (Clark, 1976; Clark and Clark, 1995). Based on the presence
of these medial moraines – and the absence of crosscutting end moraines and other ice-marginal
landforms – Clark (1976) and Clark and Clark (1995) interpreted that Sierra Nevadan glaciers
stagnated and rapidly melted at the end of the last glaciation.
*2.2.2.   Radiocarbon constraints on the timing of the Tioga 4 deglaciation*
Phillips (2016) reviewed radiocarbon constraints on the timing of the Tioga 4
deglaciation and placed this event at 15.75 ± 0.5 ka (1 σ). This determination was based on three
bulk-sediment radiocarbon dates from high-altitude ponds within the Tioga 4 glacial footprint
(Fig. 1b). From north to south, and when calibrated with Calib 8.2 (Stuiver and Reimer, 1993)
and IntCal20 (Reimer et al., 2020), these dates are 15.95 ± 0.29 ka from the Highland Lakes in
the North Fork of the Stanislaus River drainage (13,270 ± 200 [14]C-yrs; Clark et al., 1995), 15.69
± 0.09 ka from Greenstone Lake in the Lee Vining Creek drainage (13,090 ± 60 [14]C-yrs; Clark et
al., 2003), and 16.13 ± 0.10 ka from East Lake in the North Fork of the Kings River drainage
(13,400 ± 60 [14]C-yrs; Power, 1998; Fig. S2). Although a small dataset, these three dates span
~160 km of the Sierra Nevada and include locations north of, nearly adjacent to, and just south
of the new [10]Be dates we report. As these radiocarbon dates are from the basal layers of high-
elevation ponds − and there is no compelling evidence for the contaminating presence of "dead





carbon" (carbon without radiocarbon activity; a topic we discuss further in Sect. 5.2.1.) − these
dates suggest widespread deglaciation of the Sierra Nevada by ca. 16 ka, or in Phillips's (2016)
interpretation, at $15.75 \pm 0.5$ ka.

The youngest of these three minimum-limiting radiocarbon ages − the $15.69 \pm 0.09$ ka

date from Greenstone Lake (Clark et al., 2003) − is especially relevant to this manuscript.
Greenstone Lake is only 15–25 km north of Tuolumne Meadows and Lyell Canyon (Fig. 1c) and
erratic boulders deposited during the Tioga 4 deglaciation indicate that the Tioga 4 ice divide
was northeast of the modern hydrologic divide: ice flowed across Tioga Pass (TP on Fig. 1c)
from what is now the Lee Vining Creek watershed and contributed to the former Tuolumne
Glacier (Huber, 2007; Wahrhaftig et al., 2019). The bedrock geology requires these erratics to
have come from ≤4 km of Greenstone Lake (Bateman et al., 1983). After crossing Tioga Pass,
erratic boulder trains and streamlined landforms demonstrate (Wahrhaftig et al., 2019) these
originally-near-Greenstone-Lake parcels of ice passed through Tuolumne Meadows ~1–2 km
north of our sampling sites on Lembert Dome (LD on Fig. 1c) and in lower Tuolumne Meadows
(LM on Fig. 1c).

As a result of the Greenstone Lake radiocarbon date's calibrated-age-uncertainty

envelope, Greenstone Lake was both deglaciated and accumulating organic carbon by 15.42 ka
(Fig. S2; Haslett and Parnell, 2008; R Core Team, 2024). Thus, the Greenstone Lake radiocarbon
date (1) constrains the timing of deglaciation in a directly adjacent glacial system (that of Lee
Vining Canyon; LVC on Fig. 1c) and (2) suggests that Tuolumne Meadows was almost certainly
deglaciated (if not also revegetating) by 15.4 ka. We therefore adopt 15.4 ka as a minimum-
limiting constraint on the timing of deglaciation in Tuolumne Meadows and $15.75 \pm 0.5$ ka



(Phillips, 2016) as the best estimate for the timing of deglaciation in the high-elevation lake
basins elsewhere in the Sierra Nevada.
*2.2.3 Chlorine-36 constraints on the timing and duration of the Tioga 4 deglaciation*

Phillips et al. (2009) tested the geomorphic interpretation of an abrupt end to the last

glaciation in the Sierra Nevada by $^{36}$Cl-dating boulders and bedrock outcrops in the vicinity of
Bishop Creek (Fig. 1f). Applying then-current exposure-age calculation methods (i.e., Phillips et
al., 2001), five dates from the Tioga 4 moraine in the Middle Fork of Bishop Creek Canyon
("T4m" on Fig. 1e) yielded a mean age and standard deviation of 15.2 ± 1.0 ka, fifteen samples
scattered between the Tioga 4 moraine and the Recess Peak moraines yielded a mean age and
standard deviation of 14.7 ± 1.0 ka, and five samples from Humphreys Basin ("HB" on Fig. 1e),
a broad collection of cirques west of the modern drainage divide, yielded a mean age and
standard deviation of 15.2 ± 0.6 ka. Note that − although Humphreys Basin is west of the Sierra
Crest − striations indicate the Tioga ice divide was 0.5–1.0 km west of the modern hydrologic
divide in this location (Matthes, 1960; Phillips et al., 2009): Humphreys Basin contributed ice to
both the San Joaquin and Bishop Creek Glaciers. Based on these dates, Phillips et al. (2009)
concluded that the Tioga 4 deglaciation lasted <500‒1000 years.



**3.    Materials and methods**
*3.1.    Field areas and sample collection*

We sampled thirty-one boulders in two drainages on the western slope of the Sierra

Nevada (Figs. 1 and S3–S8; Tables S1–S3): (1) Tuolumne Meadows and Lyell Canyon in
Yosemite National Park and (2) Lake Thomas A. Edison and Mono Creek Canyon in the Sierra
National Forest.
3.1.1.  *Tuolumne Meadows and Lyell Canyon*

We collected eighteen samples from seven locations along the flowline of the former

Tuolumne Glacier in Yosemite National Park (Fig. 1c). From distal to proximal these are: (1) a
single sample from a boulder on a bedrock slope above lower Tuolumne Meadows (LM; 24 km
from the cirque headwall on Mount Lyell; Fig. S7), (2) three samples from Pothole Dome (PD;
22 km; Fig. S7), (3) three samples from Lembert Dome (LD; 18 km; Figs. S7 and S8), (4) a
single sample from an unnamed roche moutonnée near the junction of Tuolumne Meadows with
Lyell Canyon (TM-LC; 15 km; Fig. S6), (5) six samples from the bottom of Lyell Canyon (BLC;
6 km; ~2740 meters above sea level; Figs. S5 and S6), (6) a single sample from a mid-elevation
within Lyell Canyon (MLC; 4 km; ~220 m above the bottom of Lyell Canyon; Fig. S5), and (7)
three samples from the top of Lyell Canyon (TLC; 2 km; ~670 m above the bottom of Lyell
Canyon; Fig. S5). Collectively, these eighteen boulders range from 46 cm to 2.5 m tall and half
of the sampled surfaces (9 / 18) had glacial striations or similar evidence for no appreciable post-
glacial weathering (Table S1).

These samples provide minimum constraints on the timing and duration of the Tioga 4

deglaciation in the Tuolumne drainage because the location of the Tuolumne Glacier's Tioga 4
terminus is unknown and thus undated. Based on the Tuolumne Glacier's LGM length (94 km;





Dühnforth et al., 2010) and other Tioga 4 moraines within the Sierra Nevada being located at
~60 % LGM length (Clark, 1976; Clark and Clark, 1995; Phillips et al., 2009), the Tuolumne
Glacier's Tioga 4 terminus was likely in the lower reaches of the Grand Canyon of the Tuolumne
(Fig. 1b). If so, ice-marginal landforms recording the position of the Tuolumne Glacier's Tioga 4
terminus may not have been preserved, given the Canyon's steep slopes. We note that an
analogous situation was documented in Utah's steep-walled Big Cottonwood Canyon (Quirk et
al., 2018).
3.1.2. *Lake Thomas A. Edison and Mono Creek Canyon*

We collected thirteen samples from five locations along the flowline of the former Mono

Creek Glacier in the Sierra National Forest (Figs. 1d). From distal to proximal these are: (1) three
samples from the outermost end moraine beneath Lake Thomas A. Edison (LEMouter; 24 km
from the cirque headwall in Pioneer Basin; Figs. S4 and S5), (2) two samples from the innermost
end moraine beneath Lake Edison (LEMinner; 22 km; Fig. S4), (3) a single sample from the
lower portion of Mono Creek Canyon (lowMC; 15 km; Fig. S4), (4) two samples from the
middle portion of Mono Creek Canyon (midMC; 11 km; Figs. S3 and S4), and (5) five samples
from the upper portion of Mono Creek Canyon (uppMC; 1–5 km; Fig. S3). Note that Lake
Edison was created in 1954 C.E. with the construction of the Vermillion Valley Dam and that
approximately nine additional end moraines lie between the outermost and innermost Lake
Edison moraines (Figs. 1d, 2, and S9). Collectively, these thirteen boulders range from 35 cm to
2.4 m and 25 % of the sampled surfaces had striations or glacial polish indicating little-to-no
post-glacial erosion (Table S1).





### 3.2. Laboratory preparation


Samples were prepared at the University of Wisconsin-Madison. Collected rocks were
crushed to 425–841 μm, processed through a Frantz LB-1 Magnetic Barrier Separator to remove
magnetic grains, and etched in solutions of 2 % HF and 2 % $HNO_3$ while rotating on an APW
Wyott HR-50. Quartz purity was measured by ICP-AES analysis prior to ion exchange
chromatography. Samples with ≤200 ppm Al were dissolved in concentrated HF and $HNO_3$,
spiked with a $^9Be$ carrier solution (OSU-White; Table S3), and dried. Beryllium fluoride was
converted into $BeCl_2$ by repeatedly dissolving and drying the samples in $HClO_4$ (≥4x) before
redissolving them in 8 M HCl. Iron, Ti, and Ca were removed with anion column
chromatography (Bio-Rad AG1-X8 100–200 mesh chloride-form resin) and $NH_4OH$
precipitations. The Be fraction was separated using cation chromatography (Bio-Rad AG50W-
X8 100–200 mesh hydrogen-form resin) and precipitated in centrifuge tubes at a pH of ~8.5 by
adding $NH_4OH$. Finally, the samples were dried in quartz crucibles on a hotplate at incremental
heating steps of ~55 °C, ~70 °C, and ~180 °C before being incinerated at ~900 °C to produce
BeO, which was mixed with Nb and packed into AMS targets. Isotopic ratios were measured at
PRIME Lab, Purdue University, in relation to the 07KNSTD standard (Table S3).

### 3.3. Selection of legacy data


To complement the new $^{10}Be$ samples from Yosemite National Park and the Sierra
National Forest, we include in our analysis twenty-six $^{36}Cl$ samples (Tioga 4) reported by
Phillips et al. (2009) from the vicinity of Bishop Creek (Fig. 1e): the five boulder samples from
the crest of the Tioga 4 moraine, sixteen boulder and bedrock measurements from locations
between the Tioga 4 and Recess Peak moraines, and the five boulder samples from Humphreys
Basin. Phillips et al. (2009) reported two additional $^{36}Cl$ measurements on a single sample





(BPCR91-11) from a roche moutonnée 1.2 km behind the Tioga 4 moraine − but both ages were
anomalously old (ca. 22 ka) and we preemptively excluded this sample from our reanalysis. We
also exclude the twenty-three bedrock [10]Be samples reported by Dühnforth et al. (2010) from
Tuolumne Meadows and Lyell Canyon because uncertainties regarding the degree of inheritance
and snow shielding. These samples were collected from small bedrock protrusions (e.g., roche
moutonnées) and the heights of these protrusions were not reported. Additionally, we exclude the
one [10]Be date and nine [26]Al dates reported by James et al. (2002) on the final deglaciation of the
northwestern Sierra Nevada because of imprecision in these early cosmogenic dates.
*3.4.  Snow-shielding correction*
Snow-shielding corrections for the thirty-one new [10]Be samples from Yosemite National
Park and the Sierra National Forest were calculated using the historical relationship (ca. 1946–
2008 C.E.) between monthly average snow water equivalent (SWE) and elevation (Fig. S21), the
modern relationship between average snowpack density and SWE (Fig. S22c), and equation 3.76
from Gosse and Phillips (2001):
$$S_{snow} = \frac{1}{12}\sum_{i}^{12} e^{-\left(\frac{(z_{snow,i} - z_{sample})\,\rho_{snow,i}}{\Lambda_{f,e}}\right)}, (1)$$

where $z_{snow,i}$ is monthly average snow depth, $z_{sample}$ is boulder height, $\rho_{snow,i}$ is monthly average
snow density, and $\Lambda_{f,e}$ is the effective attenuation length of the cosmogenic ray flux. The
historical relationship between monthly average SWE and elevation (Fig. S21) was derived from
all the active California Department of Water Resources (CA DWR) snow-monitoring stations in
the drainage basins of the Tuolumne, Merced, and San Joaquin Rivers with at least ten years of
observations for a particular month (Fig. 1b; Tables S4–S6). The median number of stations over
the six snowiest months (December through May) is forty-five. The relationship between
average snowpack density and SWE (Fig. S22c) was derived from 15,971 daily observations of





snow depth and SWE within these three drainage basins. We propagated uncertainties in boulder
height, snow depth, and snow density into the shielding uncertainties (Table S1). We assume
historical snow depths and densities are representative of the post-glacial interval and do not
apply a time-evolving snow-shielding correction (e.g., Ye et al., 2023). Additional details about
how we calculated the snow-shielding corrections are in the supplement (Sect. S.5.).

For the twenty-six legacy $^{36}$Cl dates on the Tioga 4 ice advance and subsequent

deglaciation from the vicinity of Bishop Creek, we used the snow-shielding correction factors
reported by Phillips et al. (2009), which they calculated using Eq. 1 in conjunction with data
from CA DWR snow-monitoring stations near Bishop Creek.
*3.5.    Exposure-age calculations*

Surface-exposure ages were calculated using version 3.0.2 of the Balco et al. (2008)

calculator with the underlying constants last updated on 3 March 2024 (the "Version 3"
calculator), CRONUScalc 2.0 (Marrero et al., 2016), and CREp (Martin et al., 2017). CREp does
not report a version number (Dr. Pierre-Henri Blard, personal communication, 16 July 2018);
these ages were calculated on 3 March 2024. While employing additional calculators provides
little-to-no leverage on the problem of systematic uncertainties within $^{10}$Be dating – which are
chiefly related to the spatial and temporal scaling of $^{10}$Be production rates (e.g., Balco, 2011;
Borchers et al., 2016) – we are unaware of any scientific rationale for preferring the results of
one calculator over another. Our rational for providing multiple calculations is to assess the
sensitivity of the various calculators on our primary conclusions.

Across all three calculators (Balco et al., 2008; Marrero et al., 2016; Martin et al., 2017)

we used LSDn scaling (Lifton et al., 2014) and the ERA-40 atmospheric model (Uppala et al.,
2005). We preferred the LSDn scaling model over the time independent (St) and dependent (Lm)



versions of Lal-Stone scaling (Nishiizumi et al., 1989; Lal, 1991; Stone, 2000; Balco et al., 2008)
because of LSDn's grounding in first principles and its ability to explain $^{10}$Be concentrations that
appear supersaturated with St scaling (Balco, 2016). With regards to the atmospheric model, the
Version 3 calculator and CRONUScalc 2.0 are both hardcoded to use ERA-40; for CREp, we
elected to use ERA-40 over the U.S. standard atmosphere (NOAA et al., 1976) because of ERA-
40's greater fidelity to observed atmospheric pressures (Balco, 2015). As for geomagnetic field
reconstruction, the Version 3 calculator is hardcoded to use Lifton (2016) while CRONUScalc
2.0 is hardcoded to use Lifton et al. (2014); for CREp, we adopted Lifton (2016).

We used each calculator's default production-rate-calibration dataset. For the Version 3

calculator and for CRONUScalc 2.0 this is the CRONUS-Earth primary $^{10}$Be calibration dataset
(Putnam et al., 2010b; Marrero, 2012; Kelly et al., 2015; Lifton et al., 2015; Phillips et al., 2016).
As such, the Version 3 calculator uses a non-dimensional LSDn fitting parameter of 0.838 ±
0.048; it does not report a sea-level high-latitude (SLHL) production rate in atoms g$^{-1}$ quartz yr$^{-1}$.
Balco (2018) discusses this reporting decision. CRONUScalc 2.0 uses a SLHL $^{10}$Be production
rate of 3.92 ± ~0.33 atoms g$^{-1}$ quartz yr$^{-1}$ (Borchers et al., 2016; Marrero et al., 2016); it does not
report an LSDn fitting parameter. CREp's default $^{10}$Be-production-rate dataset is every published
production-rate-calibration study without a major demonstrated flaw (Martin et al., 2017); on 3
March 2024 CREp used a SLHL production rate of 4.06 ± 0.23 atoms g$^{-1}$ quartz yr$^{-1}$ based on the
work of Gosse et al. (1995), Farber et al. (2005), Balco et al. (2009), Putnam et al. (2010a),
Fenton et al. (2011), Kaplan et al. (2011), Goehring et al. (2012), Young et al. (2013), Claude et
al. (2014), Kelly et al. (2015), Martin et al. (2015), Small and Fabel (2015), Stroeven et al.
(2015), and Putnam et al. (2019). CREp also does not report an LSDn scaling factor. While we
focus this manuscript's interpretations on the ages inferred from the globally distributed



production-rate datasets, we also consider the impact on our interpretations of calculating the
exposure ages with $^{10}$Be production rates derived from the Promontory Point calibration site
(Sect. 5.2.2; Lifton et al., 2015).
For the Version 3 calculator and CREp (Martin et al., 2017), the reported internal age
uncertainties solely reflect $^{10}$Be concentration uncertainty while the external uncertainties reflect
$^{10}$Be concentration and production-rate uncertainties. For CRONUScalc 2.0 (Marrero et al.,
2016), the internal age uncertainties reflect twelve sources of uncertainty (Tables S1–S3): sample
latitude, longitude, elevation, thickness, density, topographic shielding, erosion rate, $^{10}$Be
concentration, attenuation length, boulder height, modern SWE by month as a function of
elevation (Fig. S21), and average snowpack density as a function of SWE (Fig. S22c). The
external age uncertainties we report for CRONUScalc include these same twelve sources of
uncertainty plus the production rate and scaling uncertainties (Borchers et al., 2016).
We adjusted all reported surface-exposure ages such that they are relative to 1950 C.E. in
order to facilitate comparison with the calibrated radiocarbon dates. (The Version 3 calculator
and CREp report exposure durations prior to sampling; CRONUScalc 2.0 reports exposure ages
relative to 2010 C.E.) We also rounded all cosmogenic age uncertainties mentioned in the text up
to the next highest tenths position [when reported in units of kiloannum (ka)] in order to avoid
rounding the uncertainties down to lower values than justifiable (e.g., the 0.434 ka uncertainty
for sample TM14-15 in the CRONUScalc column of Table 1 becomes 0.5 ka in the text).
*3.6.   Outlier identification and group-age and uncertainty calculations*
For the locations where we report or recalculate at least three samples, we identify
outliers using an approach inspired by Tulenko et al. (2022): we first calculate the arithmetic
mean and standard deviation of all the surface-exposure ages from a particular location and then





assess – within the CRONUScalc 2.0 ages – whether any samples deviate by more than twice
their internal age uncertainty (>2 σ) from their group's arithmetic mean. If so, we label the most
extreme deviator from the arithmetic mean an outlier across all three calculators, recalculate the
mean, and assess whether any sample still deviates by more than two standard deviations from
this new mean. If so, we iterate the outlier identification and rejection protocol described above
until all remaining samples are within twice their internal age uncertainty of their group's mean
age. For the dataset we report here, iterating this approach more than twice was not required and
we only reject four samples out of the forty-nine samples (8 %) from locations where we report
three or more samples.

For the locations where we report two or fewer samples, we identify outliers using

geomorphic relative-age relationships with the larger sampling groups.

We use the CRONUScalc results to identify outliers for three reasons. First, applying this

algorithm across all three calculators might result in some samples being rejected within one
calculator's results and not those of another. Second, using the CRONUScalc results for outlier
identification is the most conservative implementation of this outlier-identification protocol
because the CRONUScalc results include more sources of uncertainty than the results of the
other two calculators. Finally, as of this writing, CRONUScalc is the only calculator of the three
that calculates $^{36}$Cl surface-exposure ages (although we acknowledge that both the Version 3
calculator and CREp have $^{36}$Cl-calculation methods in development). Thus, at the time of
writing, using the CRONUScalc results to identify outliers provides analytic uniformity across
the combined dataset of $^{10}$Be and $^{36}$Cl ages.



To calculate each sampling location's external age uncertainty, we multiplied each

group's standard deviation by the average ratio of the external-to-internal age uncertainties in
that group.
*3.7.    Deglaciation duration calculations*

We calculated deglaciation's duration within Lyell, Mono Creek, and Bishop Creek

Canyons using $1 \times 10^5$ trial Monte Carlo simulations. As input, these simulations used each
sampling group's mean surface-exposure age and the standard deviation of the ages in that
group. We then filtered the results of these Monte Carlo simulations and identified trials with
simulated deglaciation ages consistent with the geomorphic relative-age requirements of the
sampling sites (see Sect. S.6.).
*3.8.    Recalibration of previously published datasets*

We used IntCal20 (Reimer et al., 2020) and the R package Bchron 4.7.6 (Haslett and

Parnell, 2008; R Core Team, 2024) to recalibrate the radiocarbon ages in the ca. 18–10 ka
portion of the Swamp Lake paleoenvironmental reconstruction (Street et al., 2012) and the
radiocarbon ages that constrain the timing of Great Basin pluvial-lake highstands (Munroe and
Laabs, 2013; Ibarra et al., 2014). We also used Bchron to construct a new age-depth model for
speleothem ML1 from McLean's Cave (Oster et al., 2015). Our goal with the new age-depth
models was to provide age uncertainties for the undated horizons in the Swamp Lake core and in
speleothem ML1, as the original age-depth models do not provide age uncertainties at these
depth increments.



## 4.    Results

*4.1.    New ¹⁰Be dates from Tuolumne Meadows and Lyell Canyon*

We report eighteen new ¹⁰Be samples from Tuolumne Meadows and Lyell Canyon in
Yosemite National Park. After rejecting three outliers (Sect. S.7.), the fifteen remaining samples
yield ages ranging 15.5–13.9 ka (Version 3 calculator), 16.4–14.7 ka (CRONUScalc 2.0), and
16.3–14.7 ka (CREp; Table 1). Across all three calculators, the location with the oldest surface-
exposure age is lower Tuolumne Meadows (the most distal sampling site) and the youngest
location is the bottom of Lyell Canyon (an intermediate location, but the site with the greatest ice
thickness). In the Version 3 results, the sample (TM14-15) from lower Tuolumne Meadows has
an age of $15.2 \pm 0.3$ ka (1 σ internal-age uncertainty) and the six samples from the bottom of
Lyell Canyon have a mean and standard deviation of $14.4 \pm 0.3$ ka (Fig. 3b; Table 2). In contrast,
CRONUScalc and CREp produce ages and internal uncertainties of $16.0 \pm 0.5$ ka and $16.0 \pm 0.3$
ka, respectively, for the sample from lower Tuolumne Meadows – and means and standard
deviations of $15.1 \pm 0.2$ ka and $15.1 \pm 0.3$ ka, respectively, for the six samples from the bottom
of Lyell Canyon (Fig. 3c–d; Table 2). Thus, CRONUScalc and CREp are producing surface-
exposure ages for these samples that are ca. 0.8 kyr (5 %) older than the ages produced by the
Version 3 calculator.

However, all three calculators produce nearly identical median estimates for the duration
of the Tioga 4 deglaciation in Tuolumne Meadows and Lyell Canyon (i.e., the age difference
between the lower Tuolumne Meadows sample and the mean age of the samples from the bottom
of Lyell Canyon): 0.8 kyr with a 95 % confidence range of 0.2–1.5 kyr (Version 3 calculator),
0.9 kyr with a 95 % confidence range of 0.1–1.9 kyr (CRONUScalc 2.0; Fig. 3i), and 0.9 kyr
with a 95 % confidence range of 0.3–1.5 kyr (CREp; Fig. 3j).





**Figure 3.** Sampling locations in the Tuolumne Meadows and Lyell Canyon field area, along with the surface-exposure ages and deglaciation durations inferred from those samples. **(a)** Sampling locations in Tuolumne Meadows and Lyell Canyon. Sampling-location abbreviations (from distal to proximal): LM – lower Tuolumne Meadows; PD – Pothole Dome; LM – Lembert



Dome; TM-LC – an unnamed roche moutonnée near the junction of Tuolumne Meadows with Lyell Canyon; BLC – the bottom of Lyell Canyon; MLC – a mid-elevation within Lyell Canyon; and TLC – the top of Lyell Canyon. **(b–d)** The $^{10}$Be surface-exposure ages and 1 σ internal uncertainties. The gray boxes are the mean age of the retained $^{10}$Be ages plus/minus one standard deviation. Rejected outliers are depicted with hollow symbols. Note that the extreme outlier (ca. 22–25 ka) from Pothole Dome (TM14-13) is outside the age range shown here. **(e–g)** The surface-exposure ages and 1 σ external uncertainties. The yellow shading's intensity represents the relative probability of deglaciation, according to Phillip's (2016) interpretation of the minimum-limiting radiocarbon dates. **(h–j)** Histograms for the duration of deglaciation. Each histogram is based on a 1 x 10$^5$ trial Monte Carlo simulation. We defined "valid" trials as those where the simulated deglaciation age of lower Tuolumne Meadows (LM) and the top of Lyell Canyon (TLC) were both older than the bottom of Lyell Canyon's (BLC's) simulated deglaciation age.



**Table 1.** Individual [10]Be and [36]Cl sample surface-exposure ages across the three calculators.

| Calculator: | Balco et al. (2008) v. 3.0.2 (with 3 Mar 2024 constants) | | | Marrero et al. (2016) CRONUScalc 2.0 | | | Martin et al. (2017) CREp (on 3 Mar 2024) | | |
|---|---|---|---|---|---|---|---|---|---|
| Sample name | Age (ka) | Int. Err. (ka) | Ext. Err. (ka) | Age (ka) | Int. Err. (ka) | Ext. Err. (ka) | Age (ka) | Int. Err. (ka) | Ext. Err. (ka) |
| **New Be-10 samples from Tuolumne Meadows and Lyell Canyon** | | | | | | | | | |
| *Top of Lyell Canyon (TLC)* | | | | | | | | | |
| C2-Boulder1 | 15.085 | 0.206 | 0.932 | 16.040 | 0.436 | 1.300 | 15.928 | 0.210 | 0.840 |
| C2-Boulder2 | 15.294 | 0.364 | 0.991 | 16.140 | 0.598 | 1.400 | 16.168 | 0.350 | 0.900 |
| UpLy-12-10 | 14.484 | 0.228 | 0.902 | 15.240 | 0.460 | 1.200 | 15.238 | 0.210 | 0.790 |
| *Mid-elevation within Lyell Canyon (MLC)* | | | | | | | | | |
| *TM14-01* | *15.773* | *0.265* | *0.975* | *16.740* | *0.525* | *1.400* | *16.626* | *0.260* | *0.890* |
| *Bottom of Lyell Canyon (BLC)* | | | | | | | | | |
| LoLy-11-08 | 14.436 | 0.357 | 0.940 | 15.240 | 0.402 | 1.200 | 15.169 | 0.320 | 0.830 |
| TM14-02 | 14.432 | 0.244 | 0.892 | 15.140 | 0.418 | 1.100 | 15.146 | 0.220 | 0.780 |
| TM14-03 | 14.715 | 0.301 | 0.925 | 15.440 | 0.367 | 1.100 | 15.426 | 0.290 | 0.810 |
| TM14-04 | 14.296 | 0.215 | 0.877 | 15.040 | 0.438 | 1.200 | 15.026 | 0.180 | 0.770 |
| TM14-05 | 14.040 | 0.335 | 0.900 | 14.840 | 0.454 | 1.200 | 14.796 | 0.300 | 0.790 |
| TM14-06 | 14.414 | 0.252 | 0.904 | 15.140 | 0.449 | 1.100 | 15.156 | 0.220 | 0.790 |
| *An unnamed roche moutonnée between Tuolumne Meadows and Lyell Canyon (TM-LC)* | | | | | | | | | |
| *TM14-07* | *16.233* | *0.284* | *1.019* | *17.140* | *0.575* | *1.400* | *17.086* | *0.260* | *0.900* |
| *Pothole Dome in Tuolumne Meadows (PD)* | | | | | | | | | |
| TM14-11 | 14.339 | 0.287 | 0.910 | 15.140 | 0.510 | 1.200 | 15.076 | 0.240 | 0.800 |
| TM14-12 | 15.473 | 0.269 | 0.970 | 16.440 | 0.442 | 1.300 | 16.306 | 0.240 | 0.880 |
| *TM14-13* | *22.761* | *0.474* | *1.460* | *23.940* | *0.997* | *2.000* | *23.866* | *0.430* | *1.250* |
| *Lower Tuolumne Meadows (LM)* | | | | | | | | | |
| TM14-15 | 15.207 | 0.239 | 0.936 | 16.040 | 0.434 | 1.300 | 15.996 | 0.230 | 0.860 |
| *Lembert Dome in Tuolumne Meadows (LD)* | | | | | | | | | |
| *TM14-16* | *13.941* | *0.271* | *0.872* | *14.740* | *0.413* | *1.100* | *14.696* | *0.240* | *0.770* |
| TM14-17 | 15.019 | 0.264 | 0.931 | 15.840 | 0.430 | 1.300 | 15.796 | 0.270 | 0.840 |
| TM14-18 | 15.098 | 0.249 | 0.932 | 15.940 | 0.370 | 1.200 | 15.896 | 0.250 | 0.850 |
| **New Be-10 samples from Lake Edison and Mono Creek Canyon** | | | | | | | | | |
| *Pioneer Basin and upper Mono Creek Canyon (uppMC)* | | | | | | | | | |
| MR-14-03 | 14.963 | 0.208 | 0.925 | 15.840 | 0.343 | 1.200 | 15.796 | 0.210 | 0.820 |
| MR-14-04 | 14.861 | 0.304 | 0.945 | 15.640 | 0.396 | 1.200 | 15.666 | 0.300 | 0.820 |
| MR-14-05 | 14.657 | 0.226 | 0.901 | 15.440 | 0.385 | 1.200 | 15.416 | 0.210 | 0.780 |
| MR-14-06 | 14.863 | 0.325 | 0.952 | 15.740 | 0.611 | 1.300 | 15.656 | 0.320 | 0.830 |
| MR-14-07 | 14.699 | 0.292 | 0.932 | 15.440 | 0.420 | 1.200 | 15.456 | 0.280 | 0.810 |
| *Mid Mono Creek Canyon (midMC)* | | | | | | | | | |
| *MR-13-02* | *20.077* | *0.312* | *1.234* | *21.040* | *0.505* | *1.400* | *21.027* | *0.270* | *1.000* |
| *MR-14-08* | *17.871* | *0.252* | *1.108* | *18.840* | *0.536* | *1.400* | *18.826* | *0.230* | *0.950* |
| *Lower Mono Creek Canyon (lowMC)* | | | | | | | | | |
| MR-13-01 | 15.136 | 0.433 | 1.009 | 16.040 | 0.594 | 1.400 | 15.997 | 0.420 | 0.920 |
| *Innermost moraine beneath Lake Edison (LEMinner)* | | | | | | | | | |
| LEM-14-14 | 15.534 | 0.366 | 1.005 | 16.440 | 0.532 | 1.400 | 16.406 | 0.340 | 0.910 |
| LEM-14-15 | 15.647 | 0.365 | 1.011 | 16.640 | 0.574 | 1.300 | 16.526 | 0.350 | 0.910 |
| *Outermost moraine beneath Lake Edison (LEMouter)* | | | | | | | | | |
| LEM-16-28 | 18.579 | 0.377 | 1.184 | 19.540 | 0.552 | 1.400 | 19.524 | 0.340 | 1.010 |
| *LEM-16-29* | *4.703* | *0.210* | *0.351* | *4.860* | *0.205* | *0.350* | *4.964* | *0.200* | *0.310* |
| LEM-16-30 | 19.421 | 0.336 | 1.221 | 20.340 | 0.520 | 1.400 | 20.374 | 0.290 | 1.010 |
| **Recalculated Cl-36 samples from Bishop Creek Canyon and vicinity (from Phillips et al., 2009)** | | | | | | | | | |
| *Tioga 4 moraine within the Middle Fork of Bishop Creek Canyon (T4m)* | | | | | | | | | |
| BpCR97-8 | N.A. | N.A. | N.A. | 14.740 | 0.869 | 1.500 | N.A. | N.A. | N.A. |
| BpCR97-9 | N.A. | N.A. | N.A. | 16.840 | 0.977 | 1.700 | N.A. | N.A. | N.A. |
| BpCR97-13 | N.A. | N.A. | N.A. | 17.040 | 0.927 | 1.800 | N.A. | N.A. | N.A. |
| BpCR97-14 | N.A. | N.A. | N.A. | 16.340 | 1.023 | 1.900 | N.A. | N.A. | N.A. |
| BpCR97-15 | N.A. | N.A. | N.A. | 16.440 | 0.996 | 1.800 | N.A. | N.A. | N.A. |
| *Scattered boulders and bedrock outcrops between the Tioga 4 (T4m) and Recess Peak moraines in Bishop Creek Canyon (SB&O)* | | | | | | | | | |
| BpCr95B-6(99) | N.A. | N.A. | N.A. | 14.940 | 0.815 | 1.400 | N.A. | N.A. | N.A. |
| BPCR96-1 | N.A. | N.A. | N.A. | 14.640 | 0.734 | 1.500 | N.A. | N.A. | N.A. |
| BpCr96-16 | N.A. | N.A. | N.A. | 16.640 | 1.164 | 1.500 | N.A. | N.A. | N.A. |
| BpCr96-17 | N.A. | N.A. | N.A. | 14.740 | 0.718 | 1.400 | N.A. | N.A. | N.A. |



| | | | | | | | | | |
|---|---|---|---|---|---|---|---|---|---|
| BpCr95-3(99) | N.A. | N.A. | N.A. | 15.340 | 0.818 | 1.400 | N.A. | N.A. | N.A. |
| BPCR96-10 | N.A. | N.A. | N.A. | 15.940 | 0.998 | 2.000 | N.A. | N.A. | N.A. |
| BPCR96-11 | N.A. | N.A. | N.A. | 14.040 | 0.715 | 1.400 | N.A. | N.A. | N.A. |
| BpCr95B-1(99) | N.A. | N.A. | N.A. | 16.240 | 0.877 | 1.400 | N.A. | N.A. | N.A. |
| *BpCR97-3* | *N.A.* | *N.A.* | *N.A.* | *17.640* | *0.973* | *1.400* | *N.A.* | *N.A.* | *N.A.* |
| BpCR97-4 | N.A. | N.A. | N.A. | 17.540 | 0.964 | 1.700 | N.A. | N.A. | N.A. |
| *BpCR97-2* | *N.A.* | *N.A.* | *N.A.* | *11.940* | *0.757* | *1.000* | *N.A.* | *N.A.* | *N.A.* |
| BpCR97-1a | N.A. | N.A. | N.A. | 16.440 | 0.906 | 1.300 | N.A. | N.A. | N.A. |
| BpCR97-1b | N.A. | N.A. | N.A. | 16.540 | 0.876 | 1.300 | N.A. | N.A. | N.A. |
| BpCr96-21 | N.A. | N.A. | N.A. | 14.940 | 0.765 | 1.100 | N.A. | N.A. | N.A. |
| BpCr96-22 | N.A. | N.A. | N.A. | 15.840 | 1.612 | 1.800 | N.A. | N.A. | N.A. |
| BpCr95-2(99) | N.A. | N.A. | N.A. | 15.540 | 0.784 | 1.200 | N.A. | N.A. | N.A. |
| ***Humphreys Basin (HB)*** | | | | | | | | | |
| HB97-1 | N.A. | N.A. | N.A. | 15.940 | 0.868 | 1.200 | N.A. | N.A. | N.A. |
| HB97-2 | N.A. | N.A. | N.A. | 17.340 | 0.849 | 1.300 | N.A. | N.A. | N.A. |
| HB97-3 | N.A. | N.A. | N.A. | 16.040 | 0.685 | 1.100 | N.A. | N.A. | N.A. |
| HB97-4 | N.A. | N.A. | N.A. | 15.340 | 0.727 | 1.400 | N.A. | N.A. | N.A. |
| HB97-5 | N.A. | N.A. | N.A. | 16.440 | 0.814 | 1.300 | N.A. | N.A. | N.A. |



**Table 2.** Mean surface-exposure age and standard deviation for each sampling location. Location abbreviations: TLC – Top of Lyell Canyon; MLC – Middle elevation within Lyell Canyon; BLC – Bottom of Lyell Canyon; TM-LC – an unnamed roche moutonnée near the junction of Lyell Canyon with Tuolumne Meadows; PD – Pothole Dome; LM – lower Tuolumne Meadows; LD – Lembert Dome; uppMC – Pioneer Basin and the upper reaches of Mono Creek Canyon; midMC – mid-valley location within Mono Creek Canyon; lowMC – the lower reaches of Mono Creek Canyon; LEMinner – the innermost moraine beneath Lake Edison; LEMouter – the outermost moraine beneath Lake Edison; T4m – the Tioga 4 moraine in the Middle Fork of Bishop Creek Canyon; SB&O – scattered boulders and outcrops between the Tioga 4 moraine and the Recess Peak moraines; and HB – Humphrey's Basin.

| Sampling location | $n_i$ | $n_f$ | Balco et al. (2008) v. 3.0.2 (with 3 Mar 2024 constants) | | | Marrero et al. (2016) CRONUScalc 2.0 | | | Martin et al. (2017) CREp (on 3 Mar 2024) | | |
|---|---|---|---|---|---|---|---|---|---|---|---|
| | | | Age (ka) | 1 σ internal (ka) | 1 σ external (ka) | Age (ka) | 1 σ internal (ka) | 1 σ external (ka) | Age (ka) | 1 σ internal (ka) | 1 σ external (ka) |
| *Tuolumne Meadows and Lyell Canyon* | | | | | | | | | | | |
| TLC | 3 | 3 | 14.954 | 0.421 | 1.572 | 15.807 | 0.493 | 1.303 | 15.778 | 0.483 | 1.663 |
| MLC | 1 | 0 | NA | NA | NA | NA | NA | NA | NA | NA | NA |
| BLC | 6 | 6 | 14.389 | 0.220 | 0.723 | 15.140 | 0.200 | 0.548 | 15.120 | 0.206 | 0.667 |
| TM-LC | 1 | 0 | NA | NA | NA | NA | NA | NA | NA | NA | NA |
| PD | 3 | 2 | 14.906 | 0.802 | 2.717 | 15.790 | 0.919 | 2.438 | 15.691 | 0.870 | 2.981 |
| LM | 1 | 1 | 15.207 | 0.239 | 0.936 | 16.040 | 0.434 | 1.300 | 15.996 | 0.230 | 0.860 |
| LD | 3 | 3 | 14.686 | 0.646 | 2.261 | 15.507 | 0.666 | 1.983 | 15.463 | 0.666 | 2.158 |
| *Lake Edison and Mono Creek Canyon* | | | | | | | | | | | |
| uppMC | 5 | 5 | 14.809 | 0.127 | 0.448 | 15.620 | 0.179 | 0.521 | 15.598 | 0.158 | 0.498 |
| midMC | 2 | 0 | NA | NA | NA | NA | NA | NA | NA | NA | NA |
| lowMC | 1 | 1 | 15.137 | 0.434 | 1.009 | 16.040 | 0.570 | 1.400 | 15.997 | 0.420 | 0.920 |
| LEMinner | 2 | 2 | 15.591 | 0.081 | 0.223 | 16.540 | 0.141 | 0.344 | 16.466 | 0.085 | 0.222 |
| LEMouter | 3 | 2 | 19.003 | 0.593 | 2.008 | 19.940 | 0.566 | 1.513 | 19.949 | 0.601 | 1.920 |
| *Bishop Creek Canyon and vicinity* | | | | | | | | | | | |
| T4m | 5 | 5 | NA | NA | NA | 16.280 | 0.907 | 1.646 | NA | NA | NA |
| SB&O | 16 | 14 | NA | NA | NA | 15.926 | 0.973 | 1.565 | NA | NA | NA |
| HB | 5 | 5 | NA | NA | NA | 16.220 | 0.740 | 1.190 | NA | NA | NA |





*4.2.     New $^{10}$Be dates from Lake Thomas A. Edison and Mono Creek Canyon*
We report thirteen new $^{10}$Be samples from Lake Edison and Mono Creek Canyon. After
rejecting three outliers (Sect. S.7.), the ten remaining samples yield ages ranging 19.4–14.7 ka
(Version 3 calculator), 20.3–15.4 ka (CRONUScalc 2.0), and 20.4–15.4 ka (CREp; Table 1).
The mean age of the outermost Lake Edison moraine varies by 0.9 kyr (5 %) across the
three calculators (Table 2): the Version 3 calculator dates stabilization of the outermost moraine
to 19.0 ± 0.6 ka (mean and standard deviation; n = 2; Fig. 4c) while CRONUScalc dates it to
19.9 ± 0.6 ka (Fig. 4d) and CREp dates it to 19.9 ± 0.7 ka (Fig. 4e).
Likewise, the innermost Lake Edison moraine's age also varies by ca. 1 kyr across the
calculators (Table 2): the Version 3 calculator suggests stabilization at 15.6 ± 0.1 ka (mean and
standard deviation; n = 2; Fig. 4c) while CRONUScalc and CREp place it at 16.5 ± 0.2 ka (Fig.
4d) and 16.5 ± 0.1 ka (Fig. 4e), respectively. Thus, despite calculator-based differences of 0.9
kyr (6 %) in the timing of this event, all three calculators indicate that the outermost Lake Edison
moraine predates the onset of rapid deglaciation in the Mono Creek drainage basin by ca. 3.4
kyr, with the Version 3 calculator producing a median estimate of 3.4 kyr (and with a 95 %
confidence range of 2.2–4.6 kyr), CRONUScalc producing a median estimate of 3.4 kyr (2.2–4.5
kyr), and CREp producing a median estimate of 3.5 kyr (2.3–4.7 kyr).
Following abandonment of the innermost Lake Edison moraine (i.e., the start of the Tioga
4 deglaciation), the ice-margin retreated up Mono Creek Canyon. A single sample from lower
Mono Creek Canyon (lowMC), 7 km behind the innermost Lake Edison moraine and 15 km
from the cirque headwall in Pioneer Basin, yields exposure ages of 15.1 ± 0.5 ka (1 σ internal;
Version 3; Fig. 4c), 16.0 ± 0.6 ka (CRONUScalc; Fig. 4d), and 16.0 ± 0.5 ka (CREp; Fig. 4e;
Table 2). The five samples from upper Mono Creek Canyon (1–5 km from the cirque headwall)





produce mean ages and standard deviations of 14.8 ± 0.2 ka (the Version 3 calculator; Fig. 4c),
15.6 ± 0.2 ka (CRONUScalc; Fig. 4d), and 15.6 ± 0.2 ka (CREp; Fig. 4e; Table 2). All three
calculators produce similar median estimates for the duration of the Tioga 4 deglaciation in
Mono Creek Canyon (Fig. 4i–k): 0.8 kyr (with a 95 % confidence interval of 0.5–1.1 kyr; the
Version 3 calculator), 1.0 kyr (0.5–1.4 kyr; CRONUScalc 2.0), and 0.9 kyr (0.6–1.2 kyr; CREp).





**Figure 4. (a)** Sampling locations in the Lake Edison and Mono Creek Canyon field area.
Sampling-location abbreviations (from distal to proximal): LEMouter – the outermost moraine of
the Lake Edison moraine complex; LEMinner – the innermost Lake Edison moraine; lowMC –



lower Mono Creek Canyon; midMC – middle Mono Creek Canyon; and uppMC – upper Mono
Creek Canyon. **(b)** 2014 aerial photo of Lake Edison at a low-water stage with moraine crests
traced in orange. **(c–e)** Beryllium-10 ages and 1 σ internal uncertainties from the Version 3
Calculator **(c)**, CRONUScalc 2.0 **(d)**, and CREp **(e)**. Gray boxes enclose each sample group's
mean age plus/minus one standard deviation. Rejected outliers shown with hollow symbols. Note
the extreme outlier (ca. 5 ka) from the outermost Lake Edison moraine (LEM-14-29) is outside
the displayed age range. **(f–h)** Beryllium-10 ages and 1 σ external uncertainties. The yellow
shading's intensity is proportional to the relative probability of deglaciation, per Phillip's (2016)
interpretation of basal radiocarbon dates. **(i–k)** Results of a 1 x $10^5$ trial Monte Carlo simulation
of deglaciation's duration in this drainage basin. We defined "valid" trials as those where the
simulated deglaciation ages run from oldest to youngest in the following order: the outermost
Lake Edison moraine, the innermost Lake Edison moraine, lower Mono Creek Canyon, and
upper Mono Creek Canyon.





*4.3.    Recalculated $^{36}Cl$ dates from the vicinity of Bishop Creek Canyon*

We report twenty-six recalculated $^{36}Cl$ ages from the vicinity of Bishop Creek Canyon.
The five boulders from the Tioga 4 moraine within the Middle Fork of Bishop Creek Canyon
produce a mean age of 16.3 ± 1.7 ka (1 σ external uncertainty), fourteen samples (we reject two
samples as outliers; Sect. S.7.) from boulders and bedrock outcrops scattered between the Tioga
4 and Recess Peak moraines (which we will henceforth refer to as the "valley-bottom samples"
for brevity) produce a mean age and standard deviation of 15.9 ± 1.6 ka, and the five boulders
from Humphreys Basin produce a weighted-mean age of 16.2 ± 1.2 ka (Fig. 5c; Table 2).

Using these mean ages and their associated 1 σ internal uncertainties as input into a
Monte Carlo model that requires Tioga 4 moraine abandonment (and stabilization) to precede
deglaciation of Humphreys Basin and the valley-bottom locations results in 61 % of the Monte
Carlo trials being retained and produces a median estimate for the duration of the Tioga 4
deglaciation within Bishop Creek Canyon of 1.3 kyr with a 95 % confidence range of 0.1–3.3
kyr (Fig. 5d).





**Figure 5.** Phillips et al. (2009) Tioga 4 sampling locations in Bishop Creek Canyon and its
vicinity, along with the recalculated surface-exposure ages and inferred deglaciation durations.



**(a)** Phillips et al. (2009) Tioga-4 sampling locations (purple circles). Abbreviations: Ti1 – the
Tioga 1 moraine; Ti2/3 – the Tioga 2/3 moraine; Ti4 – the Tioga 4 moraine; HB – Humphrey's
Basin; BCC – Bishop Creek Canyon; NF BCC – the North Fork of Bishop Creek Canyon; MF
BCC – the Middle Fork of Bishop Creek Canyon; SF BCC – the South Fork of Bishop Creek
Canyon. Solid black lines are moraines; the dashed black line is a likely (Tioga 4) ice-marginal
position (Phillips et al., 2009). **(b)** Chlorine-36 surface-exposure ages and 1 σ internal age
uncertainties. Circles are boulder samples; squares are bedrock samples. Solid symbols are
retained samples; hollow symbols are rejected outliers. The gray boxes are the mean age of the
retained $^{36}$Cl ages plus/minus one standard deviation. **(c)** Chlorine-36 ages and 1 σ external age
uncertainties. The hatched boxes enclose the mean age of each sampling group, plus/minus one
standard deviation. The yellow shading's intensity is proportional to deglaciation's relative
probability, according to Phillip's (2016) interpretation. **(d)** The distribution of Tioga 4
deglaciation durations in the vicinity of Bishop Creek Canyon, based on a 1 x 10$^5$ trial Monte
Carlo simulation. We define "valid" trials as those where the Tioga 4 moraine has the oldest
simulated deglaciation age and deglaciation's duration as the time difference between the Tioga
4 moraine's stabilization and the youngest deglaciation age from the other two sampling groups.



*4.4.     Updated chronologies for previously published datasets*

Revising the age-depth model for the Swamp Lake paleoenvironmental reconstruction

from one based on linear interpolation between adjacent radiocarbon dates to one that permits
sedimentation rates to vary throughout the core suggests that age uncertainties on the core's
undated horizons are ~2–3 times larger than the age uncertainties on the calibrated radiocarbon
dates. Although these age uncertainties vary with core-depth distance to the nearest radiocarbon
date, representative 1 σ uncertainties for the undated horizons in the Swamp Lake core of
greatest relevance to the Tioga 4 deglaciation are ca. 0.40 kyr verses ca. 0.15 kyr for the
radiocarbon dates themselves (Fig. S23; Table S7). Note that – despite updating Swamp Lake's
age-depth model from IntCal04 (Reimer et al., 2004) to IntCal20 (Reimer et al., 2020) – the
median age estimates for the radiocarbon dates in Swamp Lake's age-depth model are only
slightly changed (≤0.15 kyr) from the values reported by Street et al. (2012); the advantage of the
new age-depth model that we report here for this core is in the better representing the true dating
uncertainties.

With regards to the radiocarbon constraints on the timing of Great Basin lake highstands,

updating the various radiocarbon calibrations previously used [i.e., IntCal09 (Reimer et al.,
2009) and IntCal13 (Reimer et al., 2013)] to IntCal20 (Reimer et al., 2020) results in new
calibrated ages ranging from 0.4 kyr younger to 0.3 kyr older than the ages reported by Munroe
and Laabs (2013) and Ibarra et al. (2014), with a mean adjustment of 0.1 kyr younger (Fig. S24;
Table S8).

Finally, for the paleoenvironmental reconstruction from McLean's Cave, the median age

estimate in Bchron's age-depth model for the various measurement depths in speleothem ML1
ranges from 0.147 kyr younger than the ages reported by Oster et al. (2015) to 0.731 kyr older,



with a mean difference of essentially zero (0.047 kyr older; Figs. S25–S26; Table S9). For the
portion of the speleothem with the greatest relevance to this manuscript [sampling depths
132.00–141.85 mm, with $^{230}$Th/U dates of 15.433 ± 0.065 ka (1 σ uncertainty) and 16.718 ±
0.159 ka at those depths, respectively (Oster et al., 2015)], the Bchron age-depth model has
median ages that are 0.110–0.510 kyr older than the ages reported by Oster et al. (2015), with a
mean difference of 0.325 kyr. Bchron's 1 σ age uncertainties for the undated horizons over this
sampling-depth interval range from ±0.066 kyr (at the interval's younger, shallower end) to
±0.156 kyr (at the interval's older, deeper end).
**5. Discussion**

This manuscript's goal is to place the Sierra Nevada's final deglaciation in a robust

regional and global context. That narrative, however, depends upon the cosmogenic dates and the
three calculators produce exposure ages that disagree by ~5 % (ca. 1 kyr) despite identical inputs.
Thus, this discussion is structured as follows. First, we compare the calculated ages for each
calculator and document their differences (and similarities). Second, we use the radiocarbon
constraints on the timing of the Tioga 4 deglaciation to assess the accuracy of the calculators for
these particular $^{10}$Be and $^{36}$Cl samples. Then, after adopting a preferred set of exposure ages, we
return to this manuscript's primary focus and discuss the final deglaciation of the Sierra Nevada
within a regional and global context.
*5.1.    Comparison of the cosmogenic surface-exposure-age calculators*

For the thirty-one new $^{10}$Be samples presented here, CRONUScalc 2.0 and CREp (on 3

Mar 2024) generated surface-exposure ages that were ~5 % older than those produced by the
Version 3 calculator (Fig. 6). In particular, the CRONUScalc ages averaged 5.4 % older (range:
3.2–6.3 %) and the CREp ages averaged 5.3 % older (range: 4.7–5.7 %). These differences





exceed the relative uncertainty in the $^{10}$Be concentrations of the samples, which average 1.8 %,
and range 1.3–4.4 % (Table S3). Additionally, the age differences between the CRONUScalc 2.0
and the Version 3 calculator results are larger than their combined internal age uncertainties for
twenty-six of the thirty-one $^{10}$Be samples (84 %); likewise, the age differences between the
results from CREp and the Version 3 calculator are larger than their combined internal age
uncertainties for thirty of the samples (97 %). Thus, by these measures, the age differences
returned by the calculators are significant.

This ~5 % age difference cannot be explained by the scaling or atmospheric models, as

all three calculators used LSDn scaling (Lifton et al., 2014) and the ERA-40 atmospheric
reanalysis (Uppala et al., 2005), and the ~5 % age difference also cannot be explained by the
calculators using different geomagnetic-field reconstructions and $^{10}$Be-production-rate-
calibration datasets. Although the geomagnetic-field reconstruction is not constant across the
calculators, two lines of evidence suggest this variability is not the source of the ~5 % age
difference: First, the ages from the Version 3 calculator and CREp were both calculated with the
Lifton (2016) geomagnetic reconstruction, and yet the two sets of ages differ by ~5 %. Second,
recalculating the exposure ages of these thirty-one samples in CREp (the only calculator of the
three that readily enables users to select their preferred geomagnetic reconstruction) with the
Lifton et al. (2014) reconstruction (which is used by CRONUScalc) changes the CREp exposure
ages by <1 % on average (Table S10). Both the Version 3 calculator and CRONUScalc 2.0 used
the CRONUS-Earth primary $^{10}$Be-calibration dataset, so the dataset used to calibrate the $^{10}$Be
production rate also cannot be the reason for the ~5 % age difference across the calculators.
Lastly, we note that the corrections for topographic and snow shielding are not particularly large
(ranging from 1−14 %), nor are the erosion rates we infer for these samples ($\leq 1$ mm yr$^{-1}$) – and,



in any event, these corrections are a constant for any particular sample across the three
calculators. Thus, the ~5 % age difference between the calculators cannot be explained by
scaling model, the atmospheric and magnetic reconstructions, the choice of calibration dataset, or
particulars about the samples (i.e., topographic shielding, snow shielding, or erosion rate).

We hypothesize that the ~5 % age difference is the result of the calculators using

different algorithms for converting a suite of calibration samples into a single value (be it a non-
dimensional LSDn scaling factor or a SLHL production rate) for scaling to other locations.
Unfortunately, assessing this hypothesis is hindered by the Version 3 calculator reporting a
LSDn scaling factor (and not also a SLHL $^{10}$Be production rate) and CRONUScalc 2.0 and
CREp reporting SLHL production rates (and not also LSDn scaling factors). Fortunately,
determining why the calculators are returning different surface-exposure ages is not essential to
this paper's purpose. Instead, we focus on two tasks: (1) assessing which calculator is more
likely producing more accurate ages for the samples in the dataset reported here; and then (2)
placing these dates within the context of previous paleoclimate research.

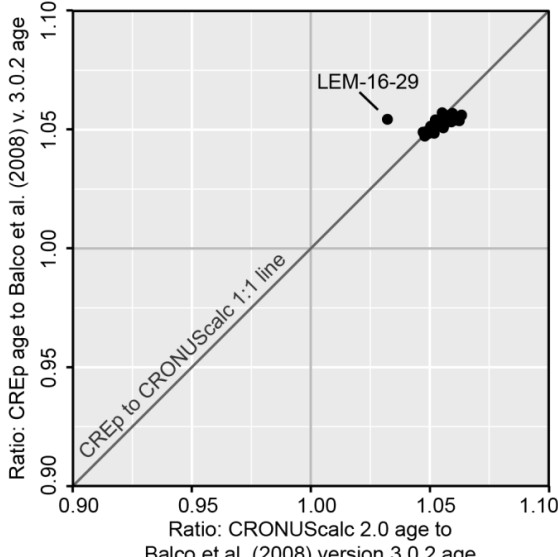


**Figure 6.** Comparison of the [10]Be ages from the three calculators. The Version 3 calculator
produces exposure dates that are ~5% younger than the ages produced by CRONUScalc 2.0 and
CREp for the thirty-one [10]Be samples in this dataset. The outlier is the ca. 5 ka boulder from
Lake Edison, which is ca. 9–19 kyr younger than the other samples.

*5.2    Assessment of the cosmogenic surface-exposure-age calculators*

If we accept the interpretation that Tuolumne Meadows deglaciated before 15.4 ka (as

suggested by the Greenstone Lake radiocarbon date; Fig. S2; Clark et al., 2003) and that the

high-elevation lake basins of the Sierra Nevada deglaciated at 15.75 ± 0.5 ka (Phillips, 2016), as

suggested by the bulk-organic radiocarbon dates (Sect. 2.2.2), then the probability distributions

on the cosmogenic surface-exposure dates suggest that the CRONUScalc 2.0 and CREp dates are

probably more accurate than the dates from the Version 3 calculator for the limited range of

elevations (2300–3500 m), latitudes (37–38° N), longitudes (118–120° W), and [10]Be

concentrations (3–6 x $10^5$ atoms g$^{-1}$) represented by the thirty-one [10]Be samples in this

manuscript's dataset (Figs. 3e–g and 4f–h). We emphasize this assessment is (1) probabilistic

and (2) predicated on the accuracy of the bulk-sediment radiocarbon dates. Because bulk-





sediment radiocarbon dates can be biased toward older ages by the presence of inorganic "dead"
carbon, we first assess the possibility that these bulk-sediment radiocarbon dates from the Sierra
Nevada are contaminated with radioactively dead carbon (Sect. 5.2.1.) before returning to the
topic of the surface-exposure calculators and their relative accuracy for the Tioga 4 deglaciation
in the Sierra Nevada (Sect. 5.2.2.).
*5.2.1    On the accuracy of the bulk-organic radiocarbon dates*

Bulk-organic radiocarbon dates are problematic for accurate chronologies because trace

amounts of inorganic carbon, if present in the sediments, will bias the radiocarbon dates toward
older ages due to the "hardwater effect" (Godwin, 1951, p. 306; Deevey et al., 1954; Oana and
Deevey, 1960; Philippsen, 2013). However, three lines of evidence suggest dead carbon is not a
significant concern for these samples. First, carbonate bedrock is absent within the radiocarbon-
dated drainage basins; instead, these watersheds are etched into the granitic rocks of the Sierra
Nevada (Bateman and Wones, 1972; Bateman et al., 1983; Huber, 1983).

Second, Clark and Gillespie (1997) tested for dead carbon in four other granitic

watersheds of the Sierra Nevada by dating macrofossils (twigs, branches, and charcoal) and
directly adjacent bulk sediment (organic silt, gyttja, and peat) in four cores. In four of five cases,
these paired radiocarbon dates agree within 1 σ. In the case of the exception, the bulk sediment
(gyttja) is ca. 1.1 kyr older than the macrofossil, which was a piece from a "large" branch. While
this age offset could be explained by 10 % of the carbon in the gyttja being dead with respect to
$^{14}$C activity, Clark and Gillespie (1997) rejected this interpretation because – crucially – one of
the four other radiocarbon-dated pairs in their dataset came from the same sediment core (BL-2,
from Upper Baboon Lake in the Bishop Creek drainage basin; Fig. 1f) and the macrofossil and
bulk-sediment in this other radiocarbon-date pair had indistinguishable ages: the macrofossil





(twig) dated to 7,900 ± 60 $^{14}$C-yrs and the bulk sediment (gyttja) dated to 7,890 ± 60 $^{14}$C-yrs.
Therefore, Clark and Gillespie (1997) interpreted the mismatched radiocarbon-date pair as
resulting from the large branch sinking into then-flocculent gyttja when it landed in the pond – a
phenomenon observable within modern ponds and interpreted for other lacustrine cores (e.g.,
Davis et al., 2019, Fig. 16) – and we accept their interpretation for the observed age offset in this
pair of radiocarbon dates.

Finally, Swamp Lake also occupies a granitic watershed within the Sierra Nevada and all

thirteen radiocarbon dates in its age-depth model were on bulk sediment; Street et al. (2012)
tested the core's sediments for carbonates by fumigating samples with concentrated HCl and
comparing them with unfumigated samples. Street et al. (2012) found no detectable carbonates
within the Swamp Lake core.

Although a lack of compelling evidence for dead carbon within the lacustrine sediments

of five granitic watersheds in the Sierra Nevada [i.e., Swamp Lake plus the four investigated by
Clark and Gillespie (1997)] does not preclude its presence in the sediments of other granitic
watersheds in the Sierra Nevada, there is no evidence for dead carbon in the lacustrine sediments
of the Sierra Nevada's granitic watersheds. Therefore, we accept the bulk-organic radiocarbon
dates reviewed in Sect. 2.2.2 as accurate minimum limits on the timing of the Tioga 4
deglaciation.
*5.2.2. On the accuracy of the cosmogenic surface-exposure-age calculators*

Accepting the bulk-organic radiocarbon dates as accurate minimum limits on the timing

of the Tioga 4 deglaciation suggests that the CRONUScalc and CREp dates are probably more
accurate than the Version 3 calculator's dates for the $^{10}$Be samples reported here. Our argument
is as follows: After excluding outliers, nine sampling locations remain (Table 2). For six of these





nine locations (lower Tuolumne Meadows, Pothole Dome, Lembert Dome, the top of Lyell
Canyon, lower Mono Creek Canyon, and upper Mono Creek Canyon), the CRONUScalc and
CREp dates produce mean sampling group ages closely bracketing 15.75 ka (≤0.3 kyr). These
mean ages range from 16.0–15.5 ka (Table 2). For the three remaining locations, larger
deviations from 15.75 ka seem reasonable. These three exceptions are (from oldest to youngest):
(1) the outermost moraine beneath Lake Edison (which predates the innermost Lake Edison
moraine by ca. 3.4 kyr across all three calculators; Fig. 4c–e); (2) the innermost moraine beneath
Lake Edison, which records the start of the Tioga 4 deglaciation, and not the deglaciation of
high-elevation lake basins, as the radiocarbon dates do; and (3) the bottom of Lyell Canyon,
which has comparatively young exposure ages across all three calculators and likely hosted a
substantially greater Tioga 4 ice thickness than the radiocarbon-dated lake basins (Wahrhaftig et
al., 2019).

In contrast to CRONUScalc and CREp, the Version 3 calculator produces mean ages that

are 0.5–1.1 kyr younger than 15.75 ka for the six locations mentioned above (Figs. 3e and 4f).
These mean ages range from 15.2–14.7 ka (Table 2). Likewise, within this set of results, the
innermost Lake Edison moraine likely stabilized ca. 0.1–0.5 kyr after the start of organic
accumulation in the high basins of the Sierra Nevada. Additionally, the Version 3 calculator's
exposure ages from locations analogous to the radiocarbon-dated lake basins [such as the top of
Lyell Canyon (with a mean age of 15.0 ka) and the upper reaches of Mono Creek Canyon (14.8
ka)] postdate the high-elevation lake-basin radiocarbon dates by ca. 0.9 kyr (Figs. 3e and 4f).
Shifting the bulk-organic radiocarbon ages 0.9 kyr younger, into agreement with the most
probably ages from the Version 3 calculator, would require 7–11 % of the carbon in the
sediments of Highlands, Greenstone, and East Lakes be "dead" with respect to $^{14}$C activity.





761    Considering Tuolumne Meadows in particular, where the radiocarbon-dated deglacial

762 chronology is the strongest of all our field areas, CRONUScalc and CREp produce mean

763 deglaciation ages of 16.0–15.5 ka for lower Tuolumne Meadow, Pothole Dome, and Lembert

764 Dome – while the Version 3 calculator produces mean ages of 15.2–14.7 ka (Table 2). In

765 comparison, Greenstone Lake was both deglaciated and accumulating organic carbon by 15.4 ka

766 (Fig. S2; Sect. 2.2.2) – and Greenstone Lake was ≤4 km from the accumulation zone that fed ice

767 into Tuolumne Meadows (Huber, 2007; Wahrhaftig et al., 2019). If Greenstone Lake was

768 deglaciated and accumulating organic carbon by 15.4 ka, then it seems likely that our sampling

769 sites in lower Tuolumne Meadows (~30 m above the modern floor of Tuolumne Meadows) and

770 on Pothole Dome (~70 m) and Lembert Dome (~230 m) were also deglaciated by that time.

771    We tested whether recalculating the $^{10}$Be surface-exposure ages with the most precisely

772 determined $^{10}$Be production rate from western North America (the Promontory Point production

773 rate; Lifton et al., 2015) would shift the ages older – and thus bring the results of the Version 3

774 calculator into better agreement with the radiocarbon dates from the Sierra Nevada (Fig. S2). We

775 instead find that the Lifton et al. (2015) calibration dataset results in even younger exposure ages

776 (Table S11). Therefore, with the evidence available to us, we conclude that the Version 3

777 calculator is probably producing exposure ages that are ~5 % too young for the thirty-one $^{10}$Be

778 samples in this location and that the CRONUScalc and CREp dates are more consistent with the

779 current radiocarbon chronology.

780    Accordingly, with the goal of streamlining the Discussion going forward, we will

781 henceforth exclusively refer to the CRONUScalc 2.0 ages, which are indistinguishable from the

782 CREp ages. This interpretive decision results in the following deglacial narrative: the innermost

783 Lake Edison moraine stabilized a few hundred years prior to the start of organic accumulation in





the high-elevation ponds of the Sierra Nevada (Fig. 4g), the six sampling locations mentioned
above were then exposed to cosmic rays around the same time that the ponds began
accumulating organic carbon, and deglaciation concluded with the melting of the valley-bottom
ice remnants, where Tioga 4 ice thicknesses were the greatest (e.g., the bottom of Lyell Canyon;
Figs. 3f). This narrative is consistent with geomorphic interpretations of rapid deglaciation and
ice thinning in the Sierra Nevada at the end of the last glaciation (Clark, 1976; Clark and Clark,
1995) and with observations from Alaska (Cushing, 1891; Reid, 1896; Field, 1947; Mickelson,
1971; Syverson, 1995), and interpretations (Goldthwait, 1938; Lowell, 1985; Lowell et al., 1990)
and dating (Bierman et al., 2015; Davis et al., 2015; Koester et al., 2017, 2021; Corbett et al.,
2019; Drebber et al., 2023; Halsted et al., 2024) from New England and adjacent regions (Barth
et al., 2019) for how rapid deglaciation proceeds in mountainous landscapes (Goldthwait and
Mickelson, 1982).
*5.3    New $^{10}$Be and revised $^{36}$Cl constraints on the timing and duration of the Tioga 4*

*deglaciation*

*5.3.1.   On the timing of the Tioga 4 deglaciation*

Within this context, the best direct estimate for the culmination of the Tioga 4 readvance

and the start of the Tioga 4 deglaciation is 16.4 ± 0.8 ka (1 σ external; Fig. 7a). This age is the
mean and standard deviation of the seven cosmogenic surface-exposure ages from the two Tioga
4 moraines dated by this study: the two $^{10}$Be dates from the innermost Lake Edison moraine
(which independently dates to 16.5 ± 0.4 ka; Fig. 4g; Table 2) and the five $^{36}$Cl dates from the
Tioga 4 moraine in Bishop Creek Canyon (which independently dates to 16.2 ± 1.7 ka; Fig. 5c).
Although the most distal sampling location in Tuolumne Meadows does not date the start of the




Tioga 4 deglaciation, we note that it also produces an exposure age (16.0 ± 1.3 ka) similar to the
stabilization ages of these two innermost moraines.

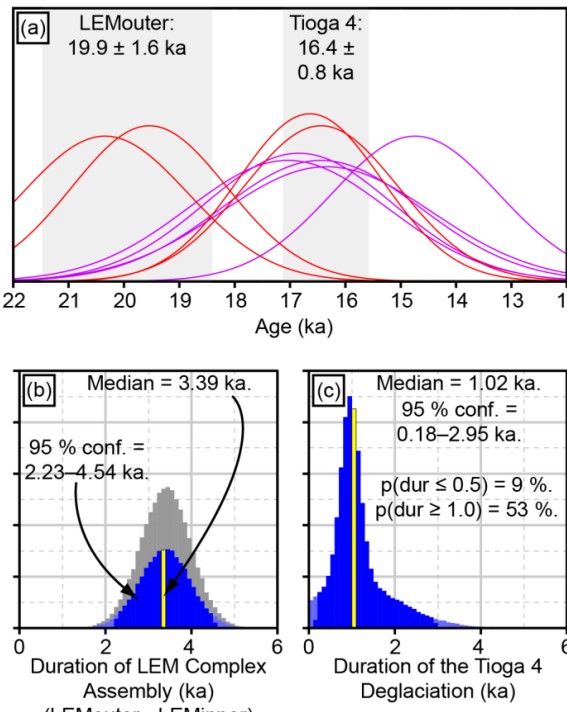

**Figure 7. (a)** The 68 % confidence (1 σ) ranges for the outermost Lake Edison moraine
(LEMouter) and for the start of the Tioga 4 deglaciation (which is dated by two [10]Be and five
[36]Cl dates across two innermost moraines). Red lines are the probability densities of the
individual [10]Be dates (as calculated by CRONUScalc 2.0) and the purple lines are the same for
the [36]Cl samples. Samples constraining the end of the Tioga 4 deglaciation (rather than its
beginning) are omitted for clarity. **(b)** The inferred duration of moraine construction in the Lake
Edison moraine complex. **(c)** Duration of the Tioga 4 deglaciation in 1 x $10^5$ Monte Carlo
simulations (5 x $10^4$ from Mono Creek Canyon and 5 x $10^4$ from Bishop Creek Canyon).

Prior to the start of the Tioga 4 deglaciation, regional climate experienced a few thousand

years that were at least episodically conducive to moderate glaciation in the Sierra Nevada. This
interpretation follows from two observations: first, the 3.4 kyr difference in the mean ages of the
outermost and innermost Lake Edison moraines (with a 95% confidence interval of 2.2–4.5 kyr;
Fig. 7b); second, the presence of approximately nine additional (undated) end moraines between





these two moraines (Fig. 4b). Although these numerous, closely spaced moraines (~11 in 2 km)
are suggestive of glacial advances driven by interannual climate variability (e.g., Anderson et al.,
2014; Barth et al., 2017), where stochastic variations in temperature and precipitation are
superimposed on a climate that is approximately stable over a longer time, the presence of the
Lake Edison moraine complex does not require this interpretation. The moraines might truly
represent large-magnitude and long-term variations in summer temperature and winter
precipitation in the Sierra Nevada over the ca. 20–16 ka interval. However – regardless of
whether the moraines in the Lake Edison complex represent interannual climate variability (i.e.,
weather), climate change, or some combination thereof – they indicate the former Mono Creek
Glacier's ELA dropped to ~2800 m (Phillips, 2016) between ca. 20 ka and ca. 16.5 ka before
returning to higher elevations.
*5.3.2.  On the duration of the Tioga 4 deglaciation*

The glaciers of the Sierra Nevada rapidly melted after the start of the Tioga 4

deglaciation at 16.4 ± 0.8 ka; our best estimate for the full duration of the Tioga 4 deglaciation in
the central Sierra Nevada from innermost moraine stabilization to the melting of the last
remanent ice masses is 1.0 kyr, with a 95 % confidence range of 0.2–3.0 kyr (Fig. 7c).

Although the original analysis of the [36]Cl dates from the vicinity of Bishop Creek Canyon

concluded that the Tioga 4 deglaciation was probably ≤500 years and certainly <1000 years
(Phillips et al., 2009) – we find a ~50 % probability that the full duration of the Tioga 4
deglaciation in the central Sierra Nevada was 1000 years or longer and only a ~10 % probability
that it was 500 years or less (Fig. 7c). However, despite finding that the time required for
deglaciation was ca. 500 years longer than interpreted by Phillips et al. (2009), the [10]Be dates
from Lake Edison and Mono Creek Canyon (Fig. 4) support the fundamental conclusion of



Phillips et al. (2009) that the deglacial history of Bishop Creek Canyon and its vicinity (Fig. 5) is
representative of other locations within the Sierra Nevada.

Moreover, the new [10]Be dates are consistent with the interpretations of multiple

researchers (Clark, 1976; Clark and Clark, 1995; Phillips et al., 2009) that the Sierra Nevada's
final deglaciation was a geomorphically abrupt event. For instance, the bottom of Lyell Canyon
sampling location has the youngest mean surface-exposure age across all the sampling locations
reported here (Fig. 3c; Table 2) – and a trimline-based ice-thickness reconstruction (Wahrhaftig
et al., 2019) indicates the bottom of Lyell Canyon hosted an ice thickness greater than any other
location we sampled. Tioga 4 ice thicknesses over the bottom of Lyell Canyon sampling site
(~600 m) were ~2‑4x greater than the ice thicknesses over the sampling sites in Tuolumne
Meadows (~300 m) and at the top of Lyell Canyon (~150 m). Additionally, ice thicknesses in
Lyell Canyon were also ~50 % greater than ice thicknesses over Greenstone Lake (~400 m), thus
explaining why the bottom of Lyell Canyon almost certainly (99.4 % probability, per the
radiocarbon date's probability-density distribution from Bchron (Haslett and Parnell, 2008) and
our Monte Carlo modeling of the surface-exposure ages) deglaciated after the beginning of
organic accumulation in Greenstone Lake. The observation that the locations with the thickest
ice were the last to deglaciate, regardless of their distance from the cirque headwalls, indicates
that the deglaciation was driven by a relatively sudden and large rise in the ELA.
*5.4.*    *Geomorphic and climatological constraints on the magnitude of deglacial climate change*

Geomorphic and modern climatological observations suggest that this widespread and

rapid deglaciation (the Tioga 4 deglaciation) was driven by a summertime warming of ≥2 °C, a
wintertime drying of ≥35 %, or some combination thereof. The temperature-change
reconstruction is based on the interpretation that the deglaciation was driven by ≥600 m rise in





the ELA (as reviewed in Sect. 2.2.1.) and the observation that modern surface-temperature lapse
rates are ~3–4 °C km$^{-1}$ in the higher elevations of the Sierra Nevada and the Cascade Range
(Wolfe, 1992). Importantly, the similar lapse rates in both mountain ranges indicate modern
high-elevation lapse rates are mostly independent of latitude and we assume a similar
independence for the deglaciation period (Ye et al., 2023). Combining the minimum ELA change
with the estimated lapse rate suggests the Sierra Nevada's final deglaciation was driven by a
summertime warming of ≥2 °C, assuming no change in winter precipitation (Leonard et al.,

2023).

Conversely, if we assume no change in summer temperatures and accept a 20 %

precipitation decrease as the mass-balance equivalent of a 1 °C warming (Oerlemans, 2005), then
the Tioga 4 deglaciation could have alternatively been driven by a 35 % (or greater) reduction in
winter precipitation – with intermediate temperature and precipitation combinations possibly,
and perhaps likely.
*5.5.    Constraints from regional paleoclimate proxies on the timing and nature of the climate*

*change that drove the Tioga 4 deglaciation*

We examine five paleoclimate records from western North America and adjacent

portions of the Pacific Ocean that further illuminate the timing and drivers of the Tioga 4
deglaciation: (1) the lacustrine core from Swamp Lake (~40 km west of Tuolumne Meadows;
Fig. 1b; Street et al., 2012), (2) speleothem ML1 from McLean's Cave (~100 km west-northwest
of Tuolumne Meadows; Fig. 1b; Oster et al., 2015), (3) twenty-five radiocarbon dates from
seventeen Great Basin Pleistocene lakes (Fig. 1a; Munroe and Laabs, 2013; Ibarra et al., 2014),
(4) ODP core 893A from the Santa Barbara Basin (Fig. 1a; Hendy et al., 2002), and (5) the
MD02-2515 sea-surface temperature (SST) record from the Gulf of California (Fig. 1a;



McClymont et al., 2012). We compare these records with the $\delta^{18}$O record from Ostolo Cave in
Spain, which records the timing of Heinrich Event 1 (Bernal-Wormull et al., 2021; Pérez-Mejías
et al., 2021).
*5.5.1. Swamp Lake*

The paleoenvironmental proxy records from Swamp Lake (Fig. 1a) suggest the central

Sierra Nevada warmed at 16.4 ± 0.4 ka and that this warmth persisted for ca. 1.3–1.6 kyr. The
magnetic susceptibility of Swamp Lake's sediments decreased more than thirty-fold over a ca.
0.1 kyr interval at 16.4 ± 0.4 ka and remained at near-zero values for ca. 1.6 kyr (Fig. 8a).
Simultaneously, the core's total organic carbon (TOC) content started trending higher at 16.5 ±
0.4 ka and increased twenty-fold over ca. 1.3 kyr. Similar trends are recorded in the other proxy
records from Swamp Lake, such as: the tripling of the molar ratio of carbon to nitrogen, a nine-
fold increase in biogenic silica concentration, and a local $\partial^{13}$C maxima (Street et al., 2012).
These changes reflect the return of sub-alpine conifers and associated terrestrial vegetation in
Swamp Lake's small (1.3 km$^2$) drainage basin, which stabilized soils and allowed a near-order-
of-magnitude increase in diatom abundance via decreased turbidity (Street et al., 2012). Upward
migration of the upper treeline is usually driven by warming temperatures (e.g., LaMarche and
Mooney, 1967; Denton and Karlén, 1977; Kullman, 1986, 1995; Scuderi, 1987). We infer that
the Sierra Nevada warmed at 16.4 ± 0.4 ka and that these warmer temperatures lasted for ca. 1.3–
1.6 kyr.





**Figure 8.** The $^{10}$Be- and $^{36}$Cl-dated Tioga 4 deglaciation (16.4 ± 0.8 ka) in the context of other
paleoclimate records. **(a)** The total organic carbon (TOC) and magnetic susceptibility records
from Swamp Lake (~40 km west of Tuolumne Meadows; Street et al., 2012) on the revised age-



depth model that we present here (Sect. S.8.; Fig. S23; Table S7). **(b)** The δ¹³C and δ¹⁸O records from McLean's Cave (~100 km west-northwest of Tuolumne Meadows; Oster et al., 2015) on the revised age-depth model that we present here (Sect. S.10.; Fig. S25; Table S9). **(c)** The δ¹⁸O record from speleothem OST2 (Ostolo Cave, Spain), which constrains the start of Heinrich Event 1 to 16.22 ± 0.04 ka and its end to 16.04 ± 0.04 ka (Bernal-Wormull et al., 2021). **(d)** The timings of Great Basin lake highstands (Munroe and Laabs, 2013; Ibarra et al., 2014), as recalculated here (Table S8). **(d)** The *G. bulloides* δ¹⁸O and *N. pachyderma* δ¹⁸O records from ODP core 893A from the Santa Barbara Basin (Hendy et al., 2002); note that lower δ¹⁸O values record warmer temperatures. **(e)** The U$^{K'}_{37}$-index reconstruction of sea-surface temperatures from core MD02-2514 in the Gulf of California; the precision of these temperature estimates is ±0.6 °C (McClymont et al., 2012).

### 5.5.2. *McLean's Cave*

The δ¹³C and δ¹⁸O records from McLean's Cave (Fig. 8b; Oster et al., 2015) suggest the west-central Sierra Nevada warmed and dried between 16.20 ± 0.13 ka and 14.71 ± 0.19 ka. Interestingly, these dates are indistinguishable from the start of Heinrich Event 1 in the North Atlantic (16.22 ± 0.04 ka; Fig. 8c; Bernal-Wormull et al., 2021; Pérez-Mejías et al., 2021) and the start of the Bølling (14.74 ± 0.06 ka; Lemieux-Dudon et al., 2010). The δ¹³C record trends higher over that 1.5 kyr interval, driven by increased prior calcite precipitation (PCP) and/or decreased soil respiration, both of which record a drying climate (Oster et al., 2015). The δ¹⁸O record reaches a local minimum nearly contemporaneously with the just mentioned δ¹³C minimum (the δ¹⁸O minimum is one isotopic-measurement-interval earlier in the core, at 16.23 ± 0.13 ka) and rises, episodically, for ca. 0.8 ka. The 2.17 ‰ δ¹⁸O increase over that interval might reflect warmer temperatures in the Sierra Nevada (up to 3–4 °C; Rozanski et al., 1993), a ~40% reduction in moisture delivery to the range from the North Pacific (relative to moisture sourced from the subtropics), a change in the δ¹⁸O of the moisture-source regions, or some combination thereof. Thus, the δ¹³C record implies drying of the west-central Sierra Nevada starting at 16.20 ± 0.13 ka and the δ¹⁸O record permits both it and warming.

### 5.5.3. *Great Basin lakes*



Considering the western United States more broadly, the temporal and spatial pattern of
Great Basin lake-level highstands (Fig. 8d) and numerical modeling suggests that these
highstands were fed by an invigorated subtropical jet stream and a deeper and southeastward-
shifted Aleutian Low, which collectively increased moisture transport into the Great Basin via
atmospheric rivers (McClymont et al., 2012; Chiang et al., 2014; McGee et al., 2018). Notably,
the southern limit of lake-level highstands shows a ~6° northward jump between ca. 16.4 ka and
ca. 16.2 ka, suggesting a major reorganization in atmospheric circulation over western North
America at that time (Fig. 8d), coincident with the deglaciation of the Sierra Nevada. In
particular, the ~6° northward jump is defined by (1) Lake Cochise (at 32° N; "LC" on Figs. 1a
and 8d) abandoning its 1274 m shoreline at 16.37 ± 0.21 ka (Waters, 1989; Munroe and Laabs,
2013), (2) Lake Estancia (at 35° N; "LE" on Figs. 1a and 8d) abandoning its 1890 m shoreline at
16.16 ± 0.15 (Allen and Anderson, 2000; Munroe and Laabs, 2013), and by (3) the abrupt rise in
$\delta^{13}$C in McLean's Cave (at 38° N; Oster et al., 2015) at 16.20 ± 0.13 ka. Note that the highstands
of meltwater-fed Owens Lake (at 36° N) and Lake Russell (at 38° N) post-date this atmospheric
reorganization due to the release of meltwater from the rapidly melting glaciers of the Sierra
Nevada.
*5.5.4.  The Santa Barbara Basin and the Gulf of California*
The interpretation that rapid deglaciation began in the Sierra Nevada at ca. 16.2 ka is also
consistent with marine-sediment cores from the Santa Barbara Basin and the Gulf of California.
Although the Santa Barbara Basin core (ODP core 893A; Hendy et al., 2002) unfortunately lacks
data between ca. 16.6 ka and ca. 15.7 ka due to a core-recovery failure, the average $\delta^{18}$O value of
*G. bulloides* (a surface-dwelling planktonic foraminifera) decreased approximately 1.4 ‰ over
this missing interval (Fig. 8e), which is consistent with warming. Changes in the core's





967 planktonic-foraminifera assemblage suggests Santa Barbara Basin SSTs warmed ~5 °C over this

968 missing interval. This large change in SSTs suggests that the Aleutian Low weakened and

969 migrated to the northwest, causing the California Current to weaken (Hendy, 2010), slowing the

970 flow of subpolar water into the basin (Hendy and Kennett, 2000; Hendy et al., 2002).

971   Similarly, the $U^{K'}_{37}$ reconstruction from Gulf of California core MD02-2515 (McClymont

972 et al., 2012) indicates SSTs there warmed $4.0 \pm 0.9$ °C between 16.8 ka and 16.0 ka (Fig. 8f). A

973 parallel $TEX^{H}_{86}$ reconstruction from the core demonstrates little-to-no warming over this time

974 interval at the thermocline. The divergence in these temperature proxies implies a sharp

975 reduction in upwelling, which in turn suggests a reduction in northwesterly winds over the Gulf

976 of California (Ganeshram and Pedersen, 1998), such as might accompany a northwestward

977 migration of the northwest Pacific's subtropical high (McGee et al., 2018). Such a reorganization

978 in atmospheric circulation would have shutdown upwelling in the Gulf of California and allowed

979 warm, tropical waters to flow into it, producing the warmer SSTs, as is currently observed during

980 Northern Hemisphere spring (Ganeshram and Pedersen, 1998; McClymont et al., 2012).

981   If the magnitudes of these SST warmings in the Santa Barbara Basin and the Gulf of

982 California are also representative of the summer air-temperature change in the central Sierra

983 Nevada, then the warming would have been more than enough to deglaciate the range (Sect.

984 5.4.).

985 5.6. *Hypothesized cause of the Tioga 4 deglaciation*

986   The close correspondence in timing between the start of the Tioga 4 deglaciation in

987 California ($16.20 \pm 0.13$ ka, as indicated by the $\delta^{13}C$ data from McLean's Cave; Fig. 8d) and the

988 start of Heinrich Event 1 in the North Atlantic ($16.22 \pm 0.04$ ka; Fig. 8c) suggests a mechanistic

989 connection between the two events. We hypothesize that Heinrich Event 1 withdrew such a



volume of ice from the Laurentide Ice Sheet (LIS) that atmospheric circulation reorganized over

North America in response. As the LIS shrinks and thins, the Aleutian Low weakens and moves

northward while the subtropical high in the eastern Pacific strengthens (Bartlein et al., 1998;

Otto-Bliesner et al., 2006; Wong et al., 2016; Jones et al., 2018). Also, as the LIS thins, it

becomes a less formidable obstacle to atmospheric circulation, which enables the polar jet stream

– formerly split by the LIS (Fig. 9a), with a weaker branch passing to the north of the ice sheet

and a stronger branch passing to the south (Kutzbach and Wright Jr, 1985; Manabe and Broccoli,

1985; Kutzbach and Guetter, 1986; Bromwich et al., 2004; Löfverström et al., 2014; Lora et al.,

2016; Wang et al., 2024) – to reunify (Fig. 9b). We interpret that these changes in the Aleutian

Low and the Subtropical High caused the ITCZ to shift northward over the eastern Pacific

Ocean, weakening the subtropical jet stream (McGee et al., 2018), reducing the frequency of

atmospheric rivers, and causing them to "land" on North America to the north of the central

Sierra Nevada (which is at 37–38° N). Based on the timing and latitude of lake-level highstands

(Fig. 8d), we interpret this northward shift in the mean latitude of landfalling atmospheric rivers

to be ~6°.

Interestingly, the TraCE-21k transient simulation of deglacial climate produced only one

major reorganization in atmospheric circulation over North America's west coast during the

deglaciation period – and it was a 7° northward shift in the polar jet stream (He et al., 2013; Lora

et al., 2016). This northward shift in the polar jet stream was coincident with polar jet

reunification and that reunification was triggered by a ~1.4 km thinning of the LIS, with the

TraCE-21k model switching from ICE-5G's "15.0 ka" ice-sheet topography to ICE-5G's "14.0

ka" topography (Peltier, 2004) at 13.87 ka in the transient model. Although we interpret

deglaciation of the Sierra Nevada as occurring at ca. 16.2 ka and being driven by a northward





expansion of Hadley Cell circulation on the eastern Pacific and a ~6° northward shift in the mean
latitude of land-falling atmospheric rivers (and not directly by a northward shift in the polar jet
stream), the TraCE-21k model results do underscore the sensitivity of North American
atmospheric circulation to LIS thickness and contain a northward shift in atmospheric circulation
that is comparable to what we see in the lake-level-highstand data (Fig. 8d).
Supporting evidence for the ca. 16.2 ka deglaciation of the Sierra Nevada being driven by
an atmospheric reorganization over North America in response to Heinrich Event 1 comes from
the West Antarctica Ice Sheet (WAIS) Divide deuterium record (Jones et al., 2018). This record
indicates interannual to decadal climate variability in the high southern latitudes was almost
twice as large during the LGM as it was during the Holocene. Jones et al. (2018) link this
enhanced variability to the mean location of tropical convection in the Pacific, with a thick LIS
causing a deeper and south-shifted Aleutian Low that weakens tropical Pacific winds. The
weaker winds then cause the western warm pool to spill eastward, as during modern El Niño
events. The enhanced isotopic variability at WAIS Divide comes to an abrupt end at ca. 16 ka,
implying a substantial thinning of the LIS at that time and a concomitant reorganization in
atmospheric circulation over North America (and the tropical Pacific).
In summary, we infer Heinrich Event 1 sufficiently thinned the LIS at ca. 16.2 ka to
trigger an atmospheric reorganization (Fig. 9). That atmospheric reorganization brought drier
winters and warmer summers to what is now the southwestern United States. In response, the
central and southern Sierra Nevada essentially deglaciated and formerly expansive lakes in
California, Arizona, New Mexico, and Utah desiccated (McGee et al., 2018). Offshore western
North America, the weakening of the Aleutian Low reduced the influx of polar water into the
Santa Barbara Basin, causing SSTs there to warm (Hendy and Kennett, 2000; Hendy et al., 2002;





Hendy, 2010), and the northward repositioning of the northeastern Pacific's subtropical high
shutdown upwelling in the Gulf of California, allowing subtropical water to enter the Gulf and
warm SSTs there (Ganeshram and Pedersen, 1998; McClymont et al., 2012).

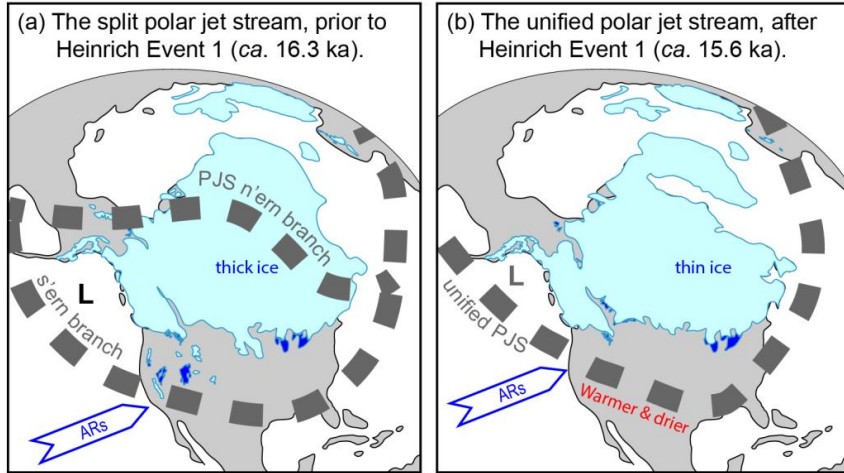

**Figure 9.** Cartoon illustrating how we propose the polar jet stream evolved in response to
Heinrich Event 1 and how we envision the principal location in landfalling atmospheric rivers
(ARs) changed. The removal of ice from the Laurentide Ice Sheet and the ice sheet's associated
height reduction facilitated reunification of the polar jet stream (PJS) and a northward shift in its
mean position (relative to its southern branch's former position). This reorganization in
atmospheric circulation brought warmer and drier conditions to California and the southern Great
Basin, driving the Tioga 4 deglaciation in the Sierra Nevada and the abandonment of lake-level
highstands in the southern Great Basin. Ice sheet and lake reconstructions simplified from Dyke
et al. (2003).



**6. Conclusions**
We present thirty-one new [10]Be and twenty-six recalculated [36]Cl dates (Phillips et al.,
2009) on the timing and duration of the Tioga 4 deglaciation in the Sierra Nevada Mountains of
California. We calculate these dates using three different surface-exposure calculators [the
Version 3 calculator (Balco et al., 2008), CRONUScalc 2.0 (Marrero et al., 2016), and CREp
(Martin et al., 2017)] and we interpret that the CRONUScalc and CREp results (which are
essentially identical) are a few percent more accurate (~5 %) than the results from the Version 3
calculator for the samples in this dataset. Adopting the CRONUScalc results as our preferred set
of surface-exposure dates, we find that the Tioga 4 deglaciation – which followed the last major
Pleistocene readvance in the Sierra Nevada and essentially deglaciated the range – began at 16.4
± 0.8 ka ($n$ = 7). Furthermore, we find that the range deglaciated in ca. 1.0 kyr (median estimate,
with a 95% confidence range of 0.2–3.0 kyr).
In addition, we use Bchron 4.7.6 (Haslett and Parnell, 2008) to construct new age-depth
models for the paleoenvironmental record from Swamp Lake (Fig. S23; Table S7; Street et al.,
2012) and for speleothem ML1 from McLean's Cave (Fig. S25; Table S9; Oster et al., 2015).
The principal benefit of these new age-depth models is that they provide age-uncertainty
estimates for the undated horizons in these records, which the original age-depth models lacked.
Examining the 16.4 ± 0.8 ka portions of these records, we find evidence of warming at 16.4 ± 0.4
ka in the Swamp Lake core (with no commentary on potential precipitation changes) and
evidence of drying (and possibly also warming) at 16.20 ± 0.13 ka in the McLean's Cave
speleothem. These three events – the onset of the Tioga 4 deglaciation, as recorded by [10]Be and
[36]Cl concentrations in morainal boulders, the warming recorded at Swamp Lake, and the drying
recorded at McLean's Cave – are indistinguishable in age and we interpret them to be different



manifestations of the same event: a warming and drying of the central Sierra Nevada that started
at 16.20 ± 0.13 ka.

We place the warming and drying of the central Sierra Nevada (36‑38° N) in a regional

context by comparing it with the timing of lake-level highstands in the Great Basin (Munroe and
Laabs, 2013; Ibarra et al., 2014) and with the start of rising SSTs in the Santa Barbara Basin
(Hendy et al., 2002) and the Gulf of California (McClymont et al., 2012). We find evidence
indicating a ~6° northward shift in the mean latitude of landfalling atmospheric rivers at ca. 16.2
ka.

Finally, we hypothesize that the Sierra Nevada's deglaciation and the southern Great

Basin's desiccation were the result of Heinrich Event 1 (Heinrich, 1988; Hemming, 2004) –
which started at 16.22 ± 0.04 ka and ended at 16.04 ± 0.04 ka (Bernal-Wormull et al., 2021;
Pérez-Mejías et al., 2021) – reducing the height and extent of the Laurentide Ice Sheet such that
it no longer split the polar jet stream into northern and southern branches as it passed over North
America. The polar jet stream reunified in response to a thinner and smaller ice sheet, and
adopted a more northerly position (as compared with the mean position of the jet stream's former
southern branch), bringing warmer and drier weather to California and the southern Great Basin,
driving the Tioga 4 deglaciation, lowering lake levels throughout the region, and warming SSTs
in the Santa Barbara Basin and the Gulf of California.



**Data availability**

The data necessary to recalculate the surface-exposure ages of the new $^{10}$Be concentrations

reported here are available in this article's online supplement. The supplement also contains field

photos of the sampled boulders and the new age-depth models we report for the Swamp Lake

and McLean's Cave paleoenvironmental reconstructions. The necessary files to reproduce the

snow-shielding calculations are available at:

https://github.com/BerylliumBecker/SierraNevadaSnowDepths.

**Author contributions**

RAB conceptualized the study. RAB, SAM, BT, and MWC secured the funding for fieldwork

and lab analyses. RAB collected the boulders for $^{10}$Be dating, with the aid of field assistants

acknowledged below. RAB and AMB processed the samples for $^{10}$Be, with the aid of lab

assistants acknowledged below. MWC and PRIME Lab colleagues measured $^{10}$Be/$^9$Be in the

samples. RAB calculated the snow-shielding correction factors, developed the new age-depth

models for the Swamp Lake and McLean's Cave paleoenvironmental reconstructions, and

recalculated the ages of legacy $^{36}$Cl and $^{14}$C data. RAB, AMC, SAM, and BT interpreted the

results. RAB wrote the manuscript's first draft and created the figures and tables. All the authors

edited the manuscript.

**Competing interests**

The authors declare that they have no competing interests.




**Acknowledgements**
Sampling permission was granted by the citizens of the United States, via their representatives in
the National Park Service and the U.S. Forest Service. We thank Greg Stock (Yosemite National
Park) and USFS District Ranger Ray Porter (Sierra National Forest) for their help in navigating
the permitting process. JT Holcombe, Will Montz, and Michael Bahrmasel helped collect the
$^{10}$Be samples. Alexander Horvath, Levi Mitchell, and Elizabeth Ceperley assisted with the
laboratory work. We thank them all. Funding was provided by the UW-Madison Department of
Geoscience, AAPG Foundation's Robert and Carolyn Maby Memorial Grant, a Geological
Society of America student research grant, and a Doctoral Dissertation Improvement Award
from the National Science Foundation (grant #1303194).





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
