# Peer review of "Deglaciation of the Sierra Nevada (USA) during Heinrich Event 1"

_EGUsphere, 2025_

## Author Comment (AC1)

**Response to Reviewer 1** (Dr. Matteo Spagnolo)

RC = Reviewer Comment

AR = Author Response

| | |
|---|---|
| **RC 1** | Dear Editor and Authors,

I read with great interest this manuscript, which attempts to chronologically constraint the post-Tioga4 deglaciation in the Sierra Nevada, linking it to key palaeoclimatic events that likely drove a relatively rapid and widespread glacier retreat in this region.

The quality and quantity of the work undertaken and presented is truly impressive. The authors have really considered many different aspects which could have impacted the results and looked at these in great detail. A good number of other proxies are also revisited and considered to strengthen their interpretation. The implications from a paleoglaciology and palaeoclimatology point of view are important. And the topic is certainly of interest to a wide international audience. All in all, I therefore encourage a swift acceptance for publication of the submitted manuscript. However, I believe there are a number of potential improvements, the most important ones I list below, in no particular order. I consider these of a minor nature, and I believe it should take relatively little effort to address these. Notice that I will also attach a version of the manuscript with some 55 comments and additional edits, which might be useful to improve the manuscript too. |
| **AR 1** | Dear Dr. Spagnolo: Thank you for these encouraging words and suggestions for improvement! |
| **RC 2** | 1. The manuscript is too long. I appreciate there is a lot to talk about, but to make it as attractive as possible to a wide audience I recommend some trimming. Could some aspects be moved the supplementary material? For example, do we really need such long narrative around the age calculations (though I appreciate the destination journal somewhat justify it)? Could this be moved to the supplementary and referred to in the main text? |
| **AR 2** | The manuscript is indeed long – and we accept the wisdom of moving some of the material (such as the discussion regarding the age-calculation methods) to the supplement, which will be reflected in our revised version. |
| **RC 3** | 2. A clearer justification of the main research focus, i.e. duration of deglaciation (rather than what most other cosmo glacio papers deal with, that is moraine ages). Why does it matter? |

| | |
|---|---|
| **AR 3** | The manuscript has three foci: (1) determining which cosmogenic surface-exposure calculator is producing ages most consistent with the radiocarbon dates on the timing of the Tioga 4 deglaciation, (2) applying the preferred set of ages to determine the timing and duration of the Tioga 4 deglaciation, and then (3) placing that timing (and that duration) in a larger context via a reanalysis of regional (and remote) paleoclimate proxies. We will emphasize this more clearly in the text.

Our principal motivation in attempting to constrain the Tioga 4 deglaciation's duration is motivated by the question *What can we reasonably infer about the past from the $^{10}Be$ concentration of scattered boulders in this formerly glaciated landscape?* That said, a quantitative reconstruction of deglaciation's duration is a jumping off point for reconstructing the minimum rate of equilibrium-line altitude (ELA) rise (e.g., Vacco et al., 2010).

We will make this point more explicit in a revised version of the manuscript.

Additionally, as the manuscript currently notes, deglaciation's duration provides a maximum constraint on the duration of the climate change that drove the deglaciation – although we do not think this maximum constraint is closely limiting. |
| **RC 4** | 3. Is there a wider lesson to be learned, which could be of relevance to other palaeoglaciology/climatology studies elsewhere, i.e. not specific to the Sierra or North America? |
| **AR 4** | Thank you for this prompt; we see two wider lessons to be learned from this manuscript:
1. **Geochronometer Considerations:** The three cosmogenic surface-exposure calculators we consider here produce ages that are offset by 5% on average despite identical inputs. For researchers working elsewhere, they should consider whether their interpretations are calculator dependent.
2. **Paleoclimate Inferences:** The timing of rapid deglaciation in the Sierra Nevada (USA) is indistinguishable from the timing of Heinrich Event 1 (HE1) in the North Atlantic, which suggests a causal link between the two. For researchers working elsewhere on the ca. 16 ka time period, they should consider whether this atmospheric reorganization also impacted their field area.

In our revised version, we will include more explicit language to emphasize and better summarize these primary points. |
| **RC 5** | 4. Somewhat related to point 1, though I appreciate the effort, I think that the work undertaken to establish snow shielding is a bit of an overkill, given that |

| | |
|---|---|
| | ultimately we have very little knowledge of how this varied in the past. Can the past 50 or a 100 years or even a few centuries be truly representative of conditions over a timeframe 2 orders of magnitude longer? I suggest the text that relates to this is reduced considerably. |
| AR 5 | Extrapolating snow conditions in the Sierra Nevada from the last ~50-100 years to the last *ca.* 16,000 years is indeed a challenging – but a snow correction is clearly required for developing an absolute chronology (see, for example, Fig. 2-5 in the lead author's dissertation) and extrapolating from recent snow observations to the distant past is the most commonly used approach to solving this problem (e.g., Gosse and Phillips, 2001).

Moreover, recent snow conditions in the Sierra Nevada (*i.e.*, the last ~50-100 years) seem to be reasonably representative of average snow conditions over the last *ca*. 16,000 years. For example, if *no* snow correction is applied to the ages, then variations in boulder height explain ~30% of the age variations between the samples (depending on choice of calculator (Fig. 2-5 in Becker's dissertation). However, if we (1) apply a correction for seasonal snow shielding and (2) assume it is equal to historical snow conditions in the Sierra Nevada (since ca. 1946), then boulder height explains <1% of the age variation.

And, unlike in most other surface-exposure dating studies that use historical snow conditions to correct for snow conditions over the postglacial interval, we propagate the modern uncertainties in snow-water equivalent (SWE) vs. elevation (Fig. S21 in the manuscript) and in average snowpack density vs. SWE (Fig. S22c) into our preferred set of cosmogenic surface-exposure age uncertainties for this study (*i.e.*, those from the CRONUScalc 2.0 (Marrero et al., 2016) calculator).

Thus, compared with most previous cosmogenic surface-exposure dating studies that use historical snow conditions to correct for snow shielding over the "post-glacial interval" (however long that interval might be), the approach we describe in this manuscript a more comprehensive treating of this uncertainty and thus is more likely to be representative of that longer interval (the past *ca*. 16,000 years in this case) than the reported age uncertainties in those other publications.

All this said, while we wish to preserve the supplement's detailed description of the snow-shielding correction – because future work (including some of our upcoming work) would like to cite back to this paper – we have made the following changes to shorten the supplement:
  1. We have moved Tables S4–S6 from the supplement to GitHub; the supplement now contains links to these tables in Excel and pdf formats.
  2. In Section S.5.1.1., we refer readers to Appendix B of Becker's dissertation for a description of how the daily snow-water equivalent (SWE) records were cleaned. |

|  | 3. We have moved Figures S11–S19 from the supplement to GitHub; the supplement now contains links to these figures. Figure S10 is retained in the supplement to give readers a flavor for the information that these figures contain. |
|  | Collectively, these three changes have shortened the portion of the supplement devoted to the snow-shielding correction by 50 %, from 24 pages to 12, while preserving all the information that was in it before. |
| RC 6 | 5. Please try to steer away from the description of glacial dynamics in terms of percentage of LGM glacier length and use ELA instead, even if transient (not connected to a moraine), which is far more important and robust glacier mass balance wise. |
| AR 6 | We agree with this suggestion and we will replace most of our "percentage of LGM glacier length" estimates with toe-to-headwall-based ELA reconstructions. With regards to the Tioga 4 readvance, previous authors have noted that it extended to ~60 % LGM-glacier length (e.g., Clark, 1976; Clark and Clark, 1995; Phillips et al., 2009), and we would prefer to retain mention of this relative length in the introductory sections of the manuscript for consistency with the previously published literature, while supplementing it with ELA reconstructions. |
| RC 7 | 6. Could you quantify a retreat rate? This will help putting your "rapid and abrupt" deglaciation into context. |
| AR 7 | Yes, these rates were reported in Becker's dissertation and we will include those values in our revised manuscript. |
| RC 8 | 7. Figure-wise, all great and useful. Dare I ask you to consider providing a 3D cartoon-like/schematic diagram showing the glacier retreating through the studied sites, exemplifying key geomorphological/geological evidence that justify your conclusion along with the ages of the various samples? These diagrams could perhaps be added to figs 3-5. |
| AR 8 | Our concern with providing a 3D cartoon-like / schematic diagram that shows the various phases of glacial retreat is that these cartoons would be fairly unconstrained with regard to volumetric changes and would be more accurate if informed by a numerical model of ice-extent through deglaciation. Figures 3-5 provide what is perhaps the best reconstruction of our glacial changes and what we are most comfortable providing at this point. |
| RC 9 | Looking forward to seeing this great work published as soon as possible, Matteo Spagnolo |

| | |
|---|---|
| **AR 9** | Thank you for your encouraging remarks. |
| **RC 10** | **The remaining reviewer comments in this document come from an annotated copy of the preprint with 56 comments by Dr. Spagnolo.**

Screenshot from the preprint…

26    0.4 ka and drying at 16.20 ± 0.13 ka. Collectively, we interpret that rapid deglaciation began at

27    16.20 ± 0.13 ka. This timing is indistinguishable from that of Heinrich Event 1 (HE1), which

Comment:
I would delete this point. It seems to contradict your earlier statement: "These dates indicate rapid deglaciation began at 16.4 ± 0.8 ka and lasted for ca. 1.0 kyr.". With relation to the timing of these two climatic signals, the deglaciation could have, and likely did, started with the slightly-earlier warming at 16.4. All ages here are very much compatible, given their error. |
| **AR 10** | Thank you for bringing this issue to our attention. We will delete this sentence and otherwise modify the abstract so it reads:

"A polar jet stream (PJS) split by the Laurentide Ice Sheet (LIS) is a well-established feature of Ice-Age atmospheric circulation in many general circulation models. California's central Sierra Nevada Mountains (37–38° N) lie near the reconstructed position of the PJS's southern branch. Previous glacial studies concluded that rapid deglaciation began here at ca. 16–15 ka after millennia of relatively glacial stability at ~60 % Last Glacial Maximum glacier length. However, this conclusion is largely based on the behavior of glaciers in a single valley (Bishop Creek Canyon). Here, we report 31 new [10]Be-derived ages from two locations (Lyell Canyon and Mono Creek Canyon) and 26 recalculated [36]Cl dates from Bishop Creek Canyon. These dates indicate rapid deglaciation began at 16.4 ± 0.8 ka. Placing two previously published paleoenvironmental reconstructions with centennial resolution on revised age-depth models, we also find evidence for coeval warming in the central Sierra Nevada at 16.4 ± 0.4 ka and drying at 16.20 ± 0.13 ka. Collectively, we interpret that rapid deglaciation began by 16.0 ka. This timing is indistinguishable from that of Heinrich Event 1 (HE1), which occurred between ca. 16.5 ka and ca. 15.9 ka. We hypothesize that the Sierra Nevada's deglaciation was driven by a northward repositioning and focusing of the winter-storm track over western North America in response to PJS reunification, bringing warmer and drier weather to the central Sierra Nevada, and that PJS reunification occurred in response HE1 thinning the LIS." |
| **RC 11** | Screenshot from the preprint…

30    western North America in response to PJS reunification, bringing warmer and drier weather to

31    the central Sierra Nevada, and that PJS reunification occurred in response HE1 thinning the LIS. |

| | |
|---|---|
| | Comment:
to |
| AR 11 | Thank you for catching this typo.  We have now made the change. |
| RC 12 | Screenshot from the preprint…

57    et al., 2010a; Laabs et al., 2011; Rasmussen et al., 2014; Bahr et al., 2018; Barth et al., 2018;

58    Pedro et al., 2018; Li and Born, 2019),

Comment:
Given the focus of the paper, it would be really important to add a sentence here on how studies on deglaciation rates could play a role in this general goal |
| AR 12 | This manuscript's focus evolved as it was drafted – and in hindsight we think the submitted version retained too much emphasis on providing readers with a single, probabilistic estimate for the duration of the Tioga 4 deglaciation, given that its duration (as we define it) would have varied with local ice thickness.

While we wish to keep this material as an ancillary point within the manuscript, we do not think the estimate is of sufficient importance to justify a sentence within the Introduction's opening paragraph. Rather, we will add a sentence on the utility of deglaciation-duration estimates to paleoclimate reconstructions to what is currently the Introduction's fourth paragraph, which introduces the duration of the Tioga 4 deglaciation as an important concept within the manuscript. |
| RC 13 | Screenshot from the preprint…

59    Field-based studies in the Sierra Nevada Mountains (USA) – such as geomorphic

60    observations (Clark, 1976; Clark and Clark, 1995) and $^{10}$Be, $^{26}$Al, and $^{36}$Cl surface-exposure

Comment:
A sentence or two, perhaps on N America, connecting the first global perspective paragraph and this very detailed, Sierra focused one would improve the narrative of the intro. |
| AR 13 | Yes, we agree. Because this manuscript focuses on Heinrich Event 1 and how it indirectly drove the Tioga 4 deglaciation, and Heinrich Events were mentioned in the opening paragraph, we will use Heinrich Events as the transition from the first to second paragraphs of the Introduction. |
| RC 14 | Screenshot from the preprint…

61    dating (James et al., 2002; Phillips et al., 2009) – indicate that the range's glaciers responded

62    strongly to a large and abrupt change in climate near the end of the last deglaciation. After Sierra

Comment: |

| | |
|---|---|
| | can you be more specific? |
| **AR 14** | Yes, we will modify this sentence so it includes the dates that James *et al.* (2002) and Phillips *et al.* (2009) inferred for this climate transition. |
| **RC 15** | Screenshot from the preprint…

62    strongly to a large and abrupt change in climate near the end of the last deglaciation. After Sierra

63    Nevadan glaciers reached their greatest extents, broadly in-phase with the global Last Glacial

64    Maximum (LGM; Phillips et al., 2009; Rood et al., 2011), they receded for several thousand

Comment:
provide a time range for this, for those who might not be familiar with the glacial history of our planet |
| **AR 15** | Yes, we agree that providing a numeric age for the LGM would improve the accessibility of this manuscript. |
| **RC 16** | Screenshot from the preprint…

64    Maximum (LGM; Phillips et al., 2009; Rood et al., 2011), they receded for several thousand

65    years – with episodic advances superimposed on this general trend – before readvancing to ~60

66    % LGM-length. Glaciers in the Sierra Nevada rapidly melted following this readvance (Clark,

Comment (on "before readvancing"):
when? |
| **AR 16** | Phillips (2016) interprets that the Tioga 4 readvance began at *ca.* 16.8 ka. We'll add that age and reference to this sentence (or otherwise revise the paragraph such that it is included). |
| **RC 17** | Screenshot from the preprint…

66    % LGM-length. Glaciers in the Sierra Nevada rapidly melted following this readvance (Clark,

67    1976; Clark and Clark, 1995), receding from ~60 % LGM-length to <3 km length (0–23 %

Comment:
how many glaciers? How widespread and generally accepted is this conclusion? |
| **AR 17** | At least fourteen glaciers in the Sierra Nevada and one in the White Mountains deglaciated in response to the climate change that drove the Tioga 4 deglaciation. These fifteen glaciers span 290 km of the range, from Old Man Mountain in the north at 39.4° N to the South Fork of Bishop Creek Canyon at 37.2° N. The conclusion that Sierra Nevadan glaciers rapidly melted following the Tioga 4 readvance is generally accepted by those who have studied the range's deglaciation (e.g., Clark, 1976; Nishiizumi et al., 1989; Clark and Clark, 1995; |

| | |
|---|---|
| | Phillips et al., 1996, 2009; Evans et al., 1997; James et al., 2002; Clark et al., 2003; Gillespie and Clark, 2011; Phillips, 2016, 2017; Putnam and Hatchett, 2017). |
| RC 18 | Screenshot from the preprint…

66   % LGM-length. Glaciers in the Sierra Nevada rapidly melted following this readvance (Clark,

67   1976; Clark and Clark, 1995), receding from ~60 % LGM-length to <3 km length (0–23 %

68   LGM-length) in "500 yr or less, and certainly less than 1000 yr" (Phillips et al., 2009, p. 1031).

Note: RC 18 is about the struck-out text, not the highlighted text and associated comment. |
| AR 18 | We will reword this sentence such that it is compatible with the reviewer's suggestion to steer away from describing glacier-length variations and instead focus on ELA variations, as described in RC/AR 6. |
| RC 19 | Screenshot from the preprint…

66   % LGM-length. Glaciers in the Sierra Nevada rapidly melted following this readvance (Clark,

67   1976; Clark and Clark, 1995), receding from ~60 % LGM-length to <3 km length (0–23 %

68   LGM-length) in "500 yr or less, and certainly less than 1000 yr" (Phillips et al., 2009, p. 1031).

Comment:
what is this chronology based on? Is the 1-23% connected with the Recess Peak, which was a few millennia later? |
| AR 19 | This chronology is based on the average age difference between $^{36}$Cl dates from the Tioga 4 moraine in the Middle Fork of Bishop Creek Canyon and from boulders in Humphreys Basin. Humphreys Basin is a relatively expansive (~17 km$^2$), low-relief (~200 m), high-altitude (~3,350–3,550 m) region between the cirques and the glacially eroded U-shaped valleys (37.265032, -118.706515). The five $^{36}$Cl dates on boulders from Humphreys Basin date the end of large-scale deglaciation in the Sierra Nevada (Clark and Gillespie, 1997; Phillips et al., 2009; Phillips, 2016).

Yes, the 1–23 % statement is connected with the Recess Peak glacial advance. The 23 % LGM-glacier length statement is based on the glacier that was east of Mount Barnard. During the LGM, this glacier was 5.13 km long and during the Recess Peak glacial maximum it was 1.20 km long (Moore, 1981).

To clarify our point, we will update the manuscript with ELA estimates, as we think these will better describe the climate changes associated with these glacial advances and retreats than the percentage-of-LGM-length statistics. |
| RC 20 | Screenshot from the preprint… |

| | |
|---|---|
| | 72    This readvance and subsequent deglaciation suggest two reorganizations in atmospheric

73    circulation over the northeastern Pacific Ocean. An earlier reorganization that brought cooler

**Purple Comment:**
final

**Orange Comment:**
this implies a pre re-organisation circulation of sort which has not been described |
| AR 20 | With regards to the purple comment, we will revise this sentence for greater clarity, as suggested.

With regards to the orange comment, yes, there would have been some sort of atmospheric circulation pattern over western North America, then there was an atmospheric reorganization at ca. 16.7 ka (Phillips, 2016, 2017) that brought cooler summers and wetter winters to the Sierra Nevada, driving the Tioga 4 readvance, and then at ca. 16.2 ka there was another atmospheric reorganization that brought warmer summers and drier winters to the Sierra Nevada. Based on the response of Sierra Nevadan glaciers to that second reorganization – widespread and rapid deglaciation – we infer that the second atmospheric reorganization was a larger magnitude event than the first reorganization. |
| RC 21 | Screenshot from the preprint…

73    circulation over the northeastern Pacific Ocean. An earlier reorganization that brought cooler

74    summers and/or wetter winters to the Sierra Nevada (driving the readvance) and a later, larger

**Comment:**
~60 % LGM-length |
| AR 21 | Yes, the earlier reorganization in atmospheric circulation is the one that drove the Tioga 4 readvance, which culminated with glacier lengths that were ~60 % LGM glacial lengths. We'd be pleased to clarify this sentence in a revised version of the manuscript. |
| RC 22 | Screenshot from the preprint…

75    magnitude and more permanent reorganization that brought warmer summers and/or drier

76    winters to the range (driving the deglaciation). This inference, derived from the Sierra Nevada's

**Comment:**
latest/final |
| AR 22 | Yes, "driving the final deglaciation" of the range – if we ignore the subsequent (ca. 14–13 ka) Recess Peak glacial advance, which was relatively minor (Clark and Gillespie, 1997; Phillips, 2016, 2017). |

| | |
|---|---|
| RC 23 | Screenshot from the preprint… |
| | 79    16.5 ka (Praetorius et al., 2020). Numerical modeling shows these cooler SSTs would have |
| | 80    generated a deeper and longer lasting snowpack in the Sierra Nevada, with warmer SSTs having |
| | 81    the opposite effect (Peteet et al., 1997). |
| | Comment:
is it truly a cause-effect mechanism? |
| AR 23 | Yes, we think oceanic conditions offshore western North America influence climate conditions onshore, such as temperature, precipitation, and snow-depth and snow-duration patterns. |
| RC 24 | Screenshot from the preprint… |
| | 82    The duration of the Tioga 4 deglaciation provides a maximum constraint on the duration |
| | 83    of the second reorganization. However, the cosmogenic surface-exposure ages constraining the |
| | Comment on "Tioga 4":
this terminology must first be introduced to the generic reader who is not familiar with the Sierra glacial history |
| AR 24 | We agree. While we initially resisted providing a full summary of Sierra Nevada glacial terminology in the Introduction – preferring to save the fuller, deeper treatment for section 2, Regional setting – we have come to realize as a result of this comment and a comment by Reviewer #2's (RC 76) that the section on the glacial terminology of the Sierra Nevada needs to be moved into the Introduction. |
| RC 25 | Screenshot from the preprint… |
| | 82    The duration of the Tioga 4 deglaciation provides a maximum constraint on the duration |
| | 83    of the second reorganization. However, the cosmogenic surface-exposure ages constraining the |
| | Comment on "deglaciation":
could you really refer to Tioga4 as a deglaciation, rather than a re-advance or stillstand during the generalised deglaciation trend of the lateglacial? If so, you should use something like "post-Tioga4 deglaciation" here. |
| AR 25 | The Tioga 1–4 terminology was introduced by Phillips et al. (1996) to describe clusters of [36]Cl-dated moraine ages in three valleys on the east side of the Sierra Nevada and one valley in the adjacent White Mountains, with the "Tioga 4" grouping being the youngest and least extensive glacial advance of the four. In the strictest definition of these terms, they refer to the moraines. However, each of these moraines reflects a period of glacial growth and a subsequent period of glacial retreat, hence our usage of the terms "Tioga 4 readvance" and "Tioga 4 deglaciation." |

| | |
|---|---|
| | We will more clearly define what we mean by these terms in a revised version of the text. |
| RC 26 | Screenshot from the preprint…

82     The duration of the Tioga 4 deglaciation provides a maximum constraint on the duration

83    of the second reorganization. However, the cosmogenic surface-exposure ages constraining the

Comment:
is this the key research question of this paper, or one of? If so, it would be good if you could spend a few more words explaining why this matter, more generally. |
| AR 26 | As mentioned in AR 3 and AR 12, constraining the duration of the post-Tioga 4 deglaciation is one of this manuscript's foci, but not the most important one. Please see AR 3 and AR 12 for more discussion of this topic. |
| RC 27 | Screenshot from the preprint…

83    of the second reorganization. However, the cosmogenic surface-exposure ages constraining the

84    duration of this deglaciation to <500–1000 years (Phillips et al., 2009) are all associated with a

Comment:
phase or stadial would be better |
| AR 27 | We agree that we should more precisely define what we mean by "the duration of deglaciation" – but we disagree that "stadial" would be a better alternative. A stadial is a period of glacial advance or of colder weather associated with a glacial advance (Foster Flint, 1971, p. 372; Andrews and Voelker, 2018). Here, we are interested in the period of glacial retreat and why the climate of the Sierra Nevada transitioned from being conducive to moderate glaciation (with glaciers at lengths equal to about 60 % their LGM lengths) to being inconducive to glaciation (with only cirque glaciers in the range, if any glaciers at all).

As described in AR 25, we will more clearly define what we mean by the terms "the Tioga 4 readvance" and "the Tioga 4 deglaciation." |
| RC 28 | Screenshot from the preprint…

85    single former glacier – the Bishop Creek Glacier – thus permitting the possibility that glacial

86    recession was substantially slower (or more rapid) elsewhere in the range.

Comment:
this |
| AR 28 | Thank you for catching this typo; we will insert the missing word. |

| | |
|---|---|
| **RC 29** | Screenshot from the preprint…
127    midMC – middle Mono Creek; and uppMC – upper Mono Creek. The other abbreviation on
128    panel d: PB – Pioneer Basin. **(e)** 2014 aerial photo of the Lake Edison moraine complex, which
129    was exposed by drought. **(f)** Map of Bishop Creek Canyon and vicinity. Sampling-location
Comment:
is this the inset in d? Also, where is f? |
| **AR 29** | Yes, any earlier version of the figure had the inset in panel (d) labeled as panel (e) and then the map of Bishop Creek Canyon (the panel in the lower right) labeled as panel (f). When we updated the figure to the current version, we forgot to update the caption. The caption is correct as written – except "(e)" should be "Inset:" |
| **RC 30** | Screenshot from the preprint…
143    The range's length, orientation, and elevation make it an effective barrier to maritime air
144    masses originating over the Pacific (e.g., Pandey et al., 1999). Furthermore, the range
Comment:
this implies a east-ward circulation, presumably present-day (what about in the past?) which you have not described |
| **AR 30** | Yes, this sentence does assume that the atmospheric circulation over the Sierra Nevada is predominantly west to east, which is the modern direction (e.g., Pandey et al., 1999), and the direction produced in numerous climate models of Ice-Age atmospheric circulation (e.g., Bartlein et al., 1998; Bromwich et al., 2004; He, 2011; Lora et al., 2016). |
| **RC 31** | Screenshot from the preprint…
146    abundant snowfall. In Tuolumne Meadows for example (~2600 m elevation), 82% of its 1985–
147    2017 precipitation arrived in the six months between November 1 and April 30 and 50% of that
148    precipitation was snow (http://cdec.water.ca.gov). Average April snow depths at the elevation of
Comment:
how much, on average, per year? |
| **AR 31** | Annual average precipitation in Tuolumne Meadows between 1985 and 2017 was 131 cm. |
| **RC 32** | Screenshot from the preprint…
153    1996; Phillips et al., 2009) and then the comparatively minor Recess Peak (Birman, 1964; Clark
154    and Gillespie, 1997) Tioga 1 predates both the local and global LGM (Phillips et al., 1996;
155    Phillips et al., 2009); Tioga 2 marks the local LGM, which was ca. 24–21 ka (e.g., Clark et al.,
Comment:
if this was not the local LGM, how did it get preserved during Tioga 2? |

| | |
|---|---|
| **AR 32** | Two Tioga 1 moraines are reported from the Sierra Nevada, one from Little McGee Canyon (Phillips et al., 1996) and one from Bishop Creek Canyon (Phillips et al., 2009). The Tioga 1 moraine in Little McGee Canyon was preserved by normal faulting and the Tioga 1 moraine in Bishop Creek Canyon was preserved by an avulsion of the glacier through its lateral moraine. Phillips et al. (2009) speculates that the avulsion was caused by meltwater running off the glacier and cutting a channel through the lateral moraine. |
| **RC 33** | Screenshot from the preprint…

 156    2003; Phillips et al., 2009; Amos et al., 2010; Rood et al., 2011); Tioga 3 represents a readvance
 157    to ~90 % the LGM ice extent ca. 23–20 ka (e.g., Schaefer et al., 2006; Phillips et al., 2009; Stock

 Comment (line 156):
 how do we know is a readvance rather than a stillstand during retreat? |
| **AR 33** | We do not know whether the Tioga 3 moraines represent a readvance or a stillstand during retreat. Phillips et al. (1996) developed the Tioga 1–4 terminology and interpreted each of those four moraine sets as representing a glacial advance. While Phillips et al. (1996) offers no justification for interpreting the Tioga 3 moraines as a readvance versus a recessional stillstand, this interpretation has been widely accepted within the literature on the glacial history of the Sierra Nevada (e.g., Gillespie and Clark, 2011; Rood et al., 2011).

 Although we think a glacial readvance is more likely than a recessional stillstand – per the arguments of Anderson et al. (2014) that "interannual climate variability" (weather) can drive kilometer-scale glacial advances for glaciers with average lengths of 20 km – we will modify this sentence to read: "… Tioga 3 represents a readvance to (or recessional stillstand at) ~90 % LGM-glacier-length, with a reconstructed ELA on the eastern side of the Sierra Nevada of ~2,500–2,700 m (Phillips, 2017), at ca. 23–20 ka (e.g., Schaefer et al., 2006; Phillips et al., 2009; Stock…" |
| **RC 34** | Screenshot from the preprint…

 155    Phillips et al., 2009); Tioga 2 marks the local LGM, which was ca. 24–21 ka (e.g., Clark et al.,
 156    2003; Phillips et al., 2009; Amos et al., 2010; Rood et al., 2011); Tioga 3 represents a readvance
 157    to ~90 % the LGM ice extent ca. 23–20 ka (e.g., Schaefer et al., 2006; Phillips et al., 2009; Stock
 158    and Uhrhammer, 2010); and Tioga 4 marks a readvance to ~60 % LGM ice extent ca. 16 ka

 Comment (line 157):
 this timing overlaps with the LGM as described above.. |

| | |
|---|---|
| **AR 34** | The Tioga 2 moraines are stratigraphically older than the Tioga 3 moraines and the Tioga 3 moraines are stratigraphically older than the Tioga 4 moraines. The age ranges we give here for the moraines – with the Tioga 2 (local LGM) dating to ca. 24-21 ka and the Tioga 3 dating to ca. 23-20 ka, for example – reflects the uncertainty in cosmogenic surface-exposure dating. |
| **RC 35** | Screenshot from the preprint…

 157   to ~90 % the LGM ice extent ca. 23–20 ka (e.g., Schaefer et al., 2006; Phillips et al., 2009; Stock

 158   and Uhrhammer, 2010); and Tioga 4 marks a readvance to ~60 % LGM ice extent ca. 16 ka

 159   (Phillips et al., 2009; Phillips, 2016).

 Comment:
 Ditto |
| **AR 35** | The argument that the "confluence moraine" at the junction of the Middle Fork of Bishop Creek Canyon with the South Fork of Bishop Creek Canyon represents a readvance and not a stillstand of the ice margin during retreat originates with (Bateman et al., 1965, p. 173-174 of the pdf). He noted the steepness of the descending lateral moraines and interpreted that they recorded a readvance, rather than a stillstand.

 Phillips (2017, p. 538) discusses this interpretation – that the Sierra Nevada's Tioga 4 moraines mark a readvance and not a stillstand – and notes (once again) the steepness of descent of the Tioga 4 lateral moraines. Phillips (2017) also mentions the general absence of recessional moraines behind the Tioga 4 moraines, and concludes that the Tioga 4 moraines likely represent a readvance (rather than a stillstand), while noting that it is difficult to absolutely prove that they reflect a readvance.

 We will revise the text to read as follows: "… and Tioga 4 marks a readvance to (or recessional stillstand at) ~60 % LGM-glacier-length, with a reconstructed ELA on the eastern side of the Sierra Nevada of ~2,900 m (Phillips, 2017), at ca. 16 ka (Phillips et al., 2009; Phillips, 2016)." |
| **RC 36** | Screenshot from the preprint…

 165   (Clark and Gillespie, 1997). This paper focuses the Tioga 4 readvance and the subsequent

 166   deglaciation.

 Comment:
 on |
| **AR 36** | Thank you for catching that typo; we will correct the sentence. |

| RC 37 | Screenshot from the preprint… |
|---|---|
| | 171    Phillips et al., 2009; Marcott et al., 2019), the innermost end moraines of the Sierra Nevada – the

172    Tioga 4 moraines – are ~60 % the distance 15 and 22 km in Bishop Creek and Mono Creek |
| | Comment:
This comment is not pertinent to this sentence in particular, but, in general, I find it odd that so much importance is given to the glacier length (or percentage of), which would naturally vary depending on several factors, and especially the hypsometry of the glacier bed. Why not using ELA instead? |
| AR 37 | Thank you for the suggestion of using an ELA reconstruction instead of a % LGM-glacier-length reconstruction. While we will retain some mention of the Tioga 4 moraines reflecting glacier lengths that were roughly 60 % LGM-glacier lengths, to give readers a sense of the relative lengths, and because the "~60 % LGM-glacier-length" statistic is relatively common with the literature on the Tioga 4 moraines, in general we will switch the text to an ELA-based perspective. |
| RC 38 | Screenshot from the preprint… |
| | 174    1995; Phillips et al., 2009). Distal of ~60 %, end moraines are common and bedrock basins are

175    typically sediment-filled; proximal of ~60 %, bedrock basins are water-filled, bedrock-outcrop

176    exposure is excellent, and glacial sediments are primarily isolated boulders and till patches

177    (Clark, 1976; Clark and Clark, 1995). |
| | Comment:
Is this change abrupt? Otherwise one could argue that it is a simple sediment availability issue, rather than duration

Are we talking about nested Tioga 4 end moraines, or in between 3 and 4? |
| AR 38 | Clark (1976) and Clark and Clark (1995) argue this change is abrupt. The abundant end moraines between 100 % LGM-glacier length and ~60 % LGM-glacier length include the Tioga 2, 3, and 4 moraines. The innermost end moraine in Sierra Nevada valleys is typically at ~60 % LGM-glacier length (the Tioga 4 moraines); this moraine marks the boundary between to the two sedimentary regions, with bedrock basins commonly sediment-filled distal of ~60 % LGM-glacier length and commonly water-filled proximal of ~60 % LGM-glacier length. |
| RC 39 | Screenshot from the preprint… |
| | 180    (24–22 km; Figs. 1d–e and 2) – till thickness systematically decreases from the outermost to

181    innermost moraine; immediately behind the innermost moraine, till is absent for 4 km (Birman, |

| | |
|---|---|
| **AR 39** | Thank you for this suggestion; we will delete this word. |
| **RC 40** | Screenshot from the preprint…

184    also consistent with rapid deglaciation. Phillips (2016) correlated the entire Lake Edison moraine

185    complex with the Tioga 4 glacial advance and used the elevation difference between the

186    moraines and various unspecified cirque headwalls in the Mono Creek drainage to reconstruct an

187    equilibrium-line altitude (ELA) of 2800 m for the Tioga 4 advance. Based on Pioneer Basin's

Comment:
but how do we know glaciers retreated this far or disappear altogether? |
| **AR 40** | We know that Sierra Nevadan glaciers at least retreated into their cirques or *very* close to them during the deglacial phase that followed the Tioga 4 readvance because they must have been at least as small as the relatively minor Recess Peak glacial advance. The Recess Peak readvance was between ca. 14 ka and ca. 13 ka and the largest glacier in the Sierra Nevada during that readvance was approximately 4 km long (Phillips, 2017). Thus, at the end of the deglacial phase that followed the Tioga 4 readvance, the largest glacier in the Sierra Nevada cannot have been longer than 4 km.

With regards to estimating the ELA responsible for the Tioga 4 readvance (as the sentence on lines 184-187 does), it doesn't matter how extensive the glacial retreat was following the readvance. |
| **RC 41** | Screenshot from the preprint…

187    equilibrium-line altitude (ELA) of 2800 m for the Tioga 4 advance. Based on Pioneer Basin's

188    cirque-floor elevation (~3400 m), this reconstruction implies the Tioga 4 deglaciation was driven

189    by a ≥600 m ELA rise. This ELA rise is a minimum estimate because the climatological

Comment (line 188):
I think you need to be careful and very consistent with the terminology. Is Tioga 4 the name given to the re-advance? If so, best not to refer to a Tioga 4 deglaciation. Rather, you may want to say post-Tioga4 deglaciation. |
| **AR 41** | As described in AR 25, we will revise the text to clearly define our terminology.

Tioga 4 is the name given to the innermost Pleistocene end moraines of the Sierra Nevada (excluding the relatively minor Recess Peak moraines; Clark and Gillespie, 1997), which are at ~60 % LGM-glacier length (Clark, 1976; Clark and Clark, 1995; Phillips et al., 2009). The Tioga 4 term was introduced by Phillips et al. (1996). The Tioga 4 moraine is inferred to represent a glacial readvance (and not a stillstand; Bateman et al., 1965; Phillips et al., 2009). We use the term "Tioga 4 readvance" to refer to the time when Sierra Nevadan glaciers had a positive mass balance and |

| | |
|---|---|
| | were expanding to what would become their Tioga 4 limits. We use the term "Tioga 4 deglaciation" to refer to the time when Sierra Nevadan glaciers had a negative mass balance and were downwasting and shrinking back from their Tioga 4 moraines. We will more clearly define these terms in a revised version of the manuscript. |
| RC 42 | Screenshot from the preprint…

189  by a ≥600 m ELA rise. This ELA rise is a minimum estimate because the climatological

190  snowline may have risen above the elevation of the cirque floors.

Comment:
it might also be an overestimation, based on the ELA methodology applied |
| AR 42 | While all ELA estimates are subject to random errors that might produce either an overestimate or an underestimate, we are unaware of any evidence that the toe-to-headwall altitude ratio (THAR) method of reconstructing ELAs is prone to producing overestimates. Indeed, Meierding (1982) compared six methods for reconstructing former ELAs and the THAR and the accumulation-area ratio (AAR) methods produced the lowest root mean square errors (RMSEs) of all six methods. The THAR and AAR methods produced RMSEs of approximately 80 m. The other four methods considered were the glaciation threshold, the median altitude of small reconstructed glaciers, the elevation of the lowest cirque floors, and the highest elevation of lateral moraines. These four methods all produced reconstructed ELAs with higher RMSEs than the THAR and AAR methods. |
| RC 43 | Screenshot from the preprint…

205  landforms – Clark (1976) and Clark and Clark (1995) interpreted that Sierra Nevadan glaciers

206  stagnated and rapidly melted at the end of the last glaciation.

Comment:
after the Tioga 4 readvance. |
| AR 43 | Thank you for this suggested rewording. We will adopt it. |
| RC 44 | Screenshot from the preprint…

209  deglaciation and placed this event at 15.75 ± 0.5 ka (1 σ). This determination was based on three

210  bulk-sediment radiocarbon dates from high-altitude ponds within the Tioga 4 glacial footprint

211  (Fig. 1b). From north to south, and when calibrated with Calib 8.2 (Stuiver and Reimer, 1993)

Comment:
how much higher than the Tioga4 moraines in those valleys? |

| AR 44 | On average, these three high-altitude ponds are about 525 m higher in elevation than the moraines for which we report new or recalculated ages. The three ponds (north to south) are at elevations of 2,625 m (Highland Lakes), 3,091 m (Greenstone Lake), and 2,864 m (East Pond) while the dated Tioga 4 moraines are at 2,330 m (Lake Edison) and 2,340 m (Bishop Creek Canyon).

We will revise the sentence to read (new text in blue): "This determination was based on three bulk-sediment radiocarbon dates from high-altitude ponds (2,625–3,091 m) within the Tioga 4 glacial footprint (Fig. 1b)." |
|---|---|
| RC 45 | Screenshot from the preprint…

242   (Phillips, 2016) as the best estimate for the timing of deglaciation in the high-elevation lake

243   basins elsewhere in the Sierra Nevada.

Comment:
there is of course a question of representativeness. How much evidence from a few sites can be taken to describe glacier dynamics across a whole mountain range? Could there be arguments for variability in glacier dynamics? |
| AR 45 | First, while more minimum-limiting radiocarbon dates on the timing of the "Tioga 4 deglaciation" (the widespread deglaciation that occurred immediately after the culmination of the Tioga 4 readvance) would certainly be welcome, these three dates are the constraints that presently exist. With the radiocarbon (Clark et al., 1995, 2003; Power, 1998; Phillips, 2016) and cosmogenic surface-exposure dating (Rood et al., 2011) constraints presently available, there's no evidence for variability in glacier dynamics or for a latitudinal gradient in the timing of deglaciation in the Sierra Nevada, such as a later deglaciation in the northern Sierra Nevada than in the southern Sierra Nevada.

More fundamentally – within the context of this manuscript – our goal in reviewing the radiocarbon constraints on the timing of the "Tioga 4 deglaciation" is to establish a framework for evaluating the ages reported by the three online surface-exposure age calculators. Within this context, the most relevant radiocarbon date is the 15.69 ± 0.09 ka basal date from Greenstone Lake (Clark et al., 2003), which dates the deglaciation and subsequent revegetation of a landscape that was within 4 km of the Tuolumne Glacier's accumulation zone. Ice from within 4 km of Greenstone Lake flowed across Tioga Pass and then on through Tuolumne Meadows (Wahrhaftig et al., 2019), passing 1–2 km from our sampling sites in Tuolumne Meadows. Considering the full radiocarbon-dating-uncertainty envelope for the Greenstone Lake sample, we can be 99 % confident that Greenstone Lake was both deglaciated and accumulating organic carbon by 15.4 ka. Although we cannot be completely certain that our [10]Be sampling sites in Tuolumne Meadows were also deglaciated by then, it seems likely – given the proximity of the Greenstone Lake radiocarbon date to our [10]Be sampling sites and |

the methodological fact that radiocarbon dates the revegetation of a landscape while the $^{10}$Be dates the deglaciation of a landscape.

As is discussed later in the manuscript, the CRONUScalc 2.0 (Marrero et al., 2016) and CREp (Martin et al., 2017) calculators produce $^{10}$Be surface-exposure ages for our sampling sites in Tuolumne Meadows that are more consistent with the deglacial chronology provided by the Greenstone Lake radiocarbon date while version 3.0.2 of the Balco et al. (2008) calculator produces ages that are less consistent with the Greenstone Lake radiocarbon date. In particular, CRONUScalc 2.0 and CREp produce site-average surface-exposure ages of 16.0–15.5 ka for our Tuolumne Meadows sites while the Balco et al. (2008) calculator produces site-average surface-exposure ages of 15.2–14.7 ka. Thus, within the results provided by the Balco et al. (2008) calculator, even the oldest site-average surface-exposure age has a nominal age that is younger than the lower 99 % confidence bounds (15.4 ka) on the Greenstone Lake radiocarbon date.

Thus, even if there were legitimate arguments for glacier (or atmospheric) dynamics influencing the timing of deglaciation along the ~600-km length of the Sierra Nevada, none of these arguments would alter the fact that the CRONUScalc 2.0 (Marrero et al., 2016) and CREp (Martin et al., 2017) calculators are more likely producing more accurate surface-exposure ages for the $^{10}$Be samples we report in this manuscript than the "Version 3" calculator (Balco et al., 2008). Assessing the accuracy of the cosmogenic surface-exposure calculators is our driving motivation for reviewing the radiocarbon constraints on the timing of the "Tioga 4 deglaciation."

All this said, we will revise the sentence to read (new text in blue): "…(Phillips, 2016) as the best available estimate for the timing of deglaciation in the high-elevation lake basins…"

| RC 46 | Screenshot from the preprint… |
| --- | --- |
| | 256     both the San Joaquin and Bishop Creek Glaciers. Based on these dates, Phillips et al. (2009) |
| | 257     concluded that the Tioga 4 deglaciation lasted <500-1000 years. |
| | Comment: |
| | Could 2.2 overall be reduced by about 30%? I think it is a bit too much details on stuff that has been published already and which could be summarised further. |

| AR 46 | We are not comfortable reducing this section by as much as 30 % because it lays out essential background information on the deglaciation that followed the culmination of the Tioga 4 readvance – but we will reduce it by at least 10 %. |
| --- | --- |

| | |
|---|---|
| **RC 47** | Screenshot from the preprint…

 258   3.   **Materials and methods**

 Comment:
 could part of this be migrated to the supplementary? |
| **AR 47** | Yes. |
| **RC 48** | Screenshot from the preprint…

 326   anomalously old (ca. 22 ka) and we preemptively excluded this sample from our reanalysis. We

 327   also exclude the twenty-three bedrock $^{10}$Be samples reported by Dühnforth et al. (2010) from

 328   Tuolumne Meadows and Lyell Canyon because uncertainties regarding the degree of inheritance

 329   and snow shielding. These samples were collected from small bedrock protrusions (e.g. roche

 330   moutonnées and the heights of these protrusions were not reported. Additionally, we exclude the

 Comment:
 some of these could be rather tall, do you have their coordinates and could check on Lidar? Also given that some of your samples are only 35 cm. Would these additional ages disrupt your interpretation considerably? It sounds like a lot of potentially valuable samples to exclude, but I am sure you have good reasons |
| **AR 48** | We excluded the Dühnforth et al. (2010) samples from our reanalysis for two reasons: (1) uncertainty regarding their degree of inheritance and (2) uncertainty regarding the height of these roche moutonnées and thus uncertainty in the appropriate snow-shielding correction. While lidar coverage of these samples has become available since we first drafted this text, thus substantially reducing the uncertainty around roche moutonnée height, and thus about the snow-shielding correction factors, our concerns about inheritance in these samples remain. |
| **RC 49** | Screenshot from the preprint…

 333   3.4.   *Snow-shielding correction*

 334   Snow-shielding corrections for the thirty-one new $^{10}$Be samples from Yosemite National

 Comment:
 this is always a struggle but I appreciate the effort. I think it would be fair mentioning the high level of uncertainty with any such calculation, given that who knows really how much snow there was for millennia prior to the present.. How much does this correction affect your ages? |
| **AR 49** | As we acknowledge in AR 5, extrapolating snow conditions from the last ~50-100 years to the last *ca.* 16,000 years is indeed a large extrapolation (~160-320x). However, a snow correction is clearly required (see, for example, Fig. 2-5 in the |

lead author's dissertation) and extrapolating from recent snow observations to the distant past is the most commonly used approach to solving this problem.

Moreover, recent snow conditions in the Sierra Nevada (*i.e.*, the last ~50-100 years) seem to be reasonably representative of average snow conditions there over the last *ca.* 16,000 years. For example, if *no* snow correction is applied to the ages, then variations in boulder height explain 27-29% of the age variations between the samples (depending on choice of calculator (Fig. 2-5 in Becker's dissertation). However, if we (1) apply a correction for seasonal snow shielding and (2) assume it is equal to historical snow conditions in the Sierra Nevada (since 1945, more or less), then boulder height only explains 0.2-0.4% of the age variation.

And, unlike all other surface-exposure dating studies that we are aware that use historical snow conditions to correct for snow conditions over the postglacial interval, we propagate the modern uncertainties in snow-water equivalent (SWE) vs. elevation (Fig. S21 in the manuscript) and in average snowpack density vs. SWE (Fig. S22c) into our preferred set of cosmogenic surface-exposure age uncertainties (*i.e.*, those from the CRONUScalc 2.0 (Marrero et al., 2016) calculator).

Thus, compared with most previous cosmogenic surface-exposure dating studies that use historical snow conditions to correct for snow shielding over the "post-glacial interval" (however long that might be), the approach we describe in this manuscript includes more sources of uncertainty and thus is more likely to be representative of that longer interval (the past *ca.* 16,000 years in this case) than the reported age uncertainties in those other publications.

Finally, with regards to the strength of the snow-shielding correction, it varies with sample height and elevation and ranges from 0.2 % to 11.9 % for the samples we report in this manuscript.

| RC 50 | Screenshot from the preprint… |
| --- | --- |
| | 466    location is the bottom of Lyell Canyon (an intermediate location, but the site with the greatest ice |
| | 467    thickness). In the Version 3 results, the sample (TM14-15) from lower Tuolumne Meadows has |
| | Comment:
 based on? |
| AR 50 | Wahrhaftig et al. (2019), we will add this citation to the sentence, thank you. |
| RC 51 | Screenshot from the preprint… |

| | |
|---|---|
| | 624    This manuscript's goal is to place the Sierra Nevada's final deglaciation in a robust
625    regional and global context. That narrative, however, depends upon the cosmogenic dates and the

Comment:
is it fair to talk about the Sierra as a whole, given its size and the number of sites investigated? A statement on representativeness would be useful |
| **AR 51** | We will revise this sentence to read (new text in blue font): "…to place the central Sierra Nevada's final deglaciation…" |
| **RC 52** | Screenshot from the preprint…

741    mean ages range from 16.0–15.5 ka (Table 2). For the three remaining locations, larger |
| **AR 52** | Thank you for catching this issue; we will delete the word "from". |
| **RC 53** | Screenshot from the preprint…

780    Accordingly, with the goal of streamlining the Discussion going forward, we will
781    henceforth exclusively refer to the CRONUScalc 2.0 ages, which are indistinguishable from the
782    CREp ages. This interpretive decision results in the following deglacial narrative: the innermost

Comment:
the pages above this point, in this section, could be migrated to the supplementary. |
| **AR 53** | We will rewrite Section 5.2.2. *On the accuracy of the cosmogenic surface-exposure calculators* to read as follows:

"Accepting the bulk-organic radiocarbon dates as accurate minimum limits on the timing of the Tioga 4 deglaciation suggests that the CRONUScalc and CREp dates are probably more accurate than the Version 3 calculator's dates for the $^{10}$Be samples reported here. Our argument for this interpretation is provided in the online supplement (Sect. S.8.). Accordingly, with the goal of streamlining the Discussion going forward, we will henceforth exclusively refer to the CRONUScalc 2.0 ages (which are indistinguishable from the CREp ages). This interpretive decision results in the following deglacial narrative: the innermost…"

Note: This revised Section 5.2.2. does three things:
1. it retains the opening sentence of Section 5.2.2. from the submitted manuscript,
2. it adds a new sentence referring readers to the supplement, and then
3. it moves the text in lines 736–779 of the submitted manuscript to the supplement (as suggested by the reviewer). |

| | |
|---|---|
| | Thus, we fully accept the reviewer's suggestion to move the lines above 780 in this section to the supplement, with the solo exception of the section's opening sentence. The text that is being moved to the supplement will become section S8 of the supplement – and sections S8 through S14 of the submitted supplement will become sections S9 through S15 of the revised supplement. |
| RC 54 | Screenshot from the preprint…  Comment: in general or for this specific time interval? I would be careful with generalisations such as this one |
| AR 54 | This is a general statement about how deglaciation proceeds in a mountainous landscape when the ELA rapidly rises to such a height that only cirque glaciers (if that) can be sustained and follows the work of those cited above, especially the summarizing work of Goldthwait and Mickelson (1982). While we agree that care is required when making general statements, this is a general statement that is well supported by the literature (i.e., the citations in these lines of the preprint). We will clarify this sentence by modifying it to read as follows: "This narrative is consistent with geomorphic interpretations of rapid deglaciation and ice thinning in the Sierra Nevada at the end of the last glaciation (Clark, 1976; Clark and Clark, 1995) and with observations from Alaska (Cushing, 1891; Reid, 1896; Field, 1947; Mickelson, 1971; Syverson, 1995), and interpretations (Goldthwait, 1938; Lowell, 1985; Lowell et al., 1990) and dating (Bierman et al., 2015; Davis et al., 2015; Koester et al., 2017, 2021; Corbett et al., 2019; Drebber et al., 2023; Halsted et al., 2024) from New England and adjacent regions (Barth et al., 2019) for how deglaciation proceeds in a mountainous landscapes when the ELA rapidly rises and glaciers become disconnected from their accumulation zones (Goldthwait and Mickelson, 1982; Vacco et al., 2010)." |
| RC 55 | Screenshot from the preprint… |

837     the central Sierra Nevada from innermost moraine stabilization to the melting of the last

838     remanent ice masses is 1.0 kyr, with a 95 % confidence range of 0.2–3.0 kyr (Fig. 7c).

Comment:

how fast is this, really? Perhaps the narrative should focus on putting this in relative terms. Faster than (what occurred between Tioga 3 and 4? or Tioga 2 and 3? Or elsewhere in similar settings for this same time period? etc.), rather than just fast or abrupt.

How much topography and elevation was available above the Tioga 4 moraines?

And finally, would it be possible to estimate a retreat rate? For many this parameter would be easy to put into context, as present-day glaciologists often talk in terms of retreat rates

| AR 55 | The reviewer raises five to six questions in this comment:

**How fast, really, is a ca. 1.0 kyr deglaciation?**
   1.  It was fast within the context of geomorphology. Meaning, there is no record of episodic ice advances or stillstands after the Tioga 4 retreat started at ca. 16.4 ka. The Tioga 4 moraine is the innermost end moraine in the Sierra Nevada. Before then, interannual climate variability (weather) and/or genuine climate change was able to shift the glaciers of the Sierra Nevada into positive mass balance and driven them forward to generate a moraine record. After the start of the Tioga 4 deglaciation at ca. 16.4 ka, there is no geomorphic record of that happening.

**How fast was the "Tioga 4 deglaciation" compared with the ice retreat between the Tioga 2 and Tioga 3 ice advances?**
   2.  Quantitatively comparing the rate of ice retreat following the Tioga 4 with the rate of ice retreat between the Tioga 2 (LGM) glacial advance and the Tioga 3 readvance is challenging because we don't know how far glaciers receded during that interval, only that the minimum glacier extent in the Sierra Nevada between the Tioga 2 and Tioga 3 glacial advances must have been <90 % LGM-glacier length because the Tioga 3 readvance reached ~90 % LGM-glacier length.

**How fast was the "Tioga 4 deglaciation" compared with the ice retreat between the Tioga 3 and Tioga 4 ice advances?**
   3.  Likewise, quantitatively comparing the rate of ice retreat during the "Tioga 4 deglaciation" with the rate of ice retreat between the Tioga 3 and Tioga 4 readvances is also hard because we don't know how far back the ice shrank |
|---|---|

during this interval, only that it retreated from ~90 % LGM-glacier length to less than about 60 % LGM-glacier length.

**How fast was the "Tioga 4 deglaciation" compared with ice retreat rates elsewhere on the planet in mountainous settings at ca. 16 ka?**

4.  In Tuolumne Meadows and Lyell Canyon, the ice margin melted back at an average rate of 20 m yr$^{-1}$ from the lower Tuolumne Meadows sampling site to the bottom of Lyell Canyon sampling site.

    In Mono Creek Canyon, the ice margin melted back at an average rate of 13 m yr$^{-1}$ from the innermost Lake Edison moraine to the lower Mono Canyon Creek sample and then at an average rate of 31 m yr$^{-1}$ from there to the cirques in Pioneer Basin, which is consistent with a downwasting ice mass that is becoming more out of equilibrium with climate as it shrinks. These retreat rates roughly cover the ca. 16.4 ka to ca. 15.6 ka intervals.

    In comparison, in the Front Range of Colorado, Dühnforth and Anderson (2011) report ages and sampling positions from Green Lakes Valley that produce an average retreat rate of 0.8 m yr$^{-1}$ over the *much* longer interval from ca. 24 ka to ca. 12 ka; in the Upper Arkansas River basin of Colorado, Young et al. (2011) report ages and sampling positions that produce an average retreat rate of 1.7 m yr$^{-1}$ over the ca. 19 ka to ca. 14 ka interval; also in the Upper Arkansas River basin of Colorado, Schweinsberg et al. (2020) report ages and sampling positions that indicate the average ice-margin retreat rate between ca. 20.6 ka and ca. 15.6 ka was 1.1 m yr$^{-1}$ and then was 17 m yr$^{-1}$ between ca. 15.6 ka and ca. 14.1 ka.

    In contrast, ice-marginal recession rates along the southern and southeastern margin of the Laurentide Ice Sheet were faster than those we document in the central Sierra Nevada. Lowell et al. (2021) documented near-constant retreat rates of between ~40 m yr$^{-1}$ and ~60 m yr$^{-1}$ over the ca. 19 ka to ca. 10 ka interval along a 500-km-long SW-NE transect to the west of Lake Superior. Ridge et al. (2012) documented retreat rates of ~50-100 m yr$^{-1}$ (18.2–17.4 ka), ~30-40 m yr$^{-1}$ (17.4–16.2 ka), ~80–90 m yr$^{-1}$ (16.0–14.8 ka), and ~300 m yr$^{-1}$ (14.8–14.2 ka).

    In summary, the ice-marginal recession rates in the Sierra Nevada during the ca. 16 ka to ca. 15 ka "Tioga 4 deglaciation" were relatively fast compared with other ice-marginal retreat rates in the U.S. West over that interval and generally slower than LIS ice-margin retreat rates.

    As mentioned in AR 7, we will add the average rate of ice-margin recession in Tuolumne Meadows and Lyell Canyon (20 m yr$^{-1}$) and Mono Creek Canyon (first 13 m yr$^{-1}$, then 31 m yr$^{-1}$) to the manuscript. We will also

| | briefly compare these rates with rates of ice margin recession elsewhere in the American West at this time. |
|---|---|
| | **How much topography and elevation was available above the Tioga 4 moraines?**
5. The Tioga 4 moraine in Lake Edison is 1000 m below the cirque floors in Pioneer Basin and the Tioga 4 moraine in the Middle Fork of Bishop Creek Canyon is also 1000 m below Piute Pass, which is the connection between Humphreys Basin and the North Fork of Bishop Creek Canyon. During the Tioga 4 glacial stage (and during the earlier LGM (Tioga 2) and Tioga 3 stages), ice flowed from Humphreys Basin down the North Fork of Bishop Creek Canyon and into the Middle Fork of Bishop Creek Canyon.

**How many meters / year were the Tioga 4 glaciers retreating during their deglaciation?**
6. Please see our answer to question #4 above. |
| RC 56 | Screenshot from the preprint…

850    final deglaciation was a geomorphically abrupt event. For instance, the bottom of Lyell Canyon

851    sampling location has the youngest mean surface-exposure age across all the sampling locations

852    reported here (Fig. 3c; Table 2) – and a trimline-based ice-thickness reconstruction Walırhaftig

853    et al., 2019) indicates the bottom of Lyell Canyon hosted an ice thickness greater than any other

854    location we sampled. Tioga 4 ice thicknesses over the bottom of Lyell Canyon sampling site

Comment:
For Tioga4? With chronological constraints? |
| AR 56 | For the LGM (Tioga 2), with the chronological constraint being derived from the large relative-age difference between the landforms of the last glaciation (MIS 2) and the landforms of the penultimate glaciation (MIS 6). Given a typical profile of ice-thickness with distance from the ice margin (e.g., Anderson and Anderson, 2010, Fig. 8.25), the bottom of Lyell Canyon would also have had the thickest Tioga 4 ice. |
| RC 57 | Screenshot from the preprint…

861    organic accumulation in Greenstone Lake. The observation that the locations with the thickest

862    ice were the last to deglaciate, regardless of their distance from the cirque headwalls, indicates

863    that the deglaciation was driven by a relatively sudden and large rise in the ELA.

Comment:
I am not sure I follow this argument. What would the alternative option be? |

| | |
|---|---|
| **AR 57** | The alternative is a slow rise in the ELA which enables the glaciers of the central Sierra Nevada to wax and wane as they generally retreat, leaving a rich end-moraine record behind that terminates in the cirques. Although the youngest Pleistocene moraines are in the cirques or just beyond them (the Recess Peak moraines), there are no known end moraines in the Sierra Nevada between the Tioga 4 moraines and the Recess Peak moraines, suggest a large and rapid rise in the ELA over at the start of that time interval (i.e., just after the culmination of the Tioga 4 readvance). |
| **RC 58** | Screenshot from the preprint…

 https://doi.org/10.5194/egusphere-2025-1370
Preprint. Discussion started: 3 April 2025
© Author(s) 2025. CC BY 4.0 License.

 Comment:
weren't these described earlier? If so, avoid repetition |
| **AR 58** | This comment was apparently misapplied to the manuscript's header information. Our assumption is that reviewer #1 intended to apply it to the following passage instead (highlighted in yellow by us):

 867  wintertime drying of ≥35 %, or some combination thereof. The temperature-change

 868  reconstruction is based on the interpretation that the deglaciation was driven by ≥600 m rise in

 51

 https://doi.org/10.5194/egusphere-2025-1370
Preprint. Discussion started: 3 April 2025
© Author(s) 2025. CC BY 4.0 License.
[Figure]

 869  the ELA (as reviewed in Sect. 2.2.1 ) and the observation that modern surface-temperature lapse
[Figure]

 If we're correct in thinking that Reviewer #1 intended to highlight what we've highlighted in yellow, then our response is as follows:
    • No, we have not previously described our methods for converting the ELA rise into estimates of the temperature change or precipitation change |

| | |
|---|---|
| | • responsible for it and – as it happens – Reviewer #2 asked for more information about these methods (RC 88).

• Previously, in section 2.2.1, we describe how Phillips (2016) reconstructed the ELA responsible for the Tioga 4 readvance (1 sentence) and how much the ELA must have risen immediately following the Tioga 4 readvance in order to essentially deglaciate the central Sierra Nevada (1 sentence).

• In light of RC 88 (by Reviewer #2), this section will be rewritten as described in AR 88. |
| **RC 59** | Screenshot from the preprint…

874    with the estimated lapse rate suggests the Sierra Nevada's final deglaciation was driven by a

875    summertime warming of $\geq 2$ °C, assuming no change in winter precipitation (Leonard et al.,

876    2023).

Comment:
the problem with attempting to define end terms with two variables is that we do not know if both changed in the same direction, glacier mass balance wise, or not. Do other proxies tell us that P decreased during this time interval? I guess this might come in the next section |
| **AR 59** | Defining limiting, end-member changes in temperature and precipitation does assume that the other variable did not change in such a way as to require a larger change in the variable in question. For example, if winter snowfall increased during the "Tioga 4 deglaciation" (i.e., during the deglaciation that followed the culmination of the Tioga 4 readvance) then a larger summer-temperature rise would be required to deglaciation the range. However, because we only know that the ELA rise responsible for the deglaciation was *at least* 600 m, we already know the temperature rise was *at least* 2 °C, even if we assume precipitation also increased (rather than remaining constant).

For clarity, we will revise the sentence on lines 873–876 to read as follows (with new text in blue):
"Combining the ELA change ($\geq 600$ m) with the estimated lapse rate (3–4 °C km$^{-1}$) suggests the central Sierra Nevada's final deglaciation was driven by a summertime warming of $\geq 2$ °C, assuming no decrease in winter precipitation. |
| **RC 60** | Screenshot from the preprint… |

993 Otto-Bliesner et al., 2006; Wong et al., 2016; Jones et al., 2018). Also, as the LIS thins, it

994 becomes a less formidable obstacle to atmospheric circulation, which enables the polar jet stream

995 – formerly split by the LIS (Fig. 9a), with a weaker branch passing to the north of the ice sheet

996 and a stronger branch passing to the south (Kutzbach and Wright Jr., 1985; Manabe and Broccoli,

997 1985; Kutzbach and Guetter, 1986; Bromwich et al., 2004; Löfverström et al., 2014; Lora et al.,

998 2016; Wang et al., 2024) – to reunify (Fig. 9b). We interpret that these changes in the Aleutian

Comment:
this aspect is pretty central in your narration of events, but I am unclear what evidence you have to justify it. Is it really necessary to explain your results? Is it one of many hypotheses? Could it be tested at all, or is it just a guess?

| AR 60 | Generations of numerical models of varying levels of complexity have reproduced a split-polar jet stream over North America during the LGM as a result of the LIS, as cited in this sentence, with the polar jet reunifying and shifting northward, away from the Sierra Nevada, in response to LIS thinning. Thus, it is reasonable to look for evidence of that atmospheric transformation in the landscape of the Sierra Nevada.

That said, it is not necessary to explain our results and there other hypotheses, with a principal alternative being that it was the subtropical jet stream – not the polar jet stream – that brought moisture to the Sierra Nevada during the last glaciation and that its northward shift during the last glaciation was driven by changes in sea-ice extent in the North Atlantic (Chiang and Bitz, 2005; Chiang et al., 2014; Phillips, 2017; McGee et al., 2018).

Distinguishing which jet stream was most responsible for moisture delivery to the Sierra Nevada during the last glaciation and deglaciation is a challenge, and cannot be unequivocally answered by the range's glacial deposits. |
|---|---|
| RC 61 | Screenshot from the preprint…

1029      In summary, we infer Heinrich Event 1 sufficiently thinned the LIS at ca. 16.2 ka to

1030      trigger an atmospheric reorganization (Fig. 9). That atmospheric reorganization brought drier

Comment (line 1029):
wouldn't this paragraph best fit in the conclusions? |
| AR 61 | Yes, we agree and will make the change. |
| RC 62 | Screenshot from the preprint… |

| | |
|---|---|
| | 1030     trigger an atmospheric reorganization (Fig. 9). That atmospheric reorganization brought drier

1031     winters and warmer summers to what is now the southwestern United States. In response, the

**Comment:**
if this is based on the proxies described above, I believe you cannot really infer a seasonality of the climatic signals. |
| AR 62 | This sentence in the preprint is poorly written. We are inferring these possible changes from the deglaciation of the central Sierra Nevada.  Our argument that (a) winters were drier, or (b) summers were warmer, or (c) winters were drier *and* summers were warmer comes from the sensitivity of glaciers to winter precipitation and summer temperatures (e.g., Oerlemans, 2005).

Glaciers are relatively insensitive to summer precipitation changes and winter temperature changes. Thus, the glacial geomorphic record provides little-to-no insight into how summer precipitation and winter temperatures might have changed. |
| RC 63 | Screenshot from the preprint…

1031     winters and warmer summers to what is now the southwestern United States. In response, the

1032     central and southern Sierra Nevada essentially deglaciated and formerly expansive lakes in

1033     California, Arizona, New Mexico, and Utah desiccated (McGee et al., 2018). Offshore western

**Comment:**
non glacier fed |
| AR 63 | We will move the paragraph on lines 1029–1038 of the preprint to the conclusions (as suggested in RC 61) and revise it to read as follows (new text in blue):

"In summary, we infer Heinrich Event 1 sufficiently thinned the LIS at ca. 16.2 ka to trigger an atmospheric reorganization (Fig. 9). Based on the sensitivity of glaciers to winter precipitation and summer temperatures (e.g., Oerlemans, 2005), we infer that this reorganization brought drier winters, warmer summers, or both to what is now the southwestern United States. The central Sierra Nevada essentially deglaciated in response to this change, and formerly expansive lakes in California, Arizona, New Mexico, and Utah abandoned high-level shorelines (Waters, 1989; Allen and Anderson, 2000; Bacon et al., 2006; Munroe and Laabs, 2013; McGee et al., 2018). Numerical modeling indicates LIS thinning weakens the Aleutian Low (e.g., Otto-Bliesner et al., 2006; Yanase and Abe-Ouchi, 2010), and we infer weakening of the Aleutian Low reduced the influx of polar water into the Santa Barbara Basin, causing SSTs there to warm (Hendy and Kennett, 2000; Hendy et al., 2002; Hendy, 2010). Likewise, a concomitant northward migration of the northeastern Pacific's subtropical high reoriented winds over the Gulf of California |

and greatly reduced upwelling there, allowing subtropical water to enter the Gulf and warm SSTs there (Ganeshram and Pedersen, 1998; McClymont et al., 2012)."

**References Cited in this Response:**

Allen, B. D. and Anderson, R. Y.: A continuous, high-resolution record of late Pleistocene climate variability from the Estancia basin, New Mexico, Geological Society of America Bulletin, 112, 1444–1458, 2000.

Anderson, L. S., Roe, G. H., and Anderson, R. S.: The effects of interannual climate variability on the moraine record, Geology, 42, 55–58, 2014.

Anderson, R. S. and Anderson, S. P.: Geomorphology: The Mechanics and Chemistry of Landscapes, Cambridge University Press, 655 pp., 2010.

Andrews, J. T. and Voelker, A. H.: "Heinrich events"(& sediments): A history of terminology and recommendations for future usage, Quaternary Science Reviews, 187, 31–40, 2018.

Bacon, S. N., Burke, R. M., Pezzopane, S. K., and Jayko, A. S.: Last glacial maximum and Holocene lake levels of Owens Lake, eastern California, USA, Quaternary Science Reviews, 25, 1264–1282, 2006.

Balco, G., Stone, J. O., Lifton, N. A., and Dunai, T. J.: A complete and easily accessible means of calculating surface exposure ages or erosion rates from $^{10}$Be and $^{26}$Al measurements, Quaternary Geochronology, 3, 174–195, 2008.

Barth, A. M., Marcott, S. A., Licciardi, J. M., and Shakun, J. D.: Deglacial Thinning of the Laurentide Ice Sheet in the Adirondack Mountains, New York, USA, Revealed by $^{36}$Cl Exposure Dating, Paleoceanography and Paleoclimatology, 34, 946–953, https://doi.org/10.1029/2018PA003477, 2019.

Bartlein, P. J., Anderson, K. H., Anderson, P. M., Edwards, M. E., Mock, C. J., Thompson, R. S., Webb, R. S., Webb III, T., and Whitlock, C.: Paleoclimate simulations for North America over the past 21,000 years: features of the simulated climate and comparisons with paleoenvironmental data, Quaternary science reviews, 17, 549–585, 1998.

Bateman, P. C., Pakiser, L. C., and Kane, M. F.: Geology and tungsten mineralization of the Bishop district, California, with a section on gravity study of Owens Valley and a section on seismic profile, US Geological Survey, 1965.

Bierman, P. R., Davis, P. T., Corbett, L. B., Lifton, N. A., and Finkel, R. C.: Cold-based Laurentide ice covered New England's highest summits during the Last Glacial Maximum, Geology, 43, 1059–1062, https://doi.org/10.1130/G37225.1, 2015.

Bromwich, D. H., Toracinta, E. R., Wei, H., Oglesby, R. J., Fastook, J. L., and Hughes, T. J.: Polar MM5 simulations of the winter climate of the Laurentide Ice Sheet at the LGM, Journal of Climate, 17, 3415–3433, 2004.

Clark, D., Gillespie, A. R., Clark, M., and Burke, B.: Mountain glaciations of the Sierra Nevada, in: Quaternary Geology of the United States, Desert Research Institute, Reno, NV, 287–311, 2003.

Clark, D. H. and Clark, M. M.: New evidence of late-Wisconsin deglaciation in the Sierra Nevada, California, refutes the Hilgard Glaciation, in: Geological Society of America, Abstracts with Programs, 1995.

Clark, D. H. and Gillespie, A. R.: Timing and significance of Late-Glacial and Holocene cirque glaciation in the Sierra Nevada, California, Quaternary International, 38, 21–38, 1997.

Clark, D. H., Bierman, P. R., and Larsen, P.: Improving in situ cosmogenic chronometers, Quaternary Research, 44, 367–377, 1995.

Clark, M.: Evidence for rapid destruction of Latest Pleistocene glaciers of the Sierra Nevada, California, in: Geological Society of America, Abstracts with Programs, 361–362, 1976.

Corbett, L. B., Bierman, P. R., Wright, S. F., Shakun, J. D., Davis, P. T., Goehring, B. M., Halsted, C. T., Koester, A. J., Caffee, M. W., and Zimmerman, S. R.: Analysis of multiple cosmogenic nuclides constrains Laurentide Ice Sheet history and process on Mt. Mansfield, Vermont's highest peak, Quaternary Science Reviews, 205, 234–246, https://doi.org/10.1016/j.quascirev.2018.12.014, 2019.

Cushing, H. P.: Notes on the Muir Glacier region, and its geology, The American Geologist, 8, 207–230, 1891.

Davis, P. T., Bierman, P. R., Corbett, L. B., and Finkel, R. C.: Cosmogenic exposure age evidence for rapid Laurentide deglaciation of the Katahdin area, west-central Maine, USA, 16 to 15 ka, Quaternary Science Reviews, 116, 95–105, https://doi.org/10.1016/j.quascirev.2015.03.021, 2015.

Drebber, J. S., Halsted, C. T., Corbett, L. B., Bierman, P. R., and Caffee, M. W.: In Situ Cosmogenic [10]Be Dating of Laurentide Ice Sheet Retreat from Central New England, USA, Geosciences, 13, 213, https://doi.org/10.3390/geosciences13070213, 2023.

Dühnforth, M. and Anderson, R. S.: Reconstructing the Glacial History of Green Lakes Valley, North Boulder Creek, Colorado Front Range, Arctic, Antarctic, and Alpine Research, 43, 527–542, https://doi.org/10.1657/1938-4246-43.4.527, 2011.

Dühnforth, M., Anderson, R. S., Ward, D., and Stock, G. M.: Bedrock fracture control of glacial erosion processes and rates, Geology, 38, 423–426, 2010.

Evans, J. M., Stone, J. O. H., Fifield, L. K., and Cresswell, R. G.: Cosmogenic chlorine-36 production in K-feldspar, Nuclear instruments and methods in physics research section B: beam interactions with materials and atoms, 123, 334–340, 1997.

Field, W. O. Jr.: Glacier recession in Muir Inlet, Glacier Bay, Alaska, Geographical Review, 37, 349–399, 1947.

Foster Flint, R.: Glacial and Quaternary geology, 1971.

Ganeshram, R. S. and Pedersen, T. F.: Glacial-interglacial variability in upwelling and bioproductivity off NW Mexico: Implications for Quaternary paleoclimate, Paleoceanography, 13, 634–645, 1998.

Gillespie, A. R. and Clark, D. H.: Glaciations of the Sierra Nevada, California, USA, in: Developments in quaternary sciences, vol. 15, Elsevier, 447–462, 2011.

Goldthwait, J. W.: The uncovering of New Hampshire by the last ice sheet, American Journal of Science, 236, 345–372, 1938.

Goldthwait, R. P. and Mickelson, D. M.: Glacier Bay, a model for the deglaciation of the White Mountains in New Hampshire, Ohio State University, Institute of Polar Studies, 1982.

Gosse, J. C. and Phillips, F. M.: Terrestrial in situ cosmogenic nuclides: theory and application, Quaternary Science Reviews, 20, 1475–1560, 2001.

Halsted, C. T., Bierman, P. R., Shakun, J. D., Davis, P. T., Corbett, L. B., Drebber, J. S., and Ridge, J. C.: A critical re-analysis of constraints on the timing and rate of Laurentide Ice Sheet recession in the northeastern United States, Journal of Quaternary Science, 39, 54–69, https://doi.org/10.1002/jqs.3563, 2024.

He, F.: Simulating transient climate evolution of the last deglaciation with CCSM 3, 2011.

Hendy, I. L.: The paleoclimatic response of the Southern California Margin to the rapid climate change of the last 60 ka: A regional overview, Quaternary International, 215, 62–73, 2010.

Hendy, I. L. and Kennett, J. P.: Dansgaard-Oeschger cycles and the California Current System: planktonic foraminiferal response to rapid climate change in Santa Barbara Basin, Ocean Drilling Program hole 893A, Paleoceanography, 15, 30–42, 2000.

[revised manuscript text omitted]

Reid, H. F.: Glacier Bay and its glaciers, USGS, 1896.

Ridge, J. C., Balco, G., Bayless, R. L., Beck, C. C., Carter, L. B., Dean, J. L., Voytek, E. B., and Wei, J. H.: The new North American Varve Chronology: A precise record of southeastern Laurentide Ice Sheet deglaciation and climate, 18.2-12.5 kyr BP, and correlations with Greenland ice core records, American Journal of Science, 312, 685–722, 2012.

Rood, D. H., Burbank, D. W., and Finkel, R. C.: Chronology of glaciations in the Sierra Nevada, California, from $^{10}$Be surface exposure dating, Quaternary Science Reviews, 30, 646–661, 2011.

Schweinsberg, A. D., Briner, J. P., Licciardi, J. M., Shroba, R. R., and Leonard, E. M.: Cosmogenic 10Be exposure dating of Bull Lake and Pinedale moraine sequences in the upper Arkansas River valley, Colorado Rocky Mountains, USA, Quaternary Research, 97, 125–139, 2020.

Syverson, K. M.: The ability of ice-flow indicators to record complex, historic deglaciation events, Burroughs Glacier, Alaska, Boreas, 24, 232–244, 1995.

Vacco, D. A., Alley, R. B., Pollard, D., and Reusch, D. B.: Numerical modeling of valley glacier stagnation as a paleoclimatic indicator, Quaternary Research, 73, 403–409, 2010.

Wahrhaftig, C., Stock, G. M., McCracken, R. G., Sasnett, P., and Cyr, A. J.: Extent of the Last Glacial Maximum (Tioga) glaciation in Yosemite National Park and vicinity, California, US Geological Survey Scientific Investigations Series Map, 3414, 2019.

Waters, M. R.: Late Quaternary lacustrine history and paleoclimate significance of pluvial Lake Cochise, southeastern Arizona, Quaternary Research, 32, 1–11, 1989.

Yanase, W. and Abe-Ouchi, A.: A numerical study on the atmospheric circulation over the midlatitude North Pacific during the last glacial maximum, Journal of Climate, 23, 135–151, 2010.

Young, N. E., Briner, J. P., Leonard, E. M., Licciardi, J. M., and Lee, K.: Assessing climatic and nonclimatic forcing of Pinedale glaciation and deglaciation in the western United States, Geology, 39, 171–174, 2011.

---

## Author Comment (AC2)

**Response to Reviewer 2**

RC = Reviewer Comment

AR = Author Response

| RC 64 | Dear Editor and Authors: |
|---|---|
| | I appreciate the opportunity to read this Discussion paper that provides new cosmogenic nuclide ages on post-LGM moraines in the Sierra Nevada and attempts to place the glacial history of this area in a broader regional and global context. The new glacial chronologies will be of great interest to those studying the history of this region, and are an exciting addition to the multi-archive, multi-proxy body of work on western North American paleoclimate. |
| | I find the new ages to be interesting and well-presented. My primary comments on this manuscript are centered on the treatment of regional proxy information and the interpretations about the broader expression of Heinrich Stadial 1, so this is what I will discuss for the remainder of this comment. I find that the discussion of the temporal evolution of events in the North Atlantic region and the western US is a bit muddled in the discussion. The previously published proxy information that the authors present here could be treated more carefully and clearly. |
| **AR 64** | Dear Reviewer #2: |
| | Thank you for investing your time in this paper and for commenting upon its strengths and opportunities for improvement. We are grateful to receive these comments. |
| **RC 65** | The authors emphasize a very narrow window for "Heinrich Event 1" which they define from a Spanish stalagmite record from Ostolo Cave. This is shown in the graphical abstract and in Figure 8 with yellow shading and labeled "HE1". The Ostolo Cave record suggests a change in the $d^{18}O$ of the moisture source as well as temperature in an excursion contemporaneous with IRD in the Bay of Biscay (Eynaud et al., 2012) and the change in seawater isotopic composition (Voelker et al., 2009). The Ostolo Cave paper by Bernall-Wormull which you cite, interprets this negative excursion in $d^{18}O$ as follows: |
| | "An exceptional light $\delta^{18}O_{speleo}$ excursion centered at 16.2–16.0 kyr B.P. is interpreted to reflect the major phase of HE1 iceberg melting reaching the Iberian Peninsula, which drastically changed the $\delta^{18}O$ composition of regional precipitation." |
| | While this is a precisely dated record, it is a regional expression of the Heinrich Event in Spain and should be taken in context with other North Atlantic records of the event – which Bernall-Wormhull also show in their figure 4. I suggest you also |

| | compare to the records of IRD and foraminifera d$^{18}$O which suggest a broader *peak* for meltwater release between ~16.5 and 16 ka – rather than shading this sharp speleothem δ$^{18}$O excursion as the full expression of the Heinrich Event in your Figure 8 and associated discussion in the text. The record of 231Pa/230Th in North Atlantic sediments suggests that AMOC slowdown began closer to 18 ka, again reaching a minimum close to 16 ka (McManus et al., 2004). Similar timing of freshwater release to the North Atlantic is used in the TRACE and iTRACE transient climate model simulations (He et al., 2011; He et al., 2021). A summary of Heinrich Stadial 1 model results and proxy records for the western US is provided in Oster et al., 2023. This supports that the window that you have chosen for Heinrich Event 1 is too narrow. |
|---|---|
| **AR 65** | For clarity, we differentiate between Heinrich Events (such as Heinrich Event 1; HE1) and Heinrich Stadials (such as Heinrich Stadial 1; HS1). As described by Andrews and Voelker (2018) and Heath et al. (2018), Heinrich Stadials are time periods of relatively cold sea-surface temperatures in the North Atlantic while Heinrich Events are the disintegration of the Laurentide Ice Sheet's Hudson Strait ice shelf and the drainage of the land-based ice behind it into the North Atlantic (e.g., Álvarez-Solas et al., 2011; Marcott et al., 2011). Heinrich Events are recorded by layers of ice-rafted detritus (IRD) in the North Atlantic ("Heinrich Layers") with diagnostic characteristics that link those particular IRD layers with the Hudson Strait ice stream (Andrews and Voelker, 2018). Heinrich Stadials typically last a few thousand years while Heinrich Events durations are typically <1 kyr (Hemming, 2004; Andrews and Voelker, 2018) and potentially only ~200-300 years (Francois and Bacon, 1994; Dowdeswell et al., 1995; Thomson et al., 1995; Pérez-Mejías et al., 2021). Heinrich Stadial 1 began at 19.3 ka and ended at 15.3 ka (Heath et al., 2018). The causal link between Heinrich Stadials and Heinrich Events has been challenged throughout the recent literature (e.g., Álvarez-Solas et al., 2011; Marcott et al., 2011; Barker et al., 2015; Bassis et al., 2017), so for the purposes of this manuscript we make our differentiation and treat each separately.

The beginning and end of Heinrich Event 1 is less well defined than the beginning and end of Heinrich Stadial 1, but we note that Bernall-Wormull et al. (2021) describe it as lasting from ca. 16.5 ka to ca. 16.0 ka and that Pérez-Mejías et al. (2021) identify it as beginning at 16.17 ka and ending at 15.89 ka. Here, in a revised version of this manuscript, we will adopt a timing for HE1 of from ca. 16.5 ka to ca. 15.9 ka, to provide a more conservative timing of this event (compared with our original interpretation of 16.22 ± 0.04 ka to 16.04 ± 0.04 ka), as suggested by the reviewer. |
| **RC 66** | Additionally, there is a long-standing discussion of Heinrich Stadial 1 in the western US and whether it contained 2 phases  - one that was overall drier- particularly in the southwest, and one that was much wetter centered on 16 ka. This discussion is |

| | |
|---|---|
| | ongoing, but much evidence has come from the timing of lake high stands and how they have varied across the region (Broecker and Putnam, 2012; McGee et al., 2018; Hudson et al., 2019; Oster et al., 2020). At any rate, the development of Heinrich Stadial 1 in this region has been well-explored in the literature, and that discussion should be reflected here.

Following this idea, the discussion on Great Basin Lakes (Section 5.5.3) is oversimplified and under-cited. There have been very nice compilations of Great Basin Lake high stands that include analysis of the timing and geographic patterns and include modeling and other regional syntheses (McGee et al., 2018; Hudson et al., 2019). The presentation of the lake data in Figure 8d is also oversimplified which carries through to the graphical abstract. Only some of the lakes are labeled. It is unclear if only one age is provided per lake or if there are more. This needs to be clarified in the figure and caption. |
| AR 66 | We agree that the discussion regarding a two-phase Heinrich Stadial 1 in the western U.S. is poorly reviewed in the manuscript and that the manuscript should be strengthened in that regard. Likewise, we agree that the manuscript's section on Great Basin lakes is oversimplified and under-cited. We will strengthen this section of the manuscript by highlighting more of the literature on this topic and interpretations therewithin.  Again, however, our manuscript is focused primarily on the relationship between Sierra Nevadan deglaciation and Heinrich *Event* 1, not Heinrich Stadial 1.

With regards to the labeling of the lakes in Figure 8d and in the graphical abstract, and with regards to the number of dates per lake, we will more explicitly refer readers to Munroe and Laabs (2013) and Ibarra et al. (2014) for this information (or, alternatively, refer readers to our supplement for this information). We feel that including this level of detail in the graphical abstract and Figure 8d would complicate the figures and make them harder to interpret for the reader – yet we fully agree that readers should either have this information or know where to find it. |
| RC 67 | The 6-degree jump (lines 949-950 and other places) appears from Figure 8 to be defined by the high stands of Lake Cochise and Lake Russell or Surprise. However, there are numerous other high stands on your figure prior to 16.4 ka that are further north than Cochise. Cochise and Estancia are also outside of the Great Basin. It is not consistent with the uncertainty on the ages of lake high stands to pinpoint 6 degree jump within a 200 year interval. |
| AR 67 | First, the reviewer is correct that Lakes Cochise and Estancia do not lie within the Great Basin. We will correct this oversight in a revised version of the manuscript by renaming section 5.5.3 "Western U.S. paleolakes" (or equivalent), renaming panel (d) of Figure 8 "Western U.S. lake-level highstands" (or equivalent), and by replacing references to "Great Basin lakes" with references to "Western U.S. lakes". |

Second, while there were numerous lake-level highstands in what is now the western United States between 20.0 ka and 16.4 ka that were further north than 38° N (the latitude of McLean's Cave), the 6° latitudinal jump described in the text and shown on Fig. 8d is in the *southern limit* of lake-level highstands (as stated on line 949 of the preprint). Older highstands at more northerly latitudes do not change the observation that there are no lake-level highstands in the dataset at latitudes less than 38° N after the ca. 15.7 ka highstand of Lake Russell.

Finally, with regards to the 6° northward jump in the southern limit of lake-level highstands, it is in the preprint defined by three data points: the highstands of (1) Lake Cochise (at 32° N) and (2) Lake Estancia (at 35° N) and by (3) the ca. 16.2 ka $\delta^{13}C$ minimum in the McLean's Cave speleothem. However, while falling lake levels and rising $\delta^{13}C$ could justifiable be used to argue for a 6° northward shift in the *winter-storm track* or a similar meteorological phenomenon, using the $\delta^{13}C$ minimum as evidence for a 6° northward shift in the southern limit of *lake-level highstands* as we have done was over simplified.

In a revised version of the manuscript, we will redefine the 6° northward shift in the southern limit of lake-level highstands to be based on the highstands of (1) Lake Cochise (at 32° N), (2) Lake Estancia (at 35° N), (3) Owens Lake (at 36° N), and (4) Lake Russell (at 38° N). This revised definition will also result in a more conservative ca. 700-year duration for the 6° northward shift in southern limit of lake-level highstands, rather than the ca. 200-year duration we interpreted in the preprint.

| | |
|---|---|
| **RC 68** | Provide a citation for the hypothesis on the delay of Owens and Russell high stands due to meltwater (Lines 956-958). It is stated as a fact here but with a question mark in your figure – is this a hypothesis put forward by this paper or elsewhere? |
| **AR 68** | We are aware of two relevant citations. In the first, Zimmerman et al. (2011, p. 270) says, "*millennial-scale IRD variability during lake highstands may be the effect of glacial melting.*" We argue, by extension, that millennial-scale IRD variability would be associated with millennial-scale variability in meltwater input and thus millennial-scale variability in lake levels.

In the second, Munroe and Laabs (2013, p. 56) discuss why some lakes in the southern and southwestern Great Basin obtained lake-level highstands at the same time as the larger lakes in the northern Great Basin – and note "*some southwestern lakes might also have been influenced by… additions of glacial meltwater (Owens, and by downstream connection, Searles).*" We argue that Lake Russell should also be listed here, as it was also fed by glacial meltwater (e.g., Russell, 1889; Wahrhaftig et al., 2019) and its ca. 15.7 ka highstand was also included in the Munroe and Laabs (2013) compilation. |

| | |
|---|---|
| | In revising this section of the manuscript, our primary interpretation is that the southern limit of lake-level highstands shifted 6° northward over ca. 700 years (as mentioned in AR 67) – while noting that the atmospheric reorganization responsible for this migration in lake-level highstands may have potentially occurred in as little as 200 years, using the onset of rising $\delta^{13}$C in the McLean's Cave speleothem at ca. 16.2 ka as reflecting the start of drier conditions. We will cite Zimmerman et al. (2011) and Munroe and Laabs (2013) when mentioning that the highstands of Lake Russell and Owens Lake were potentially delayed by meltwater input. With regards to the statement on Fig. 8d that says "Highstands delayed by meltwater release from SN glaciers?", we will (1) retain the question mark and (2) change the statement's font color to gray, to visually deemphasize it. |
| **RC 69** | Regarding your McLean's Cave age model, there is a paper currently in review that includes new U-series dates for McLean's Cave and a new age model run using the COPRA algorithm (Breitenbach et al., 2012), that I believe shifts the minimum in d$^{13}$C of this record slightly older than what your Bchron age model put it at and closer to 16.25 ka. I realize this information is not yet accessible, but caution against putting too much stock in the exact timing of the shift in McLean's Cave $\delta^{13}$C computed from the Bchron model here. This is consistent with the comment to broaden the constraints of the timing on Heinrich Stadial 1 used in this paper, which can be done with the records available in the literature. |
| **AR 69** | We agree with the reviewer and, as noted above, will change the duration Heinrich Event 1 from 16.22–16.04 ka (as it is in the preprint) to being between ca. 16.5 ka and ca. 15.9 ka. |
| **RC 70** | I am also curious about the emphasis on the ice sheet thinning rather than changes in freshwater flux and AMOC that are frequently modeled and evaluated in discussions of Heinrich Event impacts. This comes through in the discussion of the TrACE results (Example Lines 1005 to 1017) which misses the influx of meltwater to the North Atlantic and the subsequent impacts to AMOC. Other modeled scenarios of Heinrich 1 (McGee et al., 2018; Oster et al., 2023) investigate the cascading influences of this freshwater flux to the North Atlantic. |
| **AR 70** | McGee et al. (2018) and Oster et al. (2023) were focused on the Heinrich *Stadials*, not Heinrich *Events*, which we differentiate (based on prior literature) and will better highlight in the revised version of this manuscript. We agree that Heinrich Stadials are closely linked with meltwater flux into the North Atlantic and its impacts upon AMOC, and that our manuscript should mention that fact.

Proxy evidence focusing on HE1 (and not on HS1) documents this event as happening between ca. 16.5 ka and ca. 15.9 ka (e.g., Bernal-Wormull et al., 2021; Pérez-Mejías et al., 2021) – just when the $\delta^{13}$C trend in the McLean's Cave |

| | |
|---|---|
| | speleothem reverses, and not at the beginning of Heinrich Stadial 1, which was at ca. 19–18 ka (e.g., McManus et al., 2004; Heath et al., 2018; Martin et al., 2023). |
| RC 71 | It was surprising to me to equate the changes noted around 16 ka in the proxy records from western North America with the shift in the ice sheet parameterized with ICE-5G between 15 ka and 14 ka in the model simulations. There are proxy records from western North America that cover this interval, including the records in your Figure 8. It is more appropriate to compare the atmospheric changes and drivers in TrACE at 16 ka to the proxy records from 16 ka. |
| AR 71 | Our intention with this passage was to simply highlight the sensitivity of atmospheric circulation to changes in ice sheet geometry in a state-of-the-art climate model. In our revised manuscript, we will address the reviewer's comment by removing the discussion of timing and focus simply on the response of the atmosphere to the LIS's evolving geometry. |
| RC 72 | I think the authors should consider the influences of meltwater flux, SST changes and AMOC shifts and those teleconnections which do align in timing with the shifts seen in western North American proxy records (McGee et al., 2018; Oster al., 2023). |
| AR 72 | In revising the manuscript, we will expand the discussion of Heinrich Stadial 1 in the western United States, as described in AR 66. In doing so, we will describe the evidence for a two-phase HS1 in what is now the western United States, with first phase (HS1a) lasting from 18.0 ka to 16.1 ka and the second phase (HS1b) lasting from 16.1 ka to 14.6 ka (e.g., Smolen et al., 2025). Although, as mentioned above (Ars 65 and 70), we differentiate between Heinrich Stadials and the Heinrich Events themselves (e.g., Martin et al., 2023). |
| RC 73 | Below are more targeted comments by line:

Abstract:

Line 19 – Does the "60% LGM length" refer to the length of glaciers? Not clear. |
| AR 73 | Yes. We will make this clearer. Also, as it happens, Reviewer #1 has suggested that we replace most of these % LGM-glacier length estimates with ELA estimates and we plan on making that change, as discussed in our responses to their comments. |
| RC 74 | Overall abstract is too specific with place names and is difficult to follow. |
| AR 74 | We (1) removed the place names "Swamp Lake" and "McLean's Cave" from the abstract and (2) revised it for clarity. Please see AR 10 (in our response to Reviewer #1) for the revised abstract. |

| | |
|---|---|
| **RC 75** | Graphical Abstract – For the great basin highstands – which lakes are you talking about here – what are the ages from? The citations you have are for papers that include multiple lakes, and the high stands vary across the region. Are the ages for individual lake highstands or are they curves for different lakes (multiple points per lake). (Similar comments on Figure 8). |
| **AR 75** | With regards to the graphical abstract, we think it should remain uncluttered and easy to read.

With regards to Fig. 8d, we propose clarifying these points in the caption and in the supplement. We think labeling all the lakes by name in Fig. 8d (and especially in the graphical abstract) would clutter the figure and make it harder to read – but we agree that specialists should be able to find this information in our manuscript, or find a redirection in our manuscript to where this information can be found. |
| **RC 76** | Introduction:

The discussion of timing in the introduction needs to be more concrete and include the available age information of the time periods discussed leading into this study. In part this is presented in the setting section, but the information should be included in the introduction to set up your central questions/objectives. |
| **AR 76** | Reviewer #1 raised similar points (RCs 14, 15, 16, 24, and 34) – and it indicates that the glacial history of the region (the Tioga 1-4 and Recess Peak terms) needs to be moved up from the "Regional setting" section and into the Introduction. We agree with the reviewers and will make this change. |
| **RC 77** | -Line 63 – provide a time range to go with "broadly in-phase with the global Last Glacial Maximum". |
| **AR 77** | Reviewer #1 also raised this issue (RC 15) and we will make this change. |
| **RC 78** | Lines 66-71 – please also provide timing estimates for these advances and retreats based on the references included here. It is important to situate your discussion of Heinrich 1 in time. |
| **AR 78** | We agree and will include the estimates and references in the revised manuscript. |
| **RC 79** | -Line 82 – first mention of the Tioga 4 glaciation – need to define what the timing of this glaciation is understood to be for non-specialists. |
| **AR 79** | We agree. |

| | |
|---|---|
| **RC 80** | -Line224 – do not think you need to include the phrase "is especially relevant to this manuscript". You can explain the proximity to your sites and make the relevance apparent. |
| **AR 80** | We will revise the manuscript as suggested. |
| **RC 81** | Line 617 – it is more typical to report 2 sigma uncertainties for U-series ages. |
| **AR 81** | Thank you for this comment. 1-sigma uncertainties are more common for cosmo dates. Because this paper touches upon both disciplines, we'll be explicit about our level of certainty though the manuscript, clearly identifying 1 sigma vs. 2 sigma uncertainties. |
| **RC 82** | Lines 668-674 – This reads a bit too colloquially – the discussion prior to this centers on why the different calculators may be returning different ages. But then these sentences dismiss this question and say that it is not the central task of this paper to figure this out. Suggest rewriting to this to emphasize what is important, rather than what isn't and to tone down colloquial language. |
| **AR 82** | We will edit the paragraph to emphasize what is important. The new paragraph will read:
 "We hypothesize that the ~5 % age difference is the result of the calculators using different algorithms for converting a suite of calibration samples into a single value (be it a non-dimensional LSDn scaling factor or a SLHL production rate) for scaling to other locations. In this manuscript, we focus on two tasks: (1) assessing which calculator is more likely producing more accurate ages for the samples in the dataset we report; and then (2) placing these dates within the context of previous paleoclimate research." |
| **RC 83** | Overall, I suggest fewer parenthetical observations like the example in Lines 682-683. These are distracting and are used unevenly. Just include ideas in the sentences proper. |
| **AR 83** | We will revise for clarity. The new sentence will read:
 "If we accept the interpretation that Tuolumne Meadows deglaciated before 15.4 ka, as suggested by the Greenstone Lake radiocarbon date (Fig. S2; Clark et al., 2003), and that the high-elevation lake basins of the Sierra Nevada deglaciated at 15.75 ± 0.5 ka (Phillips, 2016), as suggested by the bulk-organic radiocarbon dates (Sect. 2.2.2), then the probability distributions…" |
| **RC 84** | Line 734 – missing word "Accepting the bulk organic radiocarbon dates as accurate **provides** minimum limits…" or something like this. |
| **AR 84** | Thank you for catching this typo. We will correct the passage. |

| | |
|---|---|
| **RC 85** | Lines 741-749 – suggest adding the ages of the outliers into the paragraph in the appropriate places to aid comparison with the mean. |
| **AR 85** | We will incorporate this change. |
| **RC 86** | Line 759 – "probably" should be "probable" |
| **AR 86** | Thank you for catching this typo. We will make this correction. |
| **RC 87** | Lines 783-788 – add the numbers into this narrative so that it is easier to follow and see what your interpreted chronology is. |
| **AR 87** | We will make this correction. |
| **RC 88** | Section 5.4 – it is not clear where the constraints on temperature and precipitation changes given in this section are coming from. It does not appear that independent glacier mass-balance modeling was undertaken for this study. Where do the estimates of 2 degrees and 35% precipitation change come from? The papers cited in this section are not unique to changes in the Sierra Nevada. The Wolfe 1992 paper is also not included in the reference list. The authors should carry out the glacier mass-balance work for their location specifically or at the very least be clearer about how they arrive at these numbers for temperature and precipitation change. They can also find independent estimates of temperature and precipitation changes during this interval from climate modeling work, such as the TRACE simulations mentioned later, to compare with their estimates. |
| **AR 88** | Thank you for catching the missing reference, we will include it in our revised manuscript. With regards to the modeling, the minimum amount of warming required to deglaciate the central Sierra Nevada (assuming no change in precipitation) comes from multiplying the ELA rise responsible for the deglaciation (at least 600 m; Phillips, 2016) by a lapse rate of 3 °C km$^{-1}$, which is the lower end of the values observed by Wolfe (1992) in the Sierra Nevada and Cascade ranges.

The minimum reduction in precipitation to deglaciate the range (assuming no change in temperature) comes from the corollary of Oerlemans's (2005) statement that a 25 % increase in precipitation is required to offset a 1 °C warming. To derive the estimated winter precipitation change, we simply convert the 1.8 °C warming from our ELA calculation above to a minimum-precipitation-reduction of ~35 % applying the estimation of Oerlemans (2005).

We will add a section to the supplement describing this methodology. |

| | |
|---|---|
| **RC 89** | Lines 888-889 – The Great Basin lakes that are included in this comparison need to be listed by name and latitude as the timing of lake high stands varies with latitude during the deglaciation (see for example McGee et al., 2018). |
| **AR 89** | We will add a table to the supplement and refer to it as needed in the main text. |
| **RC 90** | Lines 892-893 – Again suggest that you expand to other North Atlantic records to document the timing of the Heinrich Event and not the regional Ostolo Cave record on its own. |
| **AR 90** | We will strengthen the manuscript here by (1) clearly differentiating Heinrich Event 1 from Heinrich Stadial 1 and (2) citing other records from the North Atlantic region and elsewhere that demonstrate that Heinrich Event 1 occurred between ca. 16.5 ka and ca. 15.9 ka (e.g., Ridge et al., 2012; Deplazes et al., 2013; Pérez-Mejías et al., 2021; Martin et al., 2023). |
| **RC 91** | Line 903 – change the partial derivative symbol being used in the carbon isotope notation to a lower-case delta symbol |
| **AR 91** | Thank you for catching the error. |
| **RC 92** | 930-931 – As stated above, 16.2 is not a defined beginning of Heinrich Event 1, rather there are records from the North Atlantic indicating that IRD and AMOC slowdown began well before this |
| **AR 92** | As discussed above, we will emphasize the difference between Heinrich Events and Heinrich Stadials to limit any future confusion. |
| **RC 93** | Line 938 – It is not clear where the estimate of a 40% reduction in North Pacific moisture is coming from. Please provide citations or analysis to support this number. |
| **AR 93** | We will add the following passage to the supplement:

"In making this calculation, of a 40 % reduction in North Pacific moisture, we assume the moisture arriving in the west-central Sierra Nevada from North Pacific and the tropics during the deglacial period had $\delta^{18}O$ values of -14.0 ± 0.4 ‰ and -4.9 ± 2.5 %, respectively. These distributions are the modern (2006–2011) distributions (Oster et al., 2012) uniformly adjusted +1.5 ‰, to reflect the deglacial ocean's higher $\delta^{18}O$ value (as a result of the preferential accumulation of $^{16}O$ in glaciers and ice sheets). Our adjustment of +1.5 ‰ offsets the observed decrease in benthic $\delta^{18}O$ from ca. 16 ka to the present (Lisiecki and Raymo, 2005). These adjusted $\delta^{18}O$ distributions suggest North Pacific moisture delivery to the west-central Sierra Nevada decreased |

| | |
|---|---|
| | from 54 % of all moisture to 31 %, a 43% relative reduction in North Pacific moisture, which we round to ~40 %." |
| **RC 94** | Lines 941-942 – The increase in speleothem d$^{13}$C beginning at ~16.25 ka does imply that conditions may start to become drier, but overall the McLean's Cave record indicates relatively wet conditions between ~16.4 and 16 ka. |
| **AR 94** | Thank for suggesting the needed clarification. We are suggesting drying relative to the conditions at that time – albeit not necessarily arid. The reduction in precipitation – if it occurred during winter – would reduce glacier mass balances, even if the conditions remained wet relative to later on / more modern conditions. As such we changed the sentence to now read as (new text in blue): *"Thus, the δ$^{13}$C record implies relative drying of the west-central Sierra Nevada starting at 16.20 ± 0.13 ka and the δ$^{18}$O record permits both it and warming."* |
| **RC 95** | Lines 986-989 – Again, these dates are not an appropriate choice for the start of the Heinrich event based on evidence form the North Atlantic. |
| **AR 95** | As mentioned above, we will alter our manuscript to clearly define the terms and differences between Heinrich Stadials and Heinrich Events. We will adopt the more conservative start date of ca. 16.5 ka for HE1. |
| **RC 96** | Lines 1018-1028: This section linking the WAIS divide record to the tropical Pacific is missing the subsequent link to western North American climate. If this is important, include a description of this mechanism with citations. The last two lines "implying a substantial thinning of the LIS" require citations. |
| **AR 96** | Thank you for prompting us to think more about this issue. The discussion about WAIS divide is not essential to the manuscript and we will remove it. |
| **RC 97** | Figure 9 – illustrates the proposed ice sheet thinning –While the majority of the text suggests a big change at 16.2 – this figure shows the resultant ice sheet much later at 15.6 ka. This is also based on another reconstruction than ICE-5G, while ICE-5G is emphasized earlier. I'm not sure how much this figure adds given the inconsistencies with the timing and other aspects of the hypothesis. |
| **AR 97** | This figure is based on Art Dyke's reconstructions of LIS *extent* over time (Dyke et al., 2003). Dyke et al. (2003) provides ice-extent reconstructions for 16.3 ka and 15.6 ka. These are the LIS ice-extent reconstructions that most closely bracket HE1. We will clarify that there isn't a Dyke et al. (2003) (or other) ice-extent reconstruction for ca. 16.2 ka, the most likely time for when HE1 began (e.g., Bernal-Wormull et al., 2021; Pérez-Mejías et al., 2021; Martin et al., 2023). |

ICE-5G is the ice-*thickness* reconstruction used by the TrACE-21k model. Thus, when discussing ice thicknesses within the context of the TrACE experiment we must discuss the ICE-5G model. That said, Dyke et al. (2003) is a more granular and precise reconstruction of LIS ice extent over time. We will clarify why we are discussing both in the text.

**References Cited in this Response**

Álvarez-Solas, J., Montoya, M., Ritz, C., Ramstein, G., Charbit, S., Dumas, C., Nisancioglu, K., Dokken, T., and Ganopolski, A.: Heinrich event 1: an example of dynamical ice-sheet reaction to oceanic changes, Climate of the Past, 7, 1297–1306, 2011.

Andrews, J. T. and Voelker, A. H.: "Heinrich events"(& sediments): A history of terminology and recommendations for future usage, Quaternary Science Reviews, 187, 31–40, 2018.

Barker, S., Chen, J., Gong, X., Jonkers, L., Knorr, G., and Thornalley, D.: Icebergs not the trigger for North Atlantic cold events, Nature, 520, 333–336, 2015.

Bartlein, P. J., Anderson, K. H., Anderson, P. M., Edwards, M. E., Mock, C. J., Thompson, R. S., Webb, R. S., Webb III, T., and Whitlock, C.: Paleoclimate simulations for North America over the past 21,000 years: features of the simulated climate and comparisons with paleoenvironmental data, Quaternary science reviews, 17, 549–585, 1998.

Bassis, J. N., Petersen, S. V., and Mac Cathles, L.: Heinrich events triggered by ocean forcing and modulated by isostatic adjustment, Nature, 542, 332–334, 2017.

Bernal-Wormull, J. L., Moreno, A., Pérez-Mejías, C., Bartolomé, M., Aranburu, A., Arriolabengoa, M., Iriarte, E., Cacho, I., Spötl, C., and Edwards, R. L.: Immediate temperature response in northern Iberia to last deglacial changes in the North Atlantic, Geology, 49, 999–1003, 2021.

Chiang, J. C., Lee, S.-Y., Putnam, A. E., and Wang, X.: South Pacific Split Jet, ITCZ shifts, and atmospheric North–South linkages during abrupt climate changes of the last glacial period, Earth and Planetary Science Letters, 406, 233–246, 2014.

Clark, P. U. and Bartlein, P. J.: Correlation of late Pleistocene glaciation in the western United States with North Atlantic Heinrich events, Geology, 23, 483–486, 1995.

Deplazes, G., Lückge, A., Peterson, L. C., Timmermann, A., Hamann, Y., Hughen, K. A., Röhl, U., Laj, C., Cane, M. A., and Sigman, D. M.: Links between tropical rainfall and North Atlantic climate during the last glacial period, Nature Geoscience, 6, 213–217, 2013.

Dowdeswell, J. A., Maslin, M. A., Andrews, J. T., and McCave, I. N.: Iceberg production, debris rafting, and the extent and thickness of Heinrich layers (H-1, H-2) in North Atlantic sediments, Geology, 23, 301–304, 1995.

Francois, R. and Bacon, M. P.: Heinrich events in the North Atlantic: radiochemical evidence, Deep Sea Research Part I: Oceanographic Research Papers, 41, 315–334, 1994.

Heath, S. L., Loope, H. M., Curry, B. B., and Lowell, T. V.: Pattern of southern Laurentide Ice Sheet margin position changes during Heinrich Stadials 2 and 1, Quaternary Science Reviews, 201, 362–379, 2018.

Hemming, S. R.: Heinrich events: Massive Late Pleistocene detritus layers of the North Atlantic and their global climate imprint, Reviews of Geophysics, 42, 2003RG000128, https://doi.org/10.1029/2003RG000128, 2004.

Jones, T. R., Roberts, W. H., Steig, E. J., Cuffey, K. M., Markle, B. R., and White, J. W. C.: Southern Hemisphere climate variability forced by Northern Hemisphere ice-sheet topography, Nature, 554, 351–355, 2018.

Lora, J. M. and Ibarra, D. E.: The North American hydrologic cycle through the last deglaciation, Quaternary Science Reviews, 226, 105991, 2019.

Marcott, S. A., Clark, P. U., Padman, L., Klinkhammer, G. P., Springer, S. R., Liu, Z., Otto-Bliesner, B. L., Carlson, A. E., Ungerer, A., Padman, J., He, F., Cheng, J., and Schmittner, A.: Ice-shelf collapse from subsurface warming as a trigger for Heinrich events, Proceedings of the National Academy of Sciences, 108, 13415–13419, https://doi.org/10.1073/pnas.1104772108, 2011.

Martin, K. C., Buizert, C., Edwards, J. S., Kalk, M. L., Riddell-Young, B., Brook, E. J., Beaudette, R., Severinghaus, J. P., and Sowers, T. A.: Bipolar impact and phasing of Heinrich-type climate variability, Nature, 617, 100–104, 2023.

McGee, D., Moreno-Chamarro, E., Marshall, J., and Galbraith, E. D.: Western U.S. lake expansions during Heinrich stadials linked to Pacific Hadley circulation, Science Advances, 4, eaav0118, https://doi.org/10.1126/sciadv.aav0118, 2018.

McManus, J. F., Francois, R., Gherardi, J.-M., Keigwin, L. D., and Brown-Leger, S.: Collapse and rapid resumption of Atlantic meridional circulation linked to deglacial climate changes, nature, 428, 834–837, 2004.

Munroe, J. S. and Laabs, B. J.: Temporal correspondence between pluvial lake highstands in the southwestern US and Heinrich Event 1, Journal of Quaternary Science, 28, 49–58, 2013.

Oerlemans, J.: Extracting a climate signal from 169 glacier records, Science, 308, 675–677, 2005.

Oster, J. L., Macarewich, S., Lofverstrom, M., De Wet, C., Montañez, I., Lora, J. M., Skinner, C., and Tabor, C.: North Atlantic meltwater during Heinrich Stadial 1 drives wetter climate with more atmospheric rivers in western North America, Science Advances, 9, https://doi.org/10.1126/sciadv.adj2225, 2023.

Pérez-Mejías, C., Moreno, A., Bernal-Wormull, J., Cacho, I., Osácar, M. C., Edwards, R. L., and Cheng, H.: Oldest Dryas hydroclimate reorganization in the eastern Iberian Peninsula after the iceberg discharges of Heinrich Event 1, Quaternary Research, 101, 67–83, 2021.

Phillips, F. M.: Cosmogenic nuclide data sets from the Sierra Nevada, California, for assessment of nuclide production models: I. Late Pleistocene glacial chronology, Quaternary Geochronology, 35, 119–129, 2016.

Phillips, F. M.: Glacial chronology of the Sierra Nevada, California, from the last glacial maximum to the Holocene, Cuadernos de investigación geográfica: Geographical Research Letters, 43, 527–552, 2017.

Ridge, J. C., Balco, G., Bayless, R. L., Beck, C. C., Carter, L. B., Dean, J. L., Voytek, E. B., and Wei, J. H.: The new North American Varve Chronology: A precise record of southeastern Laurentide Ice Sheet deglaciation and climate, 18.2-12.5 kyr BP, and correlations with Greenland ice core records, American Journal of Science, 312, 685–722, 2012.

Roy, K. and Peltier, W. R.: Glacial isostatic adjustment, relative sea level history and mantle viscosity: reconciling relative sea level model predictions for the US East coast with geological constraints, Geophysical Journal International, 201, 1156–1181, 2015.

Roy, K. and Peltier, W. R.: Space-geodetic and water level gauge constraints on continental uplift and tilting over North America: regional convergence of the ICE-6G_C (VM5a/VM6) models, Geophysical Journal International, 210, 1115–1142, 2017.

Roy, K. and Peltier, W. R.: Relative sea level in the Western Mediterranean basin: a regional test of the ICE-7G_NA (VM7) model and a constraint on late Holocene Antarctic deglaciation, Quaternary Science Reviews, 183, 76–87, 2018.

Russell, I. C.: Quaternary history of Mono Valley, California, US Government Printing Office, 1889.

Smolen, J. D., Burstyn, Y., Wang, Z., Reichgelt, T., de Wet, C., Montañez, I. P., Hren, M. T., Atekwana, E. A., Bowen, G. J., and Griffith, E. M.: Organic molecular records of fire–hydroclimate–vegetation dynamics through the last deglaciation archived in a California stalagmite, Geology, 2025.

Stern, J. V. and Lisiecki, L. E.: North Atlantic circulation and reservoir age changes over the past 41,000 years, Geophysical Research Letters, 40, 3693–3697, https://doi.org/10.1002/grl.50679, 2013.

Thomson, J., Higgs, N. C., and Clayton, T.: A geochemical criterion for the recognition of Heinrich events and estimation of their depositional fluxes by the (230Thexcess) 0 profiling method, Earth and Planetary Science Letters, 135, 41–56, 1995.

Ullman, D. J., LeGrande, A. N., Carlson, A. E., Anslow, F. S., and Licciardi, J. M.: Assessing the impact of Laurentide Ice Sheet topography on glacial climate, Climate of the Past, 10, 487–507, 2014.

Wahrhaftig, C., Stock, G. M., McCracken, R. G., Sasnett, P., and Cyr, A. J.: Extent of the Last Glacial Maximum (Tioga) glaciation in Yosemite National Park and vicinity, California, US Geological Survey Scientific Investigations Series Map, 3414, 2019.

Wolfe, J. A.: An analysis of present-day terrestrial lapse rates in the western conterminous United States and their significance to paleoaltitudinal estimates, US Geological Survey, 1992.

Yanase, W. and Abe-Ouchi, A.: A numerical study on the atmospheric circulation over the midlatitude North Pacific during the last glacial maximum, Journal of Climate, 23, 135–151, 2010.

Zimmerman, S. R., Pearl, C., Hemming, S. R., Tamulonis, K., Hemming, N. G., and Searle, S. Y.: Freshwater control of ice-rafted debris in the last glacial period at Mono Lake, California, USA, Quaternary Research, 76, 264–271, 2011.